# Risk Bounds of Accelerated SGD for Overparameterized Linear Regression

**Xuheng Li**[1], **Yihe Deng**[1], **Jingfeng Wu**[2], **Dongruo Zhou**[3], **Quanquan Gu**[1]

[1]Department of Computer Science, University of California, Los Angeles, CA 90095, USA
[2]Simons Institute, University of California, Berkeley, CA 94720, USA
[3]Department of Computer Science, Indiana University Bloomington, IN 47408, USA
`xuheng.li@cs.cula.edu, yihedeng@cs.ucla.edu,`
`uuujf@berkeley.edu, dz13@iu.edu, qgu@cs.ucla.edu`

## Abstract

Accelerated stochastic gradient descent (ASGD) is a workhorse in deep learning and often achieves better generalization performance than SGD. However, existing optimization theory can only explain the faster convergence of ASGD, but cannot explain its better generalization. In this paper, we study the generalization of ASGD for overparameterized linear regression, which is possibly the simplest setting of learning with overparameterization. We establish an instance-dependent excess risk bound for ASGD within each eigen-subspace of the data covariance matrix. Our analysis shows that (i) ASGD outperforms SGD in the subspace of small eigenvalues, exhibiting a faster rate of exponential decay for bias error, while in the subspace of large eigenvalues, its bias error decays slower than SGD; and (ii) the variance error of ASGD is always larger than that of SGD. Our result suggests that ASGD can outperform SGD when the difference between the initialization and the true weight vector is mostly confined to the subspace of small eigenvalues. Additionally, when our analysis is specialized to linear regression in the strongly convex setting, it yields a tighter bound for bias error than the best-known result.

## 1 Introduction

Momentum (Nesterov, 1983) is an important technique in optimization. In the context of convex and smooth optimization, Nesterov's momentum (accelerated gradient descent (AGD)) achieves the minimax optimal convergence rate (Nesterov, 2014) and provably accelerates the vanilla GD method. Recent work by Liu & Belkin (2018) shows that stochastic gradient descent (SGD) can also be accelerated by momentum in the overparameterized setting. However, the effect of momentum on the generalization performance is less studied. It has been empirically shown that ASGD does not always outperform SGD (Wang et al., 2023), but there has been little theoretical work justifying this observation. Notable exceptions are Jain et al. (2018) and Varre & Flammarion (2022), which provide excess risk bounds for accelerated SGD (ASGD) (a.k.a., SGD with momentum) for least squares problems in the strongly convex (Jain et al., 2018) and convex settings (Varre & Flammarion, 2022), respectively. However, both of their results are limited to the classical, finite-dimensional regime, and cannot be applied when the number of parameters exceeds the number of samples. On the other hand, a recent line of work completely characterizes the excess risk of SGD for least squares, even in the overparameterized regime (Dieuleveut & Bach, 2015; Défossez & Bach, 2015; Jain et al., 2017b; Berthier et al., 2020; Zou et al., 2021b; Wu et al., 2022). In particular, Zou et al. (2021b); Wu et al. (2022) provide finite-sample and dimension-free excess risk bounds for SGD that are sharp for each least squares instance. Given these results, it becomes imperative to thoroughly investigate whether the inclusion of momentum proves beneficial in terms of generalization, particularly in the context of least squares problems.

**Contributions.** In this paper, we tackle the question by considering ASGD for (overparameterized) linear regression problems and comparing its performance with SGD.

- Our main result provides an instance-dependent excess risk bound for ASGD that can be applied in the overparameterized regime. Similar to the bounds for SGD in Zou et al. (2021b); Wu et al. (2022), our bound for ASGD is independent of the ambient dimension and comprehensively depends on the spectrum of the data covariance matrix. When applied to the classical,

strongly-convex regime, our results recover the excess risk upper bounds in Jain et al. (2018), with significant improvements on the coefficient of the bias error.[1]

- Based on the excess risk bounds, we then compare the excess risk of ASGD and SGD. We find that the variance error of ASGD is always no smaller than that of SGD. Moreover, the bias error of ASGD is smaller than that of SGD along the small eigenvalue directions, but is larger than that of SGD along the large eigenvalue directions, with respect to the spectrum of the data covariance matrix. Thus momentum can help with generalization only if the main signals are aligned with small eigenvalue directions of the data covariance matrix and if the noise is small.

- From a technical perspective, we extend the analysis of the stationary covariance matrix in Jain et al. (2018) to the overparameterized setting, where we remove all dimension-dependent factors with a fine-grained analysis of the ASGD iterates. Our techniques might be of independent interest for analyzing ASGD in other settings.

**Notation.** In this paper, scalars are denoted by non-boldface letters. Vectors and matrices are denoted by lower-case and upper-case boldface letters, respectively. Denote linear operators on matrices by upper-case calligraphic letters. Denote the inner product of vectors by $\langle \mathbf{u}, \mathbf{v} \rangle$. For a vector $\mathbf{v}$, denote its $j$-th entry as $(\mathbf{v})_j$; For a matrix $\mathbf{M}$, denote its $ij$-entry as $(\mathbf{M})_{ij}$. For a PSD matrix $\mathbf{M}$, define $\|\mathbf{u}\|_{\mathbf{M}}^2 = \mathbf{u}^\top \mathbf{M} \mathbf{u}$. Denote the 2-norm of vector $\mathbf{v}$ as $\|\mathbf{v}\|_2 = \sqrt{\mathbf{v}^\top \mathbf{v}}$. Denote the inner product of matrices $\mathbf{A}, \mathbf{B} \in \mathbb{R}^{2d \times 2d}$ as $\langle \mathbf{A}, \mathbf{B} \rangle = \sum_{i,j=1}^{2d} (\mathbf{A})_{ij} (\mathbf{B})_{ij}$. The Kronecker product of matrices is denoted by $\otimes$. The operation of a linear matrix operator on a matrix is denoted by $\circ$.

## 2 RELATED WORK

The generalization performances of SGD and ASGD applied to *underparameterized* linear regression have been studied in a line of works, based on the technique of bias-variance decomposition. It is shown that for SGD with iterate averaging from the beginning, bias error has a convergence rate of $\mathcal{O}(1/N^2)$ and variance has a convergence rate of $\mathcal{O}(d/N)$, where $N$ is the number of calls of the stochastic oracle and $d$ is the model dimension (Défossez & Bach, 2015; Dieuleveut et al., 2017; Jain et al., 2017a). If the eigenvalue of the data covariance matrix is bounded away from zero, then the convergence rate of the bias error can be further improved with additional exponential shrinkage by taking tail averaging of the iterates (Jain et al., 2017b).

For ASGD applied to linear regression, there are two cases: one with the assumption that the eigenvalue spectrum of the data covariance matrix is bounded away from zero (strongly convex) and the other without such assumption (general convex). For strongly convex linear regression, Jain et al. (2018) show an accelerated convergence rate for the bias error of ASGD with constant stepsize and tail averaging, compared to that of tail-averaged SGD in Jain et al. (2017b). We extend the use of linear operators and the techniques for bounding the operator spectrum in Jain et al. (2018).

Recently, the generalization of ASGD applied to general convex linear regression is studied by Varre & Flammarion (2022). Their result shows the acceleration of ASGD with time-varying parameters and weighted iterate averaging, especially for large $N$. The case of general convex linear regression is closer to the overparameterized setting where fast-decaying eigenspectrum is of special interest. However, their result is not applicable to the overparameterized linear regression because of the dimensionality dependence. Additionally, their result does not reveal the exponential bias decay of ASGD with constant stepsize.

The generalization performance of overparameterized linear regression has been studied by a line of works (Bartlett et al., 2020; Tsigler & Bartlett, 2020). For SGD applied to overparameterized linear regression, Zou et al. (2021b) replace the model dimensionality $d$ with the effective dimension defined in terms of the eigenspectrum. This work manages to deal with any data covariance matrix, while prior works require certain assumptions (Dieuleveut & Bach, 2015). Wu et al. (2022) show a similar result for the last iterate of SGD with exponentially decaying stepsize.

## 3 PRELIMINARIES

### 3.1 LINEAR REGRESSION AND ASGD

The goal of linear regression is to minimize the following risk:

$$L(\mathbf{w}) := 1/2 \cdot \mathbb{E}_{(\mathbf{x}, y) \sim \mathcal{D}} \left[ (y - \langle \mathbf{w}, \mathbf{x} \rangle)^2 \right],$$

---

[1]Our excess risk bound contains an extra term, which can be removed by a fine-grained analysis used by Jain et al. (2018) in the classical regime.

where $\mathbf{x}$ is an input feature vector belonging to a Hilbert space (denoted by $\mathcal{H}$, which could be either $d$-dimensional for a finite $d$, or countably infinite dimensional), $y \in \mathbb{R}$ is the response, $\mathbf{w} \in \mathcal{H}$ is the weight vector to be optimized, and $\mathcal{D}$ is an underlying unknown distribution of the data.

We consider the ASGD algorithm with tail averaging. In detail, in the $t$-th iteration, a sample $(\mathbf{x}_t, y_t) \sim \mathcal{D}$ is observed. Then the stochastic gradient is calculated by

$$\widehat{\nabla} L(\mathbf{w}) = -(y_t - \langle \mathbf{w}, \mathbf{x}_t \rangle) \mathbf{x}_t. \tag{3.1}$$

We follow the classical ASGD scheme (Nesterov, 2014), which maintains three sequences $\mathbf{w}_t$, $\mathbf{v}_t$ and $\mathbf{u}_t$. Let $N$ be the number of samples observed, then for any $1 \le t \le N$, the update rules of $\mathbf{w}_t, \mathbf{v}_t, \mathbf{u}_t$ are as follows.

$$\mathbf{u}_{t-1} = \alpha \mathbf{w}_{t-1} + (1 - \alpha) \mathbf{v}_{t-1}, \tag{3.2}$$

$$\mathbf{w}_t = \mathbf{u}_{t-1} - \delta \widehat{\nabla} L(\mathbf{u}_{t-1}), \tag{3.3}$$

$$\mathbf{v}_t = \beta \mathbf{u}_{t-1} + (1 - \beta) \mathbf{v}_{t-1} - \gamma \widehat{\nabla} L(\mathbf{u}_{t-1}), \tag{3.4}$$

where $\alpha, \beta, \gamma, \delta > 0$ are hyperparameters. The $\mathbf{v}_t$ sequence is initialized at $\mathbf{w}_0 \in \mathcal{H}$. We remark that ASGD reduces to stochastic heavy ball (SHB, Polyak (1964)) when $\delta = 0$, so our results can be directly applied to SHB by setting $\delta = 0$ (see Appendix C for details). We also remark that ASGD reduces to SGD when $\delta = \gamma$.

In this work, following Jain et al. (2018) and Zou et al. (2021b), we consider ASGD with tail averaging. The tail-averaged final output is $\overline{\mathbf{w}}_{s,s+N} := N^{-1} \sum_{t=s}^{s+N-1} \mathbf{w}_t$. With certain assumptions, $L(\mathbf{w})$ admits a unique global optimum denoted by $\mathbf{w}^* := \operatorname{argmin}_{\mathbf{w}} L(\mathbf{w})$. We focus on the overparameterized setting, where $d \gg N$ (or possibly countably infinite).

Define the centered ASGD iterate as $\boldsymbol{\eta}_t := \begin{bmatrix} \mathbf{w}_t - \mathbf{w}^* \\ \mathbf{u}_t - \mathbf{w}^* \end{bmatrix}$. Denote the noise in each sample as $\epsilon_t := y_t - \langle \mathbf{w}^*, \mathbf{x}_t \rangle$. By (3.1), the stochastic gradient at $\mathbf{u}_{t-1}$ can be expressed as

$$\widehat{\nabla} L(\mathbf{u}_{t-1}) = -(\epsilon_t + \langle \mathbf{w}^*, \mathbf{x}_t \rangle - \langle \mathbf{u}_{t-1}, \mathbf{x}_t \rangle) \mathbf{x}_t = \mathbf{x}_t \mathbf{x}_t^\top (\mathbf{u}_{t-1} - \mathbf{w}^*) - \epsilon_t \mathbf{x}_t. \tag{3.5}$$

By substituting (3.5) into (3.3) and (3.4) and eliminating $\mathbf{v}_t$ using (3.2), we have

$$\boldsymbol{\eta}_t = \widehat{\mathbf{A}}_t \boldsymbol{\eta}_{t-1} + \boldsymbol{\zeta}_t, \quad \text{where} \quad \widehat{\mathbf{A}}_t := \begin{bmatrix} \mathbf{0} & \mathbf{I} - \delta \mathbf{x}_t \mathbf{x}_t^\top \\ -c\mathbf{I} & (1+c)\mathbf{I} - q \mathbf{x}_t \mathbf{x}_t^\top \end{bmatrix}, \quad \boldsymbol{\zeta}_t := \begin{bmatrix} \delta \cdot \epsilon_t \mathbf{x}_t \\ q \cdot \epsilon_t \mathbf{x}_t \end{bmatrix},$$

and $c := \alpha(1 - \beta), q := \alpha\delta + (1 - \alpha)\gamma$. Denote the expectation of $\widehat{\mathbf{A}}_t$ as

$$\mathbf{A} := \mathbb{E}[\widehat{\mathbf{A}}_t] = \begin{bmatrix} \mathbf{0} & \mathbf{I} - \delta \mathbf{H} \\ -c\mathbf{I} & (1+c)\mathbf{I} - q\mathbf{H} \end{bmatrix},$$

where $\mathbf{H} = \mathbb{E}_{\mathbf{x} \sim \mathcal{D}|_{\mathbf{x}}}[\mathbf{x}\mathbf{x}^\top]$ is the second-order moment matrix of the distribution $\mathcal{D}$, which is also the Hessian of $L(\mathbf{w})$. Let the eigen-decomposition of the Hessian be $\mathbf{H} = \sum_{i=1}^d \lambda_i \mathbf{v}_i \mathbf{v}_i^\top$, where $\{\lambda_i\}_{i=1}^d$ are the eigenvalues of $\mathbf{H}$ sorted in descending order with $\mathbf{v}_i$'s being the corresponding eigenvectors. Similar to Jain et al. (2018), we assume that $\mathbf{H}$ is diagonal, then $\mathbf{A}$ is block diagonal with each block being $\mathbf{A}_i := \begin{bmatrix} 0 & 1 - \delta\lambda_i \\ -c & 1 + c - q\lambda_i \end{bmatrix}$. In this work, we are particularly interested in analyzing the eigenvalues of $\mathbf{A}_i$, since the spectral norm of $\mathbf{A}_i$ determines the decay rate of the bias error in the subspace of $\lambda_i$.

## 3.2 ASSUMPTIONS

We then introduce assumptions required in our analysis, following those of Zou et al. (2021b); Wu et al. (2022). Our first assumption regularizes the moments of the data distribution.

**Assumption 3.1** (Regularity conditions). The second moment $\mathbf{H}$ exists, and $\operatorname{tr}(\mathbf{H})$ is finite. $\mathbf{H}$ is strictly positive definite, i.e., $\mathbf{H} \succ \mathbf{0}$. Thus, $L(\mathbf{w})$ admits a unique global optimum $\mathbf{w}^*$. The second-order moment of labels $\mathbb{E}[y^2]$ is also finite. Let $\mathcal{M}$ denote the fourth moment of $\mathbf{x}$:

$$\mathcal{M} := \mathbb{E}_{(\mathbf{x},y) \sim \mathcal{D}}[\mathbf{x} \otimes \mathbf{x} \otimes \mathbf{x} \otimes \mathbf{x}].$$

Then $\mathcal{M}$ exists and is finite.

Our second assumption is a proposition of the fourth moment of $\mathbf{x}$, viewed as a linear operator $\mathcal{M}$ on PSD matrices.

**Assumption 3.2** (Fourth moment condition). Assume there exists a positive constant $\psi > 0$, such that for any PSD matrix $\mathbf{A}$, it holds that

$$\mathbb{E}_{\mathbf{x} \sim \mathcal{D}}[\mathbf{x}\mathbf{x}^\top \mathbf{A} \mathbf{x}\mathbf{x}^\top] \preceq \psi \operatorname{tr}(\mathbf{H}\mathbf{A})\mathbf{H}.$$

A special case of Assumption 3.2 is when $\mathcal{D}$ is a Gaussian distribution. For that case, we have $\psi = 3$. We remark that although Assumption 3.2 does not cover some special cases, e.g., the one-hot distribution discussed in Zou et al. (2021a), similar results can still be obtained by applying our techniques with minor modifications (see Appendix J for details).

The following assumption characterizes the noise of the stochastic gradient.

**Assumption 3.3** (Noise condition). Assume that

$$\boldsymbol{\Sigma} := \mathbb{E}_{(\mathbf{x},y) \sim \mathcal{D}}[\widehat{\nabla} L(\mathbf{w}^*) \otimes \widehat{\nabla} L(\mathbf{w}^*)] = \mathbb{E}_{(\mathbf{x},y) \sim \mathcal{D}}[(y - \langle \mathbf{w}^*, \mathbf{x} \rangle)^2 \mathbf{x}\mathbf{x}^\top],$$

and $\sigma^2 := \|\mathbf{H}^{-\frac{1}{2}} \boldsymbol{\Sigma} \mathbf{H}^{-\frac{1}{2}}\|_2$ exist and are finite. Here, $\boldsymbol{\Sigma}$ is the covariance matrix of the gradient noise at $\mathbf{w}^*$. For *well-specified models* where $y_t - \langle \mathbf{w}^*, \mathbf{x}_t \rangle \sim \mathcal{N}(0, \sigma^2_{\text{noise}})$, we have $\boldsymbol{\Sigma} = \sigma^2_{\text{noise}}\mathbf{H}$ and thus $\sigma^2 = \sigma^2_{\text{noise}}$.

## 4 MAIN RESULTS

We now provide an excess risk upper bound for ASGD.

### 4.1 RISK BOUND OF ASGD IN THE HIGH-DIMENSIONAL SETTING

Before we present the results, we first introduce three quantities which are cutoffs of the spectrum of $\mathbf{H}$. The eigenvalues of $\mathbf{A}_i$ can be either complex or real, which depends on the range of $\lambda_i$. Define

$$\begin{aligned}
k^\ddagger &:= \max\{i : \lambda_i \geq (\sqrt{q - c\delta} + \sqrt{c(q - \delta)})^2/q^2\}, \\
k^\dagger &:= \max\{i : \lambda_i > (\sqrt{q - c\delta} - \sqrt{c(q - \delta)})^2/q^2\}.
\end{aligned} \tag{4.1}$$

It is easy to see that $k^\ddagger \leq k^\dagger$. For any $i \leq k^\ddagger$ and any $i > k^\dagger$, $\mathbf{A}_i$ has real eigenvalues $x_1 \leq x_2$, and for $i$ between $k^\ddagger$ and $k^\dagger$, $\mathbf{A}_i$ has complex eigenvalues $x_1, x_2$ with the same magnitude. We also define $\widehat{k}$ as

$$\widehat{k} := \max\left\{i : \lambda_i \geq (1 - c)/\delta\right\}.$$

**Parameter choice.** We select hyperparameters of ASGD as follows: We first pick a non-negative integer $\widetilde{\kappa}$. We then select parameters $\delta, \gamma, \beta, \alpha$ as follows, based on $\widetilde{\kappa}$:

$$\delta \leq \frac{1}{2\psi \operatorname{tr}(\mathbf{H})}, \quad \gamma \in \left[\delta, \frac{1}{2\psi \sum_{i > \widetilde{\kappa}} \lambda_i}\right], \quad \beta = \frac{\delta}{\psi \widetilde{\kappa} \gamma}, \quad \alpha = \frac{1}{1 + \beta}. \tag{4.2}$$

We can show that with our choice of paramters, we have $k^\ddagger \leq \widehat{k} \leq k^\dagger$ (see Appendix E.1 for details). For convenience, we introduce the following notations for submatrices of $\mathbf{H}$: for any non-negative integers $k_1 \leq k_2$, denote

$$\mathbf{H}_{k_1:k_2} := \sum_{i=k_1+1}^{k_2} \lambda_i \mathbf{v}_i \mathbf{v}_i^\top, \quad \mathbf{H}_{k_1:\infty} := \sum_{i=k_1+1}^{d} \lambda_i \mathbf{v}_i \mathbf{v}_i^\top.$$

Now we present the main result, which gives a finite excess risk bound for ASGD under the specific parameter choice (4.2).

**Theorem 4.1.** Under Assumptions 3.1, 3.2 and 3.3, with the parameter choice in (4.2), if $N(1-c) \geq 2$, the excess risk of tail-averaged iterate from ASGD satisfies:

$$\mathbb{E}[L(\overline{\mathbf{w}}_{s,s+N})] - L(\mathbf{w}^*) \leq 2 \cdot \text{EffectiveVar} + 2 \cdot \text{EffectiveBias}. \tag{4.3}$$

where the effective variance is bounded by

$$\text{EffectiveVar} \leq \sigma^2 r \left[\frac{27k^*}{2N} + 18(s+N)\gamma^2 \sum_{i > k^*} \lambda_i^2\right] + \frac{\psi r}{N} \left[\frac{9k^*}{N} + 36N\gamma^2 \sum_{i > k^*} \lambda_i^2\right] \cdot \left[\frac{14}{\delta}\|\mathbf{w}_0 - \mathbf{w}^*\|_{\mathbf{I}_{0:\widehat{k}}}^2\right.$$

$$\left. + \frac{10}{1-c}\|\mathbf{w}_0 - \mathbf{w}^*\|_{\mathbf{H}_{\widehat{k}:k^\dagger}}^2 + \frac{2}{\gamma + \delta}\|\mathbf{w}_0 - \mathbf{w}^*\|_{\mathbf{I}_{k^\dagger:k^*}}^2 + 4(s+N)\|\mathbf{w}_0 - \mathbf{w}^*\|_{\mathbf{H}_{k^*:\infty}}^2\right],$$

and the effective bias is bounded by

$$\text{EffectiveBias} \leq \frac{8(c\delta/q)^{2s}}{N^2\delta^2}\|\mathbf{w}_0 - \mathbf{w}^*\|^2_{\mathbf{H}^{-1}_{0:k^{\ddagger}}} + \frac{4s^2}{N^2}c^s\|(\mathbf{I} - \delta\mathbf{H})^{s/2}(\mathbf{w}_0 - \mathbf{w}^*)\|^2_{\mathbf{H}_{k^{\ddagger}:k^{\dagger}}}$$

$$+ \frac{16c^s}{N^2\delta^2}\|(\mathbf{I} - \delta\mathbf{H})^{s/2}(\mathbf{w}_0 - \mathbf{w}^*)\|^2_{\mathbf{H}^{-1}_{k^{\ddagger}:\widehat{k}}} + \frac{100c^s}{N^2(1-c)^2}\|(\mathbf{I} - \delta\mathbf{H})^{s/2}(\mathbf{w}_0 - \mathbf{w}^*)\|^2_{\mathbf{H}_{\widehat{k}:k^{\dagger}}}$$

$$+ \frac{18}{N^2(\gamma+\delta)^2}\left\|\left(\mathbf{I} - \frac{\gamma+\delta}{2}\mathbf{H}\right)^s(\mathbf{w}_0 - \mathbf{w}^*)\right\|^2_{\mathbf{H}^{-1}_{k^{\dagger}:k^*}} + 18\left\|\left(\mathbf{I} - \frac{\gamma+\delta}{2}\mathbf{H}\right)^s(\mathbf{w}_0 - \mathbf{w}^*)\right\|^2_{\mathbf{H}_{k^*:\infty}},$$

with $k^* = \max\{k : \lambda_k \geq 1/((\gamma+\delta)N)\}$, and

$$r := \frac{1}{1 - \psi l}, \quad l := \frac{\delta\,\text{tr}(\mathbf{H})}{2} + \frac{1}{2\psi} + \frac{\gamma}{4}\sum_{i > \widetilde{\kappa}}\lambda_i.$$

Theorem 4.1 establishes the excess risk bound of ASGD under the overparameterized setting. To our knowledge, this is the first instance-dependent bound of ASGD within each eigen-subspace of $\mathbf{H}$. Our excess bound includes both the variance term, which depends on the randomness coming from the data distribution $\mathcal{D}$, and the bias term, which includes "accelerated convergence" terms brought by the ASGD.

**Remark 4.2.** The cutoff index $k^*$ is referred to as the *effective dimension*, which can be much smaller than the model dimensionality $d$, especially when the eigenvalues decay fast. We want to emphasize that similar effective dimension has also appeared in the previous work which analyzes the convergence of SGD under the overparameterized model setting (Zou et al., 2021b; Wu et al., 2022). Nevertheless, the effective dimension of SGD is $k^*_{\text{SGD}} := \max\{k : \lambda_k \geq 1/(\delta N)\}$, which is smaller than that in ASGD. In Section 5, we will provide a comparison of the risk bounds between SGD and ASGD.

**Remark 4.3.** It is worth noting that under the parameter selection (4.2), one can verify that $\psi l < 1$. Such a condition guarantees that $r = 1/(1 - \psi l)$ is finite, which further guarantees that our derived risk bound for effective variance is valid.

### 4.2 Implication in the Classical Setting

In this subsection, we show that Theorem 4.1 implies the excess risk bound in the strongly convex setting and can recover a similar result as Jain et al. (2018). The hyperparameters of ASGD are chosen to be

$$\delta = \frac{1}{2\psi\,\text{tr}(\mathbf{H})}, \quad \gamma = \sqrt{\frac{2\delta}{\psi\mu d}}, \quad \beta = \sqrt{\frac{\mu\delta}{2\psi d}}, \quad \alpha = \frac{1}{1+\beta}, \tag{4.4}$$

where $\mu := \lambda_d$ is the smallest eigenvalue of $\mathbf{H}$. We remark that the parameter choice in (4.4) is different from the choice under the overparameterized setting given in (4.2) because $\widetilde{\kappa}$ is chosen as the model dimension $d$, and the upper bound of $\gamma$ in (4.2), which is $1/(2\psi\sum_{i>\widetilde{\kappa}}\lambda_i)$, becomes vacuous. Instead, we require $\gamma = 2\beta/\mu$ to guarantee that no eigenvalue falls in the region of small eigenvalues such that $\mathbf{A}_i$ has real eigenvalues (i.e., when $i > k^{\dagger}$, see Section I for detailed proof). The following corollary provides the excess risk bound in the strongly convex setting:

**Corollary 4.4.** Under Assumptions 3.1, 3.2 and 3.3, and with the parameter choice in (4.4), the excess risk of tail-averaged iterate from ASGD in the classical regime satisfies:

$$\mathbb{E}[L(\overline{\mathbf{w}}_{s:s+N})] - L(\mathbf{w}^*) \leq \underbrace{\frac{100}{N^2\beta^2}\exp\left(-\frac{\beta s}{2}\right)[L(\mathbf{w}_0) - L(\mathbf{w}^*)]}_{\text{Effective Bias}}$$

$$+ \underbrace{\frac{1008\psi d}{N^2\beta}[L(\mathbf{w}_0) - L(\mathbf{w}^*)] + \frac{36\sigma^2 d}{N} + \frac{128\sigma^2 d}{N^2\beta}}_{\text{Effective Variance}}.$$

Denote $\kappa := \text{tr}(\mathbf{H})/\mu$, then $\beta = \Theta(1/\sqrt{\kappa\widetilde{\kappa}})$. Assuming that $L(\mathbf{w}_0) - L(\mathbf{w}^*) = \mathcal{O}(\sigma^2)$, then the bound given in Corollary 4.4 fully recovers the excess risk upper bound given in Theorem 1 of Jain et al. (2018) in terms of exponential decay rate, leading-order variance and lower-order variance. Moreover, the coefficient of effective bias is $\mathcal{O}(\kappa\widetilde{\kappa}/N^2)$, which significantly improves upon

$\mathcal{O}(\kappa^{13/4}\widetilde{\kappa}^{9/4}d/N^2)$ given in Jain et al. (2018). It is worth noting that Liu & Belkin (2018) proved $\mathcal{O}(1)$ coefficient for effective bias of ASGD. Our result can also recover the constant coefficient when $N(1-c) \geq 2$, because $1-c = 2\alpha\beta \leq 2\beta$ and $1/(N^2\beta^2) \leq 1$. The difference in this coefficient between the bound in Liu & Belkin (2018) and ours is mainly due to slightly different treatments of terms in the form of $N^{-1}\sum_{i=0}^{N-1}(1-\beta)^i$, which is not essential.

## 5 COMPARISON BETWEEN ASGD AND SGD

In this section, we first introduce the SGD update, which is given by

$$\mathbf{w}_t^{\mathrm{SGD}} = \mathbf{w}_{t-1}^{\mathrm{SGD}} - \delta\widehat{\nabla}L(\mathbf{w}_{t-1}^{\mathrm{SGD}}),$$

where $\delta$ satisfies the requirement in (4.2). Analogous to ASGD, tail-averaged SGD is defined as $\overline{\mathbf{w}}_{s:s+N}^{\mathrm{SGD}} := N^{-1}\sum_{t=s}^{s+N-1}\mathbf{w}_t^{\mathrm{SGD}}$. The excess risk of tail-averaged SGD is then $\mathbb{E}[L(\overline{\mathbf{w}}_{s:s+N}^{\mathrm{SGD}})] - L(\mathbf{w}^*)$. We then present the following theorem, which shows the existence of linear regression instances where ASGD outperforms SGD (the proof is given in Appendix D.2):

**Theorem 5.1** (Informal). There exists a class of linear regression instances and corresponding choice of parameter such that the excess risk bound of tail-averaged ASGD satisfies

$$\mathbb{E}[L(\overline{\mathbf{w}}_{s:s+N})] - L(\mathbf{w}^*) = \mathcal{O}(\sigma^2(N^{-1/2} + N^{-2} \cdot 0.9873^s)),$$

and the excess risk bound of tail-averaged SGD satisfies

$$\mathbb{E}[L(\overline{\mathbf{w}}_{s:s+N}^{\mathrm{SGD}})] - L(\mathbf{w}^*) = \Omega(\sigma^2(N^{-1/2} + N^{-2} \cdot 0.996^s)).$$

Theorem 5.1 is inspired by the following comparison of the effective variance and bias of SGD and ASGD with the assumption that $s = \mathcal{O}(N)$. This is a technical assumption that helps to simplify excess risk bounds, and the comparison can be extended to the case of $s = \Omega(N)$. Under the same set of assumptions as Theorem 4.1, Zou et al. (2021b) prove that, with a bias-variance decomposition similar to (4.3), effective variance and effective bias of SGD satisfy:

$$\mathrm{EffectiveVar} \leq \sigma^2 r_{\mathrm{SGD}} \cdot \left[\frac{k_{\mathrm{SGD}}^*}{N} + (s+N)\delta^2 \sum_{i>k_{\mathrm{SGD}}^*} \lambda_i^2\right]$$

$$+ \frac{4\psi r_{\mathrm{SGD}}}{N} \cdot \left[\frac{1}{\delta}\|\mathbf{w}_0 - \mathbf{w}^*\|_{\mathbf{I}_{0:k_{\mathrm{SGD}}^*}}^2 + (s+N)\|\mathbf{w}_0 - \mathbf{w}^*\|_{\mathbf{H}_{k_{\mathrm{SGD}}^*:\infty}}^2\right] \cdot \left[\frac{k_{\mathrm{SGD}}^*}{N} + N\delta^2 \sum_{i>k_{\mathrm{SGD}}^*} \lambda_i^2\right],$$

$$\mathrm{EffectiveBias} \leq \frac{1}{\delta^2 N^2}\|(\mathbf{I} - \delta\mathbf{H})^s(\mathbf{w}_0 - \mathbf{w}^*)\|_{\mathbf{H}_{0:k_{\mathrm{SGD}}^*}^{-1}}^2 + \|(\mathbf{I} - \delta\mathbf{H})^s(\mathbf{w}_0 - \mathbf{w}^*)\|_{\mathbf{H}_{k_{\mathrm{SGD}}^*:\infty}}^2,$$

where $r_{\mathrm{SGD}} = (1 - \psi\delta\operatorname{tr}(\mathbf{H}))^{-1}$ and $k_{\mathrm{SGD}}^* = \max\{i : \lambda_i \geq 1/(\delta N)\}$.
**Comparison of effective variance.** Assuming that the initial variance $\mathbf{w}_0 - \mathbf{w}^*$ is bounded, the effective variance of ASGD is dominated by

$$\sigma^2 r\left[\frac{24k^*}{N} + 18(s+N)\gamma^2 \sum_{i>k^*} \lambda_i^2\right],$$

and effective variance of SGD is dominated by

$$\sigma^2 r_{\mathrm{SGD}}\left[\frac{k_{\mathrm{SGD}}^*}{N} + (s+N)\delta^2 \sum_{i>k_{\mathrm{SGD}}^*} \lambda_i^2\right].$$

Thus, ignoring $\sigma^2$, $r$ and $r_{\mathrm{SGD}}$ and constants, effective variance of ASGD in the subspace of $\lambda_i$ is $\mathcal{O}(\min\{1/N, N\gamma^2\lambda_i^2\})$, compared to $\mathcal{O}(\min\{1/N, N\delta^2\lambda_i^2\})$ for SGD. With $\gamma \geq \delta$ according to the choice of parameters in (4.2), we conclude that the excess variance of ASGD in every subspace is larger than that of SGD.
The following corollary characterizes the effective variance of ASGD when the eigenvalue spectrum decays with a polynomial or exponential rate. These examples have been studied for SGD in Zou et al. (2021b) and Wu et al. (2022).

**Corollary 5.2.** Under the same assumptions as Theorem 4.1, suppose that $\|\mathbf{w}_0 - \mathbf{w}^*\|_2$ is bounded.

1. If the spectrum is $\lambda_i = i^{-(1+r)}$ for some $r > 0$, then the effective variance is $\mathcal{O}((\widetilde{\kappa}/N)^{r/(1+r)})$.

2. If the spectrum is $\lambda_i = e^{-i}$, then the effective variance is $\mathcal{O}((\widetilde{\kappa} + \log N)/N)$.

**Remark 5.3.** For SGD, the effective variance is $\mathcal{O}((1/N)^{r/(1+r)})$ if the eigenvalue spectrum is $\lambda_i = i^{-(1+r)}$, and $\mathcal{O}(\log N/N)$ if the eigenvalue spectrum is $\lambda_i = e^{-i}$ (Zou et al., 2021b). Therefore, the effective variance of ASGD is larger than that of SGD under both eigenvalue spectra.

**Comparison of effective bias.** Effective bias of both SGD and ASGD decay exponentially in $s$ within each subspace. The decay rate of SGD is $(1 - \delta\lambda_i)^s$ in the subspace of $\lambda_i$. For ASGD,

1. When $i \le k^{\ddagger}$, the decay rate in the subspace of $\lambda_i$ is $(c\delta/q)^s$. By definition of $k^{\ddagger}$, we have $1 - \delta\lambda_i \le c\delta/q$ (see Appendix E.1 for detailed proof).

2. When $k^{\ddagger} < i \le k^{\dagger}$, the decay rate in the subspace of $\lambda_i$ is $[c(1 - \delta\lambda_i)]^{s/2}$. According to the definition of $\widehat{k}$, when $k^{\ddagger} < i \le \widehat{k}$, we have $1 - \delta\lambda_i \le \sqrt{c(1 - \delta\lambda_i)}$; When $\widehat{k} < i \le k^{\dagger}$, we have $1 - \delta\lambda_i \ge \sqrt{c(1 - \delta\lambda_i)}$.

3. When $i > k^{\dagger}$, the decay rate in the subspace of $\lambda_i$ is $(1 - (\gamma + \delta)\lambda_i/2)^s$. By the choice of parameters (4.2), we have $\gamma \ge \delta$, so $1 - (\gamma + \delta)\lambda_i/2 \le 1 - \delta\lambda_i$.

Combining the three cases above, we conclude that the effective bias of ASGD decays faster than that of SGD in eigen-subspaces of $\lambda_i$ where $i > \widehat{k}$, while it decays slower than SGD in subspaces of $\lambda_i$ where $i \le \widehat{k}$. This phenomenon is illustrated in Figure 1. Therefore, ASGD can perform better than SGD if $\mathbf{w}_0 - \mathbf{w}^*$ is mostly refined to the eigen-subspaces of $\lambda_i$ where $i > \widehat{k}$.

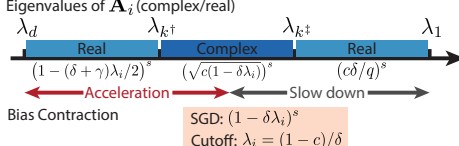

**Figure 1:** Illustration of the eigenspectrum.

We remark that this result is consistent with the acceleration of bias decay presented in Jain et al. (2018). Without instance-specific analysis, the exponential decay rate of bias is determined by the decay rate in subspace of the smallest eigenvalue. As the effective bias of ASGD decays faster than that of SGD in the eigen-subspace of small eigenvalues, the worst-case decay rate of the bias error of ASGD enjoys acceleration compared to SGD.

## 6 EXPERIMENTS

In this section, we empirically verify that ASGD can outperform SGD when $\mathbf{w}_0 - \mathbf{w}^*$ is mainly confined to the eigen-subspace of small eigenvalues.

**Data model.** Our experiments are based on the setting of overparameterized linear regression, where the model dimenstion is $d = 2000$. The data covariance matrix $\mathbf{H}$ is diagonal with eigenvalues $\lambda_i = i^{-2}$. The input $\mathbf{x}_t$ follows Gaussian distribution $\mathcal{N}(\mathbf{0}, \mathbf{H})$, so Assumption 3.2 holds with $\psi = 3$. The ground truth weight vector is $\mathbf{w}^* = \mathbf{0}$, and the label $y_t$ follows the distribution $\mathcal{N}(0, \sigma^2)$ where $\sigma^2 = 0.01$.

**Hyperparameters of ASGD and SGD.** We select parameters of ASGD so that it satisfies the requirements in (4.2). We first let $\widetilde{\kappa} = 5$. According to (4.2), $\delta$ satisfies $\delta \le 1/\pi^2$, so we pick $\delta = 0.1$, which is also the stepsize of SGD. We then let $\alpha = 0.9875$, so that $(1 - c)/\delta = 2(1 - \alpha)/\delta = 0.25 = \lambda_2$, which implies that $\widehat{k} = 2$. Finally, we select $\beta = (1 - \alpha)/\alpha$ and $\gamma = \delta/(\psi\widetilde{\kappa}\beta)$. We can verify that the parameters satisfy all requirements in (4.2).

We fix the length of tail averaging as $N = 500$, and conduct experiments on different $s$ where $s = 50, 100, 150, \ldots, 500$. In each experiment, we measure $\overline{\mathbf{w}}_{s:s+N}^{\top} \mathbf{H} \overline{\mathbf{w}}_{s:s+N}$. For each $s$, we run the experiment 10 times and take the average of the test results.

We examine three different initializations: (a) $\mathbf{w}_0 = 10 \cdot \mathbf{e}_1$, representing the case where $\mathbf{w}_0 - \mathbf{w}^*$ is mainly refined to the subspace of large eigenvalues, (b) $\mathbf{w}_0 = 10 \cdot \mathbf{e}_2$, representing the case where $\mathbf{w}_0 - \mathbf{w}^*$ is mainly refined to the subspace of $\lambda_{\widehat{k}}$, and (c) $\mathbf{w}_0 = 10 \cdot \mathbf{e}_{20}$, representing the case where $\mathbf{w}_0 - \mathbf{w}^*$ is mainly refined to the subspace of small eigenvalues. Experiment results are shown in Figure 2. We observe that ASGD indeed outperforms SGD in the scenario where $\mathbf{w}_0 - \mathbf{w}^*$ is mostly refined to the subspace of small eigenvalues, and performs worse than SGD when $\mathbf{w}_0 - \mathbf{w}^*$ is refined to the subspace of large eigenvalues. Additionally, the excess risks of SGD and ASGD are similar when $\mathbf{w}_0 - \mathbf{w}^*$ aligns with the subspace corresponding to $\lambda_{\widehat{k}}$, which is also aligns with the implication of Theorem 4.1.

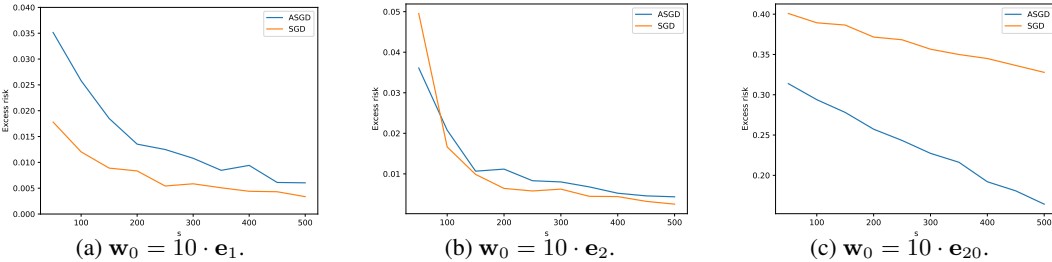

(a) $\mathbf{w}_0 = \overset{s}{10} \cdot \mathbf{e}_1$.      (b) $\mathbf{w}_0 = \overset{s}{10} \cdot \mathbf{e}_2$.      (c) $\mathbf{w}_0 = \overset{s}{10} \cdot \mathbf{e}_{20}$.

**Figure 2:** Comparison of excess risk of ASGD and SGD. The noise scale is $\sigma^2 = 0.01$. We run each experiment 10 times and take the average of the excess risk in the 10 trials.

## 7 PROOF SKETCH

In this section, we present the high-level ideas in our proof. We mainly introduce two main ideas of the proof, including (i) bias-variance decomposition, and (ii) analysis of excess risk bounds within each eigen-subspace, based on the eigenvalues of $\mathbf{A}_i$.

Define the tail averaged centered ASGD iterate as $\overline{\boldsymbol{\eta}}_{s,s+N} := N^{-1} \sum_{t=s}^{s+N-1} \boldsymbol{\eta}_t$. The excess risk is then

$$\mathbb{E}[L(\overline{\mathbf{w}}_{s,s+N})] - L(\mathbf{w}^*) = \frac{1}{2} \left\langle \begin{bmatrix} \mathbf{H} & \mathbf{0} \\ \mathbf{0} & \mathbf{0} \end{bmatrix}, \mathbb{E}[\overline{\boldsymbol{\eta}}_{s,s+N} \otimes \overline{\boldsymbol{\eta}}_{s,s+N}] \right\rangle.$$

We also define the linear operators $\mathcal{B} := \mathbb{E}[\widehat{\mathbf{A}}_t \otimes \widehat{\mathbf{A}}_t]$ and $\widetilde{\mathcal{B}} := \mathbf{A} \otimes \mathbf{A}$, which are both PSD operators. Additionally, the difference $\mathcal{B} - \widetilde{\mathcal{B}}$ is also a PSD operator, which contributes to the effect of the fourth moment in the excess risk bound. The reader can refer to Appendix F for details of the linear operators.

### 7.1 BIAS-VARIANCE DECOMPOSITION

Following the techique used extensively in previous works (Dieuleveut & Bach, 2015; Jain et al., 2018; Zou et al., 2021b; Wu et al., 2022; Liang & Rakhlin, 2020), we decompose the centered iterate $\boldsymbol{\eta}_t$ into the bias sequence $\boldsymbol{\eta}_t^{\text{bias}}$ and the variance sequence $\boldsymbol{\eta}_t^{\text{var}}$, defined recursively as

$$\boldsymbol{\eta}_t^{\text{bias}} = \widehat{\mathbf{A}}_t \boldsymbol{\eta}_{t-1}^{\text{bias}}, \quad \boldsymbol{\eta}_0^{\text{bias}} = \boldsymbol{\eta}_0; \tag{7.1}$$

$$\boldsymbol{\eta}_t^{\text{var}} = \widehat{\mathbf{A}}_t \boldsymbol{\eta}_{t-1}^{\text{var}} + \boldsymbol{\zeta}_t, \quad \boldsymbol{\eta}_0^{\text{var}} = \mathbf{0}. \tag{7.2}$$

The tail averaged iterate is then $\overline{\boldsymbol{\eta}}_{s:s+N} = \overline{\boldsymbol{\eta}}_{s:s+N}^{\text{bias}} + \overline{\boldsymbol{\eta}}_{s:s+N}^{\text{var}}$, where

$$\overline{\boldsymbol{\eta}}_{s:s+N}^{\text{bias}} := \frac{1}{N} \sum_{t=s}^{s+N-1} \boldsymbol{\eta}_t^{\text{bias}}, \quad \overline{\boldsymbol{\eta}}_{s:s+N}^{\text{var}} := \frac{1}{N} \sum_{t=s}^{s+N-1} \boldsymbol{\eta}_t^{\text{var}}. \tag{7.3}$$

The excess risk can be decomposed into bias and variance:

$$\mathbb{E}[L(\overline{\mathbf{w}}_{s:s+N})] - L(\mathbf{w}^*) = \frac{1}{2} \left\langle \begin{bmatrix} \mathbf{H} & \mathbf{0} \\ \mathbf{0} & \mathbf{0} \end{bmatrix}, \mathbb{E}[\overline{\boldsymbol{\eta}}_{s:s+N} \otimes \overline{\boldsymbol{\eta}}_{s:s+N}] \right\rangle \le 2 \cdot \text{Bias} + 2 \cdot \text{Variance},$$

where

$$\text{Bias} := \frac{1}{2} \left\langle \begin{bmatrix} \mathbf{H} & \mathbf{0} \\ \mathbf{0} & \mathbf{0} \end{bmatrix}, \mathbb{E}[\overline{\boldsymbol{\eta}}_{s:s+N}^{\text{bias}} \otimes \overline{\boldsymbol{\eta}}_{s:s+N}^{\text{bias}}] \right\rangle, \quad \text{Variance} := \frac{1}{2} \left\langle \begin{bmatrix} \mathbf{H} & \mathbf{0} \\ \mathbf{0} & \mathbf{0} \end{bmatrix}, \mathbb{E}[\overline{\boldsymbol{\eta}}_{s:s+N}^{\text{var}} \otimes \overline{\boldsymbol{\eta}}_{s:s+N}^{\text{var}}] \right\rangle.$$

Define the covariance matrices $\mathbf{B}_t := \mathbb{E}[\boldsymbol{\eta}_t^{\text{bias}} \otimes \boldsymbol{\eta}_t^{\text{bias}}]$ and $\mathbf{C}_t := \mathbb{E}[\boldsymbol{\eta}_t^{\text{var}} \otimes \boldsymbol{\eta}_t^{\text{var}}]$. The recursive forms of $\mathbf{B}_t$ and $\mathbf{C}_t$ then satisfy

$$\mathbf{B}_t = \mathcal{B} \circ \mathbf{B}_{t-1}, \quad \mathbf{B}_0 = \boldsymbol{\eta}_0 \otimes \boldsymbol{\eta}_0; \tag{7.4}$$

$$\mathbf{C}_t = \mathcal{B} \circ \mathbf{C}_{t-1} + \widehat{\boldsymbol{\Sigma}}, \quad \mathbf{C}_0 = \mathbf{0}. \tag{7.5}$$

### 7.2 PROOF OF THE BIAS BOUND

In this part, we provide an overview of the analysis of the bias bound in a simplified problem setting. We consider the last bias iterate (i.e., $N = 1$) and assume that $\mathcal{B} = \widetilde{\mathcal{B}}$. The analysis of the general

cases is given in Appendix H. According to the recursive form of $\mathbf{B}_s$ in (7.4), we have $\mathbf{B}_s = \mathcal{B}^s \circ \mathbf{B}_0$. With the assumptions that $\mathcal{B} = \widetilde{\mathcal{B}}$, we have

$$\mathbf{B}_s = \widetilde{\mathcal{B}}^s \circ \mathbf{B}_0 = \mathbf{A}^s \left( \begin{bmatrix} 1 & 1 \\ 1 & 1 \end{bmatrix} \otimes (\mathbf{w}_0 - \mathbf{w}^*)(\mathbf{w}_0 - \mathbf{w}^*)^\top \right) (\mathbf{A}^s)^\top.$$

Note that $\mathbf{A}$ is block-diagonal with each block being $\mathbf{A}_i$, so bias can be expressed as

$$\text{Bias} = \frac{1}{2} \left\langle \begin{bmatrix} \mathbf{H} & \mathbf{0} \\ \mathbf{0} & \mathbf{0} \end{bmatrix}, \mathbb{E}[\overline{\boldsymbol{\eta}}_{s:s+N}^{\text{bias}} \otimes \overline{\boldsymbol{\eta}}_{s:s+N}^{\text{bias}}] \right\rangle = \frac{1}{2} \left\langle \begin{bmatrix} \mathbf{H} & \mathbf{0} \\ \mathbf{0} & \mathbf{0} \end{bmatrix}, \mathbf{B}_s \right\rangle = \frac{1}{2} \sum_{i=1}^d \lambda_i w_i^2 \left( \mathbf{A}_i^s \begin{bmatrix} 1 \\ 1 \end{bmatrix} \right)_1^2,$$

where $w_i := (\mathbf{w}_0 - \mathbf{w}^*)_i$. The following lemma explicitly characterizes $\mathbf{A}_i^k$:

**Lemma 7.1.** Let the eigenvalues of $\mathbf{A}_i$ be $x_1$ and $x_2$. Then, for any integer $k \geq 1$, we have

$$\mathbf{A}_i^k = \begin{bmatrix} -c(1 - \delta\lambda_i) \cdot \frac{x_2^{k-1} - x_1^{k-1}}{x_2 - x_1} & (1 - \delta\lambda_i) \cdot \frac{x_2^k - x_1^k}{x_2 - x_1} \\ -c \cdot \frac{x_2^k - x_1^k}{x_2 - x_1} & \frac{x_2^{k+1} - x_1^{k+1}}{x_2 - x_1} \end{bmatrix}.$$

The detailed proof of Lemma 7.1 is given as the proof of Lemma E.3. With Lemma 7.1, we have

$$\mathrm{I} := \left( \mathbf{A}_i^s \begin{bmatrix} 1 \\ 1 \end{bmatrix} \right)_1 = (1 - \delta\lambda_i) \frac{x_2^{s-1}(x_2 - c) - x_1^{s-1}(x_1 - c)}{x_2 - x_1}.$$

For $i \leq k^\ddagger$ and $i > k^\dagger$, i.e., $\mathbf{A}_i$ has real eigenvalues $x_1 < x_2$, I decays exponentially with the same rate of $x_2^s$. For $k^\ddagger < i \leq k^\dagger$, i.e., $\mathbf{A}_i$ has complex eigenvalues with $|x_1| = |x_2|$, $|\mathrm{I}|$ is bounded by

$$|\mathrm{I}| = (1 - \delta\lambda_i) \left| \frac{x_2^{s-1}(x_2 - c) - x_1^{s-1}(x_1 - c)}{x_2 - x_1} \right| \leq \left| \frac{x_1^{s-1} + x_2^{s-1}}{2} + \frac{x_1 + x_2 - 2c}{2} \cdot \frac{x_2^{s-1} - x_1^{s-1}}{x_2 - x_1} \right|$$

$$\leq |x_2|^{s-1} + \frac{|x_1 + x_2 - 2c|}{2} \cdot \left| \frac{x_2^{s-1} - x_1^{s-1}}{x_2 - x_1} \right|,$$

where the first inequality holds because $0 \leq 1 - \delta\lambda_i \leq 1$, and the second inequality holds due to triangle inequality. For the term $|(x_2^{s-1} - x_1^{s-1})/(x_2 - x_1)|$, note that

$$\left| \frac{x_2^{s-1} - x_1^{s-1}}{x_2 - x_1} \right| = \left| \sum_{k=0}^{s-2} x_2^k x_1^{s-2-k} \right| \leq \sum_{k=0}^{s-2} |x_2^k| \cdot |x_2^{s-2-k}| = \sum_{k=0}^{s-2} |x_2|^k \cdot |x_1|^{s-2-k} = (s-1)|x_2|^{s-2},$$

where the inequality holds due to triangle inequality, and the second inequality holds because $|x_1| = |x_2|$. Therefore, the exponential decay rate of $|\mathrm{I}|$ is $|x_2|^s$. The following lemma provides tight bounds of $x_2$, thus characterizing the exponential rate of bias decay within each eigen-subspace:

**Lemma 7.2.** Let $x_1, x_2$ be the eigenvalues of $\mathbf{A}_i$. Then

(a) When $i \leq k^\ddagger$, $(c\delta - \sqrt{c(q - \delta)(q - c\delta)})/q \leq x_2 \leq c\delta/q$.

(b) When $k^\ddagger < i \leq k^\dagger$, $|x_2| = \sqrt{c(1 - \delta\lambda_i)}$.

(c) When $i > k^\dagger$, $1 - (\gamma + \delta)\lambda_i \leq x_2 \leq 1 - (\gamma + \delta)\lambda_i/2$.

The detailed proof of Lemma 7.2 is given in Appendix E.1. We can thus obtain the exponential decay rate of the effective bias.

# 8 CONCLUSION

In this work, we consider accelerated SGD with tail averaging for overparameterized linear regression. We provide instance-dependent risk bounds for accelerated SGD that are comprehensively dependent on the spectrum of the data covariance matrix. We show that the variance error of accelerated SGD is always larger than that of SGD. We also show that the bias error of accelerated SGD is smaller than that of SGD along the small eigenvalues subspace but is larger than that of SGD along the small eigenvalues subspace. These together suggest that accelerated SGD outperforms SGD only if the signals mostly align with the small eigenvalues subspaces of the data covariance and that the noise is small. Our results also improve a best-known bound for accelerated SGD in the classic regime (Jain et al., 2018).

ACKNOWLEDGEMENTS

We thank the anonymous reviewers and area chair for their helpful comments. XL, YD and QG are supported in part by the NSF grants IIS-2008981, CHE-2247426, and the Sloan Research Fellowship. The views and conclusions contained in this paper are those of the authors and should not be interpreted as representing any funding agencies.

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

## A  ADDITIONAL EXPERIMENTS

In this section, we provide additional experiments that justify the theoretical results provided in Theorem 4.1.

**Data model.** Similar to the experiments provided in Section 6, the model dimension is set to be $d = 2000$, and the input $\mathbf{x}_t$ follows Gaussian distribution $\mathcal{N}(\mathbf{0}, \mathbf{H})$. We consider $\mathbf{H}$ with three types of spectrum: (i) $\lambda_k = k^{-2}$, (ii) $\lambda_k = k \log(k+1)$, and (iii) $\lambda_k = e^{-k/2}$. The ground truth weight vector is $\mathbf{w}^* = 0$, and the label $y_t$ follows the distribution $y_t \sim \mathcal{N}(0, \sigma^2)$ where $\sigma = 0.2$.

**Hyperparameters.** We select the same hyperparameters of ASGD and SGD as the choice in Section 6, i.e., $\psi = 3, \widetilde{\kappa} = 5, \delta = 0.1, \alpha = 0.9875, \beta = (1-\alpha)/\alpha$ and $\gamma = \delta/(\psi\widetilde{\kappa}\beta)$. We fix $N = 500$ and conduct experiments on different $s$ where $s = 50, 100, \ldots, 500$.

In each experiment, we measure both the bias error $(\overline{\mathbf{w}}_{s:s+N}^{\mathrm{bias}})^\top \mathbf{H}\overline{\mathbf{w}}_{s:s+N}^{\mathrm{bias}}$ and the variance error $(\overline{\mathbf{w}}_{s:s+N}^{\mathrm{var}})^\top \mathbf{H}\overline{\mathbf{w}}_{s:s+N}^{\mathrm{var}}$. For each $s$, we run the experiment 10 times, and take the average of the test results. We examine two initializations: (a) $\mathbf{w}_0 = 10 \cdot \mathbf{e}_1$, which is the case where $\mathbf{w}_0 - \mathbf{w}^*$ is mainly refined to the subspace of large eigenvalues, and (b) $\mathbf{w}_0 = 10 \cdot \mathbf{e}_{10}$, which is the case where $\mathbf{w}_0 - \mathbf{w}^*$ is mainly refined to the subspace of small eigenvalues.

The experimental results are shown in Figures 3, 4 and 5. In all experiments, the variance error of ASGD is larger than that of SGD. However, the bias error of ASGD decays faster than that of SGD when $\mathbf{w}_0 - \mathbf{w}^*$ is mainly refined to the subspace of small eigenvalues.

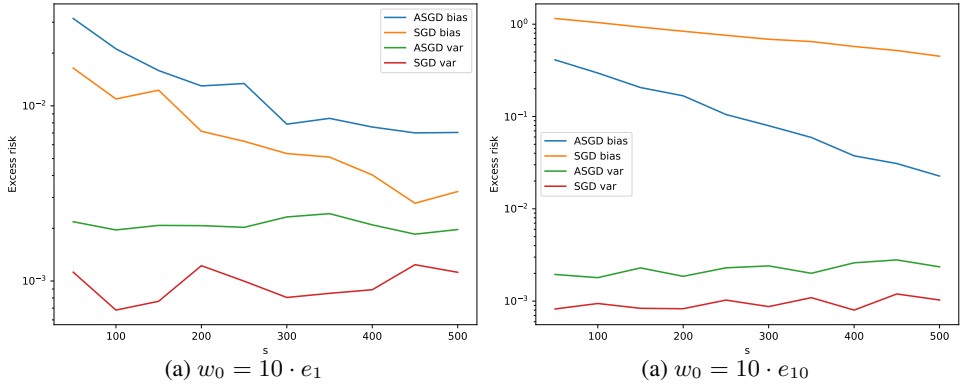

(a) $w_0 = 10 \cdot e_1$      (a) $w_0 = 10 \cdot e_{10}$

**Figure 3:** Comparison of bias error and variance error of ASGD and SGD. The spectrum of $\mathbf{H}$ is $\lambda_k = k^{-2}$.

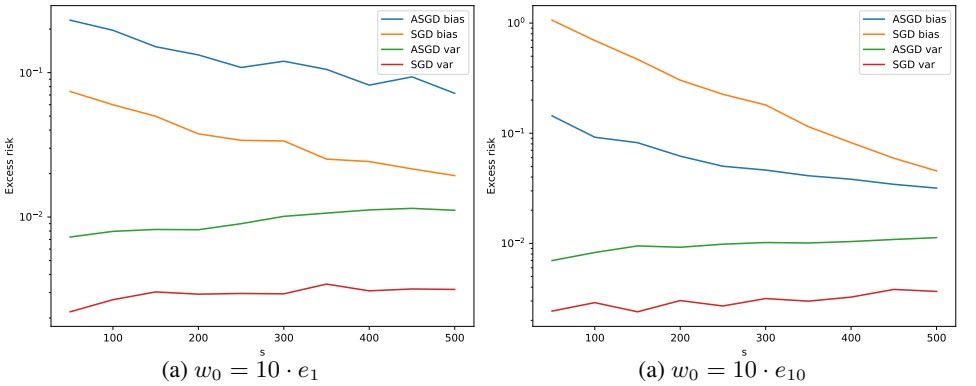

(a) $w_0 = 10 \cdot e_1$      (a) $w_0 = 10 \cdot e_{10}$

**Figure 4:** Comparison of bias error and variance error of ASGD and SGD. The spectrum of $\mathbf{H}$ is $\lambda_k = k \log(k+1)$.

## B  PARAMETER CHOICE

### B.1  DERIVATION OF PARAMETER CHOICE

Following the optimization literature (Nesterov, 1983), we first fix the relationship between $\alpha$ and $\beta$ as

$$\alpha = \frac{1}{1+\beta}. \tag{B.1}$$

We then fix

$$\delta = \psi\widetilde{\kappa}\beta\gamma, \tag{B.2}$$

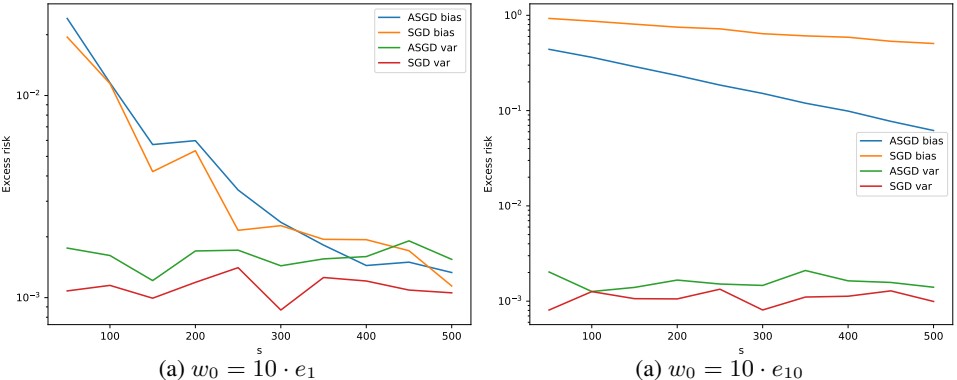

**Figure 5:** Comparison of bias error and variance error of ASGD and SGD. The specturm of $\mathbf{H}$ is $\lambda_k = e^{-k/2}$.

following Jain et al. (2018). We remark that introducing $\widetilde{\kappa}$ prevents the effect of fourth moment from blowing up (see proof of Lemma F.5). Furthermore, we require $\gamma \geq \delta$ to enforce acceleration. Then, from the requirement $\psi l < 1$, we require

$$\frac{\delta \psi \operatorname{tr}(\mathbf{H})}{2} + \frac{1}{2} + \frac{\psi \gamma}{4} \sum_{i > \widetilde{\kappa}} \lambda_i < 1.$$

Therefore, it suffices to take

$$\delta \leq \frac{1}{2\psi \operatorname{tr}(\mathbf{H})}, \quad \gamma \leq \frac{1}{2\psi \sum_{i > \widetilde{\kappa}} \lambda_i}. \tag{B.3}$$

Combining (B.1), (B.2) and (B.3), we derive the choice of parameters in (4.2).
We remark that we get rid of dimension dependency by merit of the term $\psi \gamma / 4 \cdot \sum_{i > \widetilde{\kappa}} \lambda_i$. Without this term, $\widetilde{\kappa}$ should be chosen as the model dimension $d$ (as in Jain et al. (2018)).

### B.2 DISCUSSION OF PARAMETERS

In the parameter choice (4.2), $\widetilde{\kappa}$ is a free parameter. In this section, we discuss how the choice of $\widetilde{\kappa}$ affects the excess risk bound. Suppose that both equalities are attained in (B.3). We focus the impact of $\widetilde{\kappa}$ on (i) eigenvalue cutoff $\widehat{k}$, and (ii) bias decay rate.
Note that

$$\gamma = \frac{1}{2\psi \sum_{i > \widetilde{\kappa}} \lambda_i},$$

so $\gamma$ increases as $\kappa$ increases. Furthermore,

$$\beta = \frac{\delta}{\psi \widetilde{\kappa} \gamma},$$

so $\beta$ decreases as $\widetilde{\kappa}$ increases. We also have

$$c = \alpha(1 - \beta) = \frac{1 - \beta}{1 + \beta},$$

so $c$ increases as $\widetilde{\kappa}$ increases.
$\widehat{k}$ is defined as $\widehat{k} = \max\{k : \lambda_k \geq (1 - c)/\delta\}$, so $\widehat{k}$ increases as $\widetilde{\kappa}$ increases; The bias decay rate in the subspace of the smallest eigenvalues (i.e., $i > k^\dagger$) is $1 - (\gamma + \delta)\lambda_i / 2$, so the decay rate accelerates for larger $\widetilde{\kappa}$. However, for the subspace of $\lambda_i$ where $k^\dagger < i \leq k^\ddagger$, the bias decay rate is $[c(1 - \delta\lambda_i)]^{s/2}$, so the decay rate slows down for larger $\widetilde{\kappa}$.
Combining all the above, we conclude that the choice of $\widetilde{\kappa}$ is subject to the eigenvalue spectrum of the data covariance matrix. Additionally, choosing a small $\widetilde{\kappa}$ will make the algorithm perform more like SGD.

## C IMPLICATION FOR STOCHASTIC HEAVY BALL METHOD

In this section, we extend the results we obtained for ASGD to By taking $\delta = 0$ in (3.3) and eliminating $\mathbf{v}_t$ and $\mathbf{u}_t$ using (3.2) and (3.4), we get

$$\mathbf{w}_{t+1} = \mathbf{w}_t - (1 - \alpha)\gamma \cdot \widehat{\nabla} L(\mathbf{w}_t) + \alpha(1 - \beta) \cdot (\mathbf{w}_t - \mathbf{w}_{t-1}),$$

which is exactly the form of the stochastic heavy ball (SHB) update. Therefore, the excess risk bound we presented in Theorem 4.1 can be directly applied to SHB.

As there are three free parameters but only two combinations $(1 - \alpha)\gamma$ and $\alpha(1 - \beta)$ are used, we enforce that $\beta = (1 - \alpha)/\alpha$ and define $c = \alpha(1 - \beta)$ and $q = (1 - \alpha)\gamma$, similar to ASGD. By (4.1) and the definition of $\widehat{k}$, we have $k^\ddagger = \widehat{k} = 0$. Therefore, the following corollary gives the excess risk bound of SHB:

**Corollary C.1.** Consider stochastic heavy ball (SHB) method, given by the update rule

$$\mathbf{w}_{t+1} = \mathbf{w}_t - q\widehat{\nabla}L(\mathbf{w}_t) + c(\mathbf{w}_t - \mathbf{w}_{t-1}),$$

where the hyperparameters satisfy $c \in (0, 1 - 2/N]$ and $q = (1 - c)\gamma/2$ with

$$\gamma \in \left(0, \frac{4}{\psi \operatorname{tr}(\mathbf{H})}\right).$$

Define $r_{\text{SHB}} := (1 - \psi\gamma \operatorname{tr}(\mathbf{H})/4)^{-1}$, $k^* := \max\{k : \lambda_k \geq 1/(\gamma N)\}$, and define $k^\dagger$ as in (4.1). Then we have the following upper bound for the excess risk:

$$\mathbb{E}[L(\overline{\mathbf{w}}_{s:s+N})] - L(\mathbf{w}^*) \leq 2 \cdot \text{EffectiveVar} + 2 \cdot \text{EffectiveBias},$$

where effective variance is bounded by

$$
\text{EffectiveVar} \leq \sigma^2 r_{\text{SHB}} \left[\frac{27k^*}{2N} + 18(s + N)\gamma^2 \sum_{i>k^*} \lambda_i^2\right] + \frac{\psi r_{\text{SHB}}}{N}\left[\frac{9k^*}{N} + 36N\gamma^2 \sum_{i>k^*} \lambda_i^2\right]
$$

$$
\cdot \left[\frac{10}{1-c}\|\mathbf{w}_0 - \mathbf{w}^*\|_{\mathbf{H}_{0:k^\dagger}}^2 + \frac{2}{\gamma}\|\mathbf{w}_0 - \mathbf{w}^*\|_{\mathbf{I}_{k^\dagger:k^*}}^2 + 4(s + N)\|\mathbf{w}_0 - \mathbf{w}^*\|_{\mathbf{H}_{k^*:\infty}}^2\right],
$$

and effective bias is bounded by

$$
\text{EffectiveBias} \leq c^s \cdot \left(4s^2 + \frac{100}{(1-c)^2}\right) \cdot \frac{\|\mathbf{w}_0 - \mathbf{w}^*\|_{\mathbf{H}_{0:k^\dagger}}^2}{N^2}
$$

$$
+ \frac{18}{N^2\gamma^2}\left\|\left(\mathbf{I} - \frac{\gamma\mathbf{H}}{2}\right)^s (\mathbf{w}_0 - \mathbf{w}^*)\right\|_{\mathbf{H}_{k^\dagger:k^*}^{-1}}^2 + 18\left\|\left(\mathbf{I} - \frac{\gamma\mathbf{H}}{2}\right)^s (\mathbf{w}_0 - \mathbf{w}^*)\right\|_{\mathbf{H}_{k^*:\infty}}^2.
$$

**Remark C.2.** In the eigen-subspace of $\lambda_i$, the exponential decay rate of effective bias of SHB is $\max(c^s, (1 - \gamma\lambda_i)^{2s})$, which is never faster than that of SGD. This happens because for SHB, $\gamma$ has to be smaller than that of ASGD to control the effect of stochastic gradient. We can thus demonstrate that ASGD is superior to SHB in terms of the exponitial decay rate of the bias error, which extends a similar result given by Kidambi et al. (2018) to the instance-dependent case.

## D  PROOF OF MAIN RESULTS

In this section we prove Theorems 4.1 and 5.1.

### D.1  PROOF OF THEOREM 4.1

We start with the basic bias-variance decomposition lemma.

**Lemma D.1** (Bias-variance decomposition, Jain et al. (2018)). The excess risk can be decomposed into bias and variance as

$$
\mathbb{E}[L(\overline{\mathbf{w}}_{s:s+N})] - L(\mathbf{w}^*) = \frac{1}{2}\left\langle \begin{bmatrix} \mathbf{H} & \mathbf{0} \\ \mathbf{0} & \mathbf{0} \end{bmatrix}, \mathbb{E}[\overline{\boldsymbol{\eta}}_{s:s+N} \otimes \overline{\boldsymbol{\eta}}_{s:s+N}]\right\rangle \leq 2 \cdot \text{Bias} + 2 \cdot \text{Variance}, \tag{D.1}
$$

where

$$
\text{Bias} := \frac{1}{2}\left\langle \begin{bmatrix} \mathbf{H} & \mathbf{0} \\ \mathbf{0} & \mathbf{0} \end{bmatrix}, \mathbb{E}[\overline{\boldsymbol{\eta}}_{s:s+N}^{\text{bias}} \otimes \overline{\boldsymbol{\eta}}_{s:s+N}^{\text{bias}}]\right\rangle
$$

$$
\text{Variance} := \frac{1}{2}\left\langle \begin{bmatrix} \mathbf{H} & \mathbf{0} \\ \mathbf{0} & \mathbf{0} \end{bmatrix}, \mathbb{E}[\overline{\boldsymbol{\eta}}_{s:s+N}^{\text{var}} \otimes \overline{\boldsymbol{\eta}}_{s:s+N}^{\text{var}}]\right\rangle.
$$

This indicates that the generalization error could be bounded respectively by analyzing the bias and variance. We then further decompose bias and variance.

**Lemma D.2.** Bias and Variance can be decomposed as

$$\text{Variance} = \frac{1}{2}\left\langle \begin{bmatrix} \mathbf{H} & \mathbf{0} \\ \mathbf{0} & \mathbf{0} \end{bmatrix}, \mathbf{M}_1 + \mathbf{M}_2 \right\rangle, \quad \text{Bias} = \frac{1}{2}\left\langle \begin{bmatrix} \mathbf{H} & \mathbf{0} \\ \mathbf{0} & \mathbf{0} \end{bmatrix}, \mathbf{M}_3 + \mathbf{M}_4 \right\rangle,$$

where

$$\mathbf{M}_1 := \frac{1}{N^2}\left[\sum_{k=0}^{N-1} \mathbf{A}^k\right]\mathbf{C}_s\left[\sum_{k=0}^{N-1} \mathbf{A}^k\right]^\top, \tag{D.2}$$

$$\mathbf{M}_2 := \frac{1}{N^2}\sum_{t=1}^{N-1}\left[\sum_{k=0}^{N-t-1} \mathbf{A}^k\right](\mathbf{C}_{s+t} - \widetilde{\mathcal{B}} \circ \mathbf{C}_{s+t-1})\left[\sum_{k=0}^{N-t-1} \mathbf{A}^k\right]^\top, \tag{D.3}$$

$$\mathbf{M}_3 := \frac{1}{N^2}\left[\sum_{k=0}^{N-1} \mathbf{A}^k\right]\mathbf{B}_s\left[\sum_{k=0}^{N-1} \mathbf{A}^k\right]^\top, \tag{D.4}$$

$$\mathbf{M}_4 := \frac{1}{N^2}\sum_{t=1}^{N-1}\left[\sum_{k=0}^{N-t-1} \mathbf{A}^k\right](\mathbf{B}_{s+t} - \widetilde{\mathcal{B}} \circ \mathbf{B}_{s+t-1})\left[\sum_{k=0}^{N-t-1} \mathbf{A}^k\right]^\top. \tag{D.5}$$

*Proof.* The proof largely follows Zou et al. (2021b). From the definitions of $\eta_t^{\text{bias}}$ as in (7.1), we have the following

$$\mathbb{E}[\eta_t^{\text{bias}}|\eta_{t-1}^{\text{bias}}] = \mathbb{E}[\widehat{\mathbf{A}}_t\eta_{t-1}^{\text{bias}}|\eta_{t-1}^{\text{bias}}] = \mathbf{A}\eta_{t-1}^{\text{bias}}, \tag{D.6}$$

and for $\eta_t^{\text{variance}}$ as in (7.2) we have

$$\mathbb{E}[\eta_t^{\text{var}}|\eta_{t-1}^{\text{var}}] = \mathbb{E}[\widehat{\mathbf{A}}_t\eta_{t-1}^{\text{var}} + \zeta_t|\eta_{t-1}^{\text{var}}] = \mathbf{A}\eta_{t-1}^{\text{var}}. \tag{D.7}$$

Then, regarding the term $\mathbb{E}[\overline{\eta}_{s:s+N}^{\text{var}} \otimes \overline{\eta}_{s:s+N}^{\text{var}}]$, we have

$$\mathbb{E}[\overline{\eta}_{s:s+N}^{\text{var}} \otimes \overline{\eta}_{s:s+N}^{\text{var}}]$$

$$= \frac{1}{N^2}\sum_{t=s}^{s+N-1}\left(\mathbb{E}[\eta_t^{\text{var}} \otimes \eta_t^{\text{var}}] + \sum_{k=t+1}^{s+N-1}\mathbb{E}[\eta_k^{\text{var}} \otimes \eta_t^{\text{var}}] + \sum_{k=t+1}^{s+N-1}\mathbb{E}[\eta_t^{\text{var}} \otimes \eta_k^{\text{var}}]\right)$$

$$= \frac{1}{N^2}\sum_{t=s}^{s+N-1}\left[\mathbf{C}_t + \sum_{k=t+1}^{s+N-1}\mathbf{A}^{k-t}\mathbf{C}_t + \sum_{k=t+1}^{s+N-1}\mathbf{C}_t(\mathbf{A}^{k-t})^\top\right]$$

$$= \frac{1}{N^2}\left[\sum_{k=0}^{N-1} \mathbf{A}^k\right]\mathbf{C}_s\left[\sum_{k=0}^{N-1} \mathbf{A}^k\right]^\top$$

$$+ \frac{1}{N^2}\sum_{t=1}^{N-1}\left[\sum_{k=0}^{N-t-1} \mathbf{A}^k\right](\mathbf{C}_{s+t} - \widetilde{\mathcal{B}} \circ \mathbf{C}_{s+t-1})\left[\sum_{k=0}^{N-t-1} \mathbf{A}^k\right]^\top,$$

where the second equality holds by applying (D.7) $k - t$ times, and the last inequality holds due to Lemma K.4. The decomposition of bias into $\mathbf{M}_3$ and $\mathbf{M}_4$ can be proven in exactly the same manner. □

From Lemma D.2, we can further bound the variance and bias terms as follows.
We have the following bound for variance, whose detailed proof can be found in Appendix G.

**Lemma D.3.** Under Assumptions 3.1, 3.2 and 3.3, with our choice of parameters as in (4.2), we have

$$\text{Variance} \leq \sigma^2 r\left[\frac{27k^*}{2N} + \frac{18(s+N)(q-c\delta)^2}{(1-c)^2}\sum_{i>k^*}\lambda_i^2\right].$$

where $k^* = \max\{k : \lambda_k \geq 2N(q-c\delta)/(1-c)\}$.

The following lemma provides an upper bound for the bias error, whose detailed proof can be found in Appendix H.

**Lemma D.4.** Under Assumptions 3.1, 3.2 and 3.3, and with our choice of parameters as in (4.2), we have

$$\text{Bias} \leq \text{Effective Bias} + \frac{\psi r}{N}\left[\frac{9k^*}{N} + \frac{36N(q-c\delta)^2}{(1-c)^2}\sum_{i>k^*}\lambda_i^2\right]\cdot\left[\frac{14}{\delta}\|\mathbf{w}_0-\mathbf{w}^*\|_{\mathbf{I}_{0:\widehat{k}}}^2\right.$$

$$\left. + \frac{10}{1-c}\|\mathbf{w}_0-\mathbf{w}^*\|_{\mathbf{H}_{\widehat{k}:k^\dagger}}^2 + \frac{1-c}{q-c\delta}\|\mathbf{w}_0-\mathbf{w}^*\|_{\mathbf{I}_{k^\dagger:k^*}}^2 + 4(s+N)\|\mathbf{w}_0-\mathbf{w}^*\|_{\mathbf{H}_{k^*:\infty}}^2\right],$$

where

$$\text{Effective Bias} \leq \frac{8(c\delta/q)^{2s}}{N^2\delta^2}\|\mathbf{w}_0-\mathbf{w}^*\|_{\mathbf{H}_{0:k^\ddagger}^{-1}}^2 + \frac{4s^2}{N^2}c^s\|(\mathbf{I}-\delta\mathbf{H})^{s/2}(\mathbf{w}_0-\mathbf{w}^*)\|_{\mathbf{H}_{k^\ddagger:k^\dagger}}^2$$

$$+ \frac{16c^s}{N^2\delta^2}\|(\mathbf{I}-\delta\mathbf{H})^{s/2}(\mathbf{w}_0-\mathbf{w}^*)\|_{\mathbf{H}_{k^\ddagger:\widehat{k}}^{-1}}^2 + \frac{100c^s}{N^2(1-c)^2}\|(\mathbf{I}-\delta\mathbf{H})^{s/2}(\mathbf{w}_0-\mathbf{w}^*)\|_{\mathbf{H}_{\widehat{k}:k^\dagger}}^2$$

$$+ \frac{9(1-c)^2}{2N^2(q-c\delta)^2}\left\|\left(\mathbf{I}-\frac{q-c\delta}{1-c}\mathbf{H}\right)^s(\mathbf{w}_0-\mathbf{w}^*)\right\|_{\mathbf{H}_{k^\dagger:k^*}^{-1}}^2$$

$$+ 18\left\|\left(\mathbf{I}-\frac{q-c\delta}{1-c}\mathbf{H}\right)^s(\mathbf{w}_0-\mathbf{w}^*)\right\|_{\mathbf{H}_{k^*:\infty}}^2.$$

Substituting Lemma D.3 and Lemma D.4 into (D.1) in Lemma D.1 yields our final result presented in Theorem 4.1.

### D.2 PROOF OF THEOREM 5.1

We consider the linear regression instance where the samples $\mathbf{x}_t$ follow the Gaussian distribution $\mathcal{N}(\mathbf{0},\mathbf{H})$ where $\lambda_i = i^{-2}$, so $\psi = 3$ in Assumption 3.2. The hyperparameters of ASGD are chosen as $\delta = 0.1$, $\alpha = 0.9875$, $\beta = (1-\alpha)/\alpha$, $\widetilde{\kappa} = 5$, $\gamma = \delta/(\psi\widetilde{\kappa}\beta) = 79/150$ and $N = 500$. Finally, we require $(\mathbf{w}_0-\mathbf{w}^*)_i = 0$ for $i \geq 8$.
We now present a formal expression of Theorem 5.1:

**Theorem D.5** (Restatement of Theorem 5.1). When applied to the aforementioned class of problem instances and initialization such that $\|\mathbf{w}_0-\mathbf{w}^*\|_{\mathbf{H}}^2 = \mathcal{O}(\sigma^2)$, the excess risk of SGD satisfies

$$\mathbb{E}[L(\overline{\mathbf{w}}_{s:s+N}^{\text{SGD}})] - L(\mathbf{w}^*) = \Omega(\sigma^2(N^{-1/2} + N^{-2}\cdot 0.996^s)),$$

and the excess risk of ASGD satisfies

$$\mathbb{E}[L(\overline{\mathbf{w}}_{s:s+N})] - L(\mathbf{w}^*) = \mathcal{O}(\sigma^2(N^{-1/2} + N^{-2}\cdot 0.9873^s)).$$

*Proof.* We first recall the excess risk lower bound for SGD given by Theorem 5.2 of Zou et al. (2021b):

$$\mathbb{E}[L(\overline{\mathbf{w}}_{s:s+N}^{\text{SGD}})] - L(\mathbf{w}^*) \geq \underbrace{\frac{\sigma^2}{600}\left[\frac{k_{\text{SGD}}^*}{N} + (s+N)\delta^2\sum_{i>k_{\text{SGD}}^*}\lambda_i^2\right]}_{\text{Variance}}$$

$$+ \underbrace{\frac{1}{100\delta^2 N^2}\cdot\|(\mathbf{I}-\delta\mathbf{H})^s(\mathbf{w}_0-\mathbf{w}^*)\|_{\mathbf{H}_{0:k^*}^{-1}}^2 + \frac{1}{100}\cdot\|(\mathbf{I}-\gamma\mathbf{H})^s(\mathbf{w}_0-\mathbf{w}^*)\|_{\mathbf{H}_{k^*:\infty}}^2}_{\text{EffectiveBias}},$$

As $c = 2\alpha - 1$ and $q = \alpha\delta + (1-\alpha)\gamma$, we have $c = 0.975$ and $q = 79/750$. By definition of $\widehat{k}$, $k^\ddagger$, $k^\dagger$ in (4.1), we have

$$k^\ddagger = 0,\ \widehat{k} = 2,\ k^\dagger = 6.$$

The analysis of the Variance term is given in Corollary 5.2. For the EffectiveBias term, note that all coefficients are absolute constants, so it suffices to consider the exponential decay rate in the eigen-subspace of $\lambda_7$. For SGD, the exponential decay rate is $(1-\delta\lambda_i) = 0.996^s$, and for ASGD, the exponential decay rate is $(1-(\gamma+\delta)\lambda_i/2)^s = 0.9873^s$.

$\square$

# E  PROPERTIES OF $\mathbf{A}_i$

## E.1  SEGMENTATION OF EIGEN-SUBSPACES

Recall that $\mathbf{A}_i$ is defined as

$$\mathbf{A}_i := \begin{bmatrix} 0 & 1 - \delta\lambda_i \\ -c & 1 + c - q\lambda_i \end{bmatrix}, \tag{E.1}$$

so the eigenvalues of $\mathbf{A}_i$ are

$$x_1 = \frac{1 + c - q\lambda_i}{2} - \frac{\sqrt{(1 + c - q\lambda_i)^2 - 4c(1 - \delta\lambda_i)}}{2}, \tag{E.2}$$

$$x_2 = \frac{1 + c - q\lambda_i}{2} + \frac{\sqrt{(1 + c - q\lambda_i)^2 - 4c(1 - \delta\lambda_i)}}{2}. \tag{E.3}$$

From (E.2) and (E.3), we see that whether $\mathbf{A}_i$ has complex or real eigenvalues depends on whether the following holds:

$$(1 + c - q\lambda_i)^2 - 4c(1 - \delta\lambda_i) < 0. \tag{E.4}$$

Directly solving (E.4), we have

$$(\sqrt{q - c\delta} - \sqrt{c(q - \delta)})^2/q^2 < \lambda_i < (\sqrt{q - c\delta} + \sqrt{c(q - \delta)})^2/q^2.$$

Define the eigenvalue cutoffs as

$$k^\dagger := \max\{i : \lambda_i > (\sqrt{q - c\delta} - \sqrt{c(q - \delta)})^2/q^2\}, \tag{E.5}$$

$$k^\ddagger := \max\{i : \lambda_i \geq (\sqrt{q - c\delta} + \sqrt{c(q - \delta)})^2/q^2\}, \tag{E.6}$$

and we note that

$$\frac{(\sqrt{q - c\delta} - \sqrt{c(q - \delta)})^2}{q^2} = \frac{1 - c}{q} \cdot \frac{\sqrt{q - c\delta} - \sqrt{c(q - \delta)}}{\sqrt{q - c\delta} + \sqrt{c(q - \delta)}} = \frac{(1 - c)^2}{(\sqrt{q - c\delta} + \sqrt{c(q - \delta)})^2}$$

$$\frac{(\sqrt{q - c\delta} + \sqrt{c(q - \delta)})^2}{q^2} = \frac{1 - c}{q} \cdot \frac{\sqrt{q - c\delta} + \sqrt{c(q - \delta)}}{\sqrt{q - c\delta} - \sqrt{c(q - \delta)}} = \frac{(1 - c)^2}{(\sqrt{q - c\delta} - \sqrt{c(q - \delta)})^2}$$

Thus, if $i \leq k^\ddagger$ or $i > k^\dagger$, then $\mathbf{A}_i$ has real eigenvalues; If $k^\ddagger < i \leq k^\dagger$, then $\mathbf{A}_i$ has complex eigenvalues. We also define two other important eigenvalue cutoffs

$$\widehat{k} := \max\{i : \lambda_i \geq (1 - c)/\delta\} \tag{E.7}$$

and

$$k^* := \max\left\{i : \lambda_i \geq \frac{1 - c}{2N(q - c\delta)}\right\}.$$

We have the following lemma concerning the cutoff of eigenvalues:

**Lemma E.1.** Let $k^\dagger$ and $k^\ddagger$ be defined in (E.5) and (E.6). Then we have

- For all $i > k^\dagger$, we have
$$\lambda_i \leq \frac{1 - c}{q} \leq \frac{1 - c}{\delta};$$

- For all $i \leq k^\ddagger$, we have
$$\lambda_i \geq \frac{1 - c}{\delta}.$$

*Proof.* For all $i > k^\dagger$, according to (E.5), we have

$$\lambda_i \leq \frac{1 - c}{q} \cdot \frac{\sqrt{q - c\delta} - \sqrt{c(q - \delta)}}{\sqrt{q - c\delta} + \sqrt{c(q - \delta)}} \leq \frac{1 - c}{q} \leq \frac{1 - c}{\delta},$$

where the second inequality holds because $\frac{\sqrt{q-c\delta}-\sqrt{c(q-\delta)}}{\sqrt{q-c\delta}+\sqrt{c(q-\delta)}} \leq 1$, and the last inequality holds because $q \geq \delta$.

For all $i > k^{\ddagger}$, we have

$$
\begin{aligned}
\lambda_i - \frac{1-c}{\delta} &\geq \frac{1-c}{q} \cdot \frac{\sqrt{q-c\delta}+\sqrt{c(q-\delta)}}{\sqrt{q-c\delta}-\sqrt{c(q-\delta)}} - \frac{1-c}{\delta} \\
&= (1-c)\sqrt{c(q-\delta)} \cdot \frac{(q+\delta)-\sqrt{(q-\delta)(q-c\delta)/c}}{\delta q(\sqrt{q-c\delta}-\sqrt{c(q-\delta)})} \\
&\geq \frac{(1-c)\sqrt{c(q-\delta)}}{\delta q(\sqrt{q-c\delta}-\sqrt{c(q-\delta)})} \cdot [(q+\delta)-(q-c\delta)] \\
&= \frac{(1-c^2)\sqrt{c(q-\delta)}}{q(\sqrt{q-c\delta}-\sqrt{c(q-\delta)})} \geq 0,
\end{aligned}
$$

where the first inequality holds due to (E.6), and the second inequality holds because $q - \delta \leq c(q - c\delta)$. $\qquad\square$

With Lemma E.1, we immediately know that $k^{\ddagger} \leq \widehat{k} \leq k^{\dagger}$. If we also assume that $N(1-c) \geq 2$, then

$$
\frac{1-c}{2N(q-c\delta)} \leq \frac{(1-c)^2}{4(q-c\delta)} \leq \frac{(1-c)^2}{(\sqrt{q-c\delta}+\sqrt{c(q-\delta)})^2},
$$

where the inequality holds because $c(q-\delta) \leq q - c\delta$. We thus have $k^* \geq k^{\dagger}$.

We then provide bounds for the spectral norm of $\mathbf{A}_i$. The bounds are accurate in the sense that when $x_1, x_2$ are real, the upper bound of $1 - x_2$ is at most the multiply of a constant of its lower bound.

**Lemma E.2.** Let $x_1$ and $x_2$ be defined in (E.2) and (E.3). Then we have

- If $i \leq k^{\ddagger}$, then $x_1, x_2$ are real, $x_2$ is an increasing function in $\lambda_i$, and

$$
\frac{c\delta - \sqrt{c(q-\delta)(q-c\delta)}}{q} \leq x_2 \leq \frac{c\delta}{q};
$$

- If $k^{\ddagger} < i \leq k^{\dagger}$, then $x_1, x_2$ are complex, and

$$
|x_1| = |x_2| = \sqrt{c(1-\delta\lambda_i)};
$$

- If $k > k^{\dagger}$, then $x_1, x_2$ are real, and

$$
1 - \frac{\sqrt{q-c\delta}(\sqrt{q-c\delta}+\sqrt{c(q-\delta)})}{1-c}\lambda_i \leq x_2 \leq 1 - \frac{q-c\delta}{1-c}\lambda_i
$$

*Proof.* If $i \leq k^{\ddagger}$, then by definition of $x_2$, we have

$$
\begin{aligned}
c - x_2 &= \frac{q\lambda_i + c - 1 - \sqrt{(1+c-q\lambda_i)^2 - 4c(1-\delta\lambda_i)}}{2} \\
&= \frac{2c(q-\delta)\lambda_i}{q\lambda_i + c - 1 + \sqrt{(1+c-q\lambda_i)^2 - 4c(1-\delta\lambda_i)}} \\
&= \frac{2c(q-\delta)}{q} \cdot \frac{1}{1 - \frac{1-c}{q\lambda_i} + \sqrt{\left(1 - \frac{1-c}{q\lambda_i} \cdot \frac{\sqrt{q-c\delta}+\sqrt{c(q-\delta)}}{\sqrt{q-c\delta}-\sqrt{c(q-\delta)}}\right)\left(1 - \frac{1-c}{q\lambda_i} \cdot \frac{\sqrt{q-c\delta}-\sqrt{c(q-\delta)}}{\sqrt{q-c\delta}+\sqrt{c(q-\delta)}}\right)}}
\end{aligned}
$$

Note that the denominator is decreasing as a function of $(1-c)/(q\lambda_i)$, so we have

$$
1 - \frac{1-c}{q\lambda_i} + \sqrt{\left(1 - \frac{1-c}{q\lambda_i} \cdot \frac{\sqrt{q-c\delta}+\sqrt{c(q-\delta)}}{\sqrt{q-c\delta}-\sqrt{c(q-\delta)}}\right)\left(1 - \frac{1-c}{q\lambda_i} \cdot \frac{\sqrt{q-c\delta}-\sqrt{c(q-\delta)}}{\sqrt{q-c\delta}+\sqrt{c(q-\delta)}}\right)}
$$

$$
\leq 1 - 0 + 1 = 2;
$$

we also have

$$
1 - \frac{1-c}{q\lambda_i} + \sqrt{\left(1 - \frac{1-c}{q\lambda_i} \cdot \frac{\sqrt{q-c\delta}+\sqrt{c(q-\delta)}}{\sqrt{q-c\delta}-\sqrt{c(q-\delta)}}\right)\left(1 - \frac{1-c}{q\lambda_i} \cdot \frac{\sqrt{q-c\delta}-\sqrt{c(q-\delta)}}{\sqrt{q-c\delta}+\sqrt{c(q-\delta)}}\right)}
$$

$$\geq 1 - \frac{\sqrt{q - c\delta} - \sqrt{c(q - \delta)}}{\sqrt{q - c\delta} + \sqrt{c(q - \delta)}} = \frac{2\sqrt{c(q - \delta)}}{\sqrt{q - c\delta} + \sqrt{c(q - \delta)}}.$$

Therefore, we have

$$x_2 \leq c - \frac{2c(q - \delta)}{2q} = \frac{c\delta}{q},$$

and

$$x_2 \geq c - \frac{2c(q - \delta)}{q} \cdot \frac{\sqrt{q - c\delta} + \sqrt{c(q - \delta)}}{2\sqrt{c(q - \delta)}} = \frac{c\delta - \sqrt{c(q - \delta)(q - c\delta)}}{q}.$$

If $k^{\ddagger} < i \leq k^{\dagger}$, then we have

$$x_1 x_2 = c(1 - \delta\lambda_i),$$

where $x_1 = \bar{x}_2$. Thus, $|x_1| = |x_2| = \sqrt{c(1 - \delta\lambda_i)}$.

If $i > k^{\dagger}$, then we have

$$1 - x_2 = \frac{1 - c + q\lambda_i - \sqrt{(1 + c - q\lambda_i)^2 - 4c(1 - \delta\lambda_i)}}{2}$$

$$= \frac{2(q - c\delta)\lambda_i}{1 - c + q\lambda_i + \sqrt{(1 + c - q\lambda_i)^2 - 4c(1 - \delta\lambda_i)}}. \quad \text{(E.8)}$$

Note that

$$\frac{\partial}{\partial \lambda_i} \left( 1 - c + q\lambda_i + \sqrt{(1 + c - q\lambda_i)^2 - 4c(1 - \delta\lambda_i)} \right) = q + \frac{2c\delta - q(1 + c - q\lambda_i)}{\sqrt{(1 + c - q\lambda_i)^2 - 4c(1 - \delta\lambda_i)}}$$

$$= \frac{-4c(q - \delta)(q - c\delta)}{\sqrt{(1 + c - q\lambda_i)^2 - 4c(1 - \delta\lambda_i)}\{q\sqrt{(1 + c - q\lambda_i)^2 - 4c(1 - \delta\lambda_i)} + [(1 + c)q - 2c\delta - q^2\lambda_i]\}}.$$

We also note that

$$q\sqrt{(1 + c - q\lambda_i)^2 - 4c(1 - \delta\lambda_i)} + [(1 + c)q - 2c\delta - q^2\lambda_i]$$
$$\geq (1 + c)q - 2c\delta - q^2\lambda_i$$
$$\geq (1 + c)q - 2c\delta - (\sqrt{c(q - \delta)} - \sqrt{c(q - \delta)})^2$$
$$= 2\sqrt{c(q - \delta)(q - c\delta)} \geq 0,$$

where the first inequality holds because $\sqrt{(1 + c - q\lambda_i)^2 - 4c(1 - \delta\lambda_i)} \geq 0$, the second inequality holds because due to (E.5). Therefore, we have

$$\frac{\partial}{\partial \lambda_i} \left( 1 - c + q\lambda_i + \sqrt{(1 + c - q\lambda_i)^2 - 4c(1 - \delta\lambda_i)} \right) \leq 0,$$

so $1 - c + q\lambda_i + \sqrt{(1 + c - q\lambda_i)^2 - 4c(1 - \delta\lambda_i)}$ is a function decreasing in $\lambda_i$. We thus have

$$1 - c + q\lambda_i + \sqrt{(1 + c - q\lambda_i)^2 - 4c(1 - \delta\lambda_i)} \leq 1 - c + \sqrt{(1 + c)^2 - 4c} = 2(1 - c),$$

and

$$1 - c + q\lambda_i + \sqrt{(1 + c - q\lambda_i)^2 - 4c(1 - \delta\lambda_i)}$$
$$\geq 1 - c + (1 - c)\frac{\sqrt{q - c\delta} - \sqrt{c(q - \delta)}}{\sqrt{q - c\delta} + \sqrt{c(q - \delta)}} = \frac{2(1 - c)\sqrt{q - c\delta}}{\sqrt{q - c\delta} + \sqrt{c(q - \delta)}}.$$

Therefore, we have

$$x_2 \leq 1 - \frac{2(q - c\delta)\lambda_i}{2(1 - c)} = 1 - \frac{q - c\delta}{1 - c}\lambda_i,$$

and

$$x_2 \geq 1 - \frac{2(q - c\delta)\lambda_i}{\frac{2(1 - c)\sqrt{q - c\delta}}{\sqrt{q - c\delta} + \sqrt{c(q - \delta)}}} = 1 - \frac{\sqrt{q - c\delta}(\sqrt{q - c\delta} + \sqrt{c(q - \delta)})}{1 - c}\lambda_i.$$

$\square$

## E.2 CHARACTERIZATION OF $\mathbf{A}_i^k$

Before we prove the upper bound for variance and bias, we first characterize the property of $\mathbf{A}_i^k$ for $k \geq 1$ and $i \in [1, d]$, i.e., each block of matrix $\mathbf{A}$ corresponding to each eigenvalue $\lambda_i$ of $\mathbf{H}$.

**Lemma E.3.** Let $\mathbf{A}_i$ be defined as in (E.1), write $\mathbf{A}_i^k$ as

$$\mathbf{A}_i^k = \begin{bmatrix} (\mathbf{A}_i^k)_{11} & (\mathbf{A}_i^k)_{12} \\ (\mathbf{A}_i^k)_{21} & (\mathbf{A}_i^k)_{22} \end{bmatrix}.$$

Let the eigenvalues of $\mathbf{A}_i$ be $x_1$ and $x_2$ as defined in (E.2) and (E.3). Then, for any integer $k \geq 1$, we have

$$(\mathbf{A}_i^k)_{11} = -c(1 - \delta\lambda_i)\frac{x_2^{k-1} - x_1^{k-1}}{x_2 - x_1},$$

$$(\mathbf{A}_i^k)_{12} = (1 - \delta\lambda_i)\frac{x_2^k - x_1^k}{x_2 - x_1},$$

$$(\mathbf{A}_i^k)_{21} = -c\frac{x_2^k - x_1^k}{x_2 - x_1},$$

$$(\mathbf{A}_i^k)_{22} = \frac{x_2^{k+1} - x_1^{k+1}}{x_2 - x_1}.$$

*Proof.* We prove Lemma E.3 by induction. For $k = 1$, we trivially have

$$-c(1 - \delta\lambda_i)\frac{x_2^0 - x_1^0}{x_2 - x_1} = 0, \quad (1 - \delta\lambda_i)\frac{x_2^1 - x_1^1}{x_2 - x_1} = 1 - \delta\lambda_i, \quad -c\frac{x_2^1 - x_1^1}{x_2 - x_1} = -c.$$

We also have

$$\frac{x_2^2 - x_1^2}{x_2 - x_1} = x_1 + x_2 = 1 + c - q\lambda_i.$$

Therefore, Lemma E.3 holds for $k = 1$. Suppose that the lemma holds for $k$. Note that $\mathbf{A}_i^{k+1} = \mathbf{A}_i \cdot \mathbf{A}_i^k$, so by induction hypothesis, we have

$$(\mathbf{A}_i^{k+1})_{11} = (1 - \delta\lambda_i)(\mathbf{A}_i^k)_{21} = -c(1 - \delta\lambda_i)\frac{x_2^k - x_1^k}{x_2 - x_1},$$

$$(\mathbf{A}_i^{k+1})_{12} = (1 - \delta\lambda_i)(\mathbf{A}_i^k)_{22} = (1 - \delta\lambda_i)\frac{x_2^{k+1} - x_1^{k+1}}{x_2 - x_1},$$

$$\begin{aligned}(\mathbf{A}_i^{k+1})_{21} &= -c(\mathbf{A}_i^k)_{11} + (1 + c - q\lambda_i)(\mathbf{A}_i^k)_{21} \\ &= c^2(1 - \delta\lambda_i)\frac{x_2^{k-1} - x_1^{k-1}}{x_2 - x_1} - c(1 + c - q\lambda_i)\frac{x_2^k - x_1^k}{x_2 - x_1} \\ &= c \cdot x_1 x_2 \cdot \frac{x_2^{k-1} - x_1^{k-1}}{x_2 - x_1} - c(x_1 + x_2)\frac{x_2^k - x_1^k}{x_2 - x_1} \\ &= -c\frac{x_2^{k+1} - x_1^{k+1}}{x_2 - x_1},\end{aligned}$$

$$\begin{aligned}(\mathbf{A}_i^{k+1})_{22} &= -c(\mathbf{A}_i^k)_{12} + (1 + c - q\lambda_i)(\mathbf{A}_i^k)_{22} \\ &= -c(1 - \delta\lambda_i)\frac{x_2^k - x_1^k}{x_2 - x_1} + (1 + c - q\lambda_i)\frac{x_2^{k+1} - x_1^{k+1}}{x_2 - x_1} \\ &= -x_1 x_2 \cdot \frac{x_2^k - x_1^k}{x_2 - x_1} + (x_1 + x_2) \cdot \frac{x_2^{k+1} - x_1^{k+1}}{x_2 - x_1} \\ &= \frac{x_2^{k+2} - x_1^{k+2}}{x_2 - x_1},\end{aligned}$$

where we used the property that $x_1 + x_2 = 1 + c - q\lambda_i$ and $x_1 x_2 = c(1 - \delta\lambda_i)$. Therefore, Lemma E.3 holds for $k + 1$, and induction is completed. $\square$

# F LINEAR OPERATORS AND EFFECT OF FOURTH MOMENT

## F.1 PROPERTIES OF LINEAR OPERATORS

In this section, we introduce linear operators on matrices as well as their properties. We first give the following definitions of linear operators:

$$\mathcal{I} := \mathbf{I} \otimes \mathbf{I}, \quad \mathcal{M} := \mathbb{E}[\mathbf{x} \otimes \mathbf{x} \otimes \mathbf{x} \otimes \mathbf{x}], \quad \widetilde{\mathcal{M}} := \mathbf{H} \otimes \mathbf{H},$$
$$\mathcal{B} := \mathbb{E}[\widehat{\mathbf{A}}_t \otimes \widehat{\mathbf{A}}_t], \quad \widetilde{\mathcal{B}} := \mathbf{A} \otimes \mathbf{A}. \tag{F.1}$$

$\widehat{\mathbf{A}}_t$ can be defined as the sum of deterministic component $\mathbf{V}_1$ and stochastic component $\widehat{\mathbf{V}}_2$:

$$\mathbf{V}_1 = \begin{bmatrix} \mathbf{0} & \mathbf{I} \\ -c\mathbf{I} & (1+c)\mathbf{I} \end{bmatrix}, \quad \widehat{\mathbf{V}}_2 = \begin{bmatrix} \mathbf{0} & -\delta \mathbf{x}_t \mathbf{x}_t^\top \\ \mathbf{0} & -q \mathbf{x}_t \mathbf{x}_t^\top \end{bmatrix}. \tag{F.2}$$

Define

$$\mathbf{V}_2 := \mathbb{E}[\widehat{\mathbf{V}}_2] = \begin{bmatrix} \mathbf{0} & -\delta \mathbf{H} \\ \mathbf{0} & -q\mathbf{H} \end{bmatrix}, \tag{F.3}$$

then $\mathbf{A} = \mathbf{V}_1 + \mathbf{V}_2$. We are also interested in linear operators $\mathbb{E}[\widehat{\mathbf{V}}_2 \otimes \widehat{\mathbf{V}}_2]$ and $\mathbf{V}_2 \otimes \mathbf{V}_2$. We introduce the concept of PSD operators:

**Definition F.1** (PSD operator). An operator $\mathcal{O}$ defined on symmetric matrices is called a PSD operator if $\mathbf{M} \succeq \mathbf{0}$ implies $\mathcal{O} \circ \mathbf{M} \succeq \mathbf{0}$.

The following lemma summarizes some basic properties of the linear operators.

**Lemma F.2.** The operators defined in (F.1) have the following properties:

(a) $\mathcal{M}$, $\widetilde{\mathcal{M}}$, and $\mathcal{M} - \widetilde{\mathcal{M}}$ are PSD operators.

(b) For any PSD matrix $\mathbf{M} \in \mathbb{R}^{2d \times 2d}$, let

$$\mathbf{M} := \begin{bmatrix} \mathbf{M}_{11} & \mathbf{M}_{12} \\ \mathbf{M}_{21} & \mathbf{M}_{22} \end{bmatrix}, \tag{F.4}$$

where $\mathbf{M}_{11}, \mathbf{M}_{12}, \mathbf{M}_{21}$ and $\mathbf{M}_{22}$ are $d$-by-$d$ blocks. We then have

$$\mathbb{E}[\widehat{\mathbf{V}}_2 \otimes \widehat{\mathbf{V}}_2] \circ \mathbf{M} = \begin{bmatrix} \delta^2 & \delta q \\ \delta q & q^2 \end{bmatrix} \otimes (\mathcal{M} \circ \mathbf{M}_{22}),$$
$$(\mathbf{V}_2 \otimes \mathbf{V}_2) \circ \mathbf{M} = \begin{bmatrix} \delta^2 & \delta q \\ \delta q & q^2 \end{bmatrix} \otimes (\widetilde{\mathcal{M}} \circ \mathbf{M}_{22}).$$

Thus, $\mathbb{E}[\widehat{\mathbf{V}}_2 \otimes \widehat{\mathbf{V}}_2]$ and $\mathbf{V}_2 \otimes \mathbf{V}_2$ are both PSD operators.

(c) $\mathcal{B}$ and $\widetilde{\mathcal{B}}$ are both PSD operators.

(d) $\mathcal{B} - \widetilde{\mathcal{B}} = \mathbb{E}[\widehat{\mathbf{V}}_2 \otimes \mathbf{V}_2] - \mathbf{V}_2 \otimes \mathbf{V}_2$ is a PSD operator.

*Proof.* The proof follows those of Jain et al. (2018), Zou et al. (2021b), and Wu et al. (2022).

(a) For any PSD matrix $\mathbf{M}$, we have

$$\mathcal{M} \circ \mathbf{M} = \mathbb{E}[\mathbf{x}\mathbf{x}^\top \mathbf{M}\mathbf{x}\mathbf{x}^\top] = \mathbb{E}[(\mathbf{x}^\top \mathbf{M}\mathbf{x})\mathbf{x}\mathbf{x}^\top] \succeq \mathbf{0},$$

where the inequality holds because $\mathbf{x}^\top \mathbf{M}\mathbf{x} \geq 0$ and $\mathbf{x}\mathbf{x}^\top \succeq \mathbf{0}$. Furthermore,

$$\widetilde{\mathcal{M}} \circ \mathbf{M} = \mathbf{H}\mathbf{M}\mathbf{H} \succeq \mathbf{0},$$

where the inequality holds because $\mathbf{M} \succeq \mathbf{0}$ and $\mathbf{H}$ is symmetric. Lastly,

$$(\mathcal{M} - \widetilde{\mathcal{M}}) \circ \mathbf{M} = \mathbb{E}[\mathbf{x}\mathbf{x}^\top \mathbf{M}\mathbf{x}\mathbf{x}^\top] - \mathbf{H}\mathbf{M}\mathbf{H} = \mathbb{E}[(\mathbf{x}\mathbf{x}^\top - \mathbf{H})\mathbf{M}(\mathbf{x}\mathbf{x}^\top - \mathbf{H})] \succeq \mathbf{0},$$

where the inequality holds because $\mathbf{M} \succeq \mathbf{0}$ and $\mathbf{x}\mathbf{x}^\top - \mathbf{H}$ is symmetric.

(b) Note that $\mathbf{M}_{22} \succeq \mathbf{0}$ because $\mathbf{M} \succeq \mathbf{0}$. We thus have

$$
\begin{aligned}
\mathbb{E}[\widehat{\mathbf{V}}_2 \otimes \widehat{\mathbf{V}}_2] \circ \mathbf{M} &= \mathbb{E}\left[\widehat{\mathbf{V}}_2 \mathbf{M} \widehat{\mathbf{V}}_2^\top\right] \\
&= \mathbb{E}\left[\begin{bmatrix} \mathbf{0} & -\delta\mathbf{x}_t\mathbf{x}_t^\top \\ \mathbf{0} & -q\mathbf{x}_t\mathbf{x}_t^\top \end{bmatrix} \begin{bmatrix} \mathbf{M}_{11} & \mathbf{M}_{12} \\ \mathbf{M}_{21} & \mathbf{M}_{22} \end{bmatrix} \begin{bmatrix} \mathbf{0} & \mathbf{0} \\ -\delta\mathbf{x}_t\mathbf{x}_t^\top & -q\mathbf{x}_t\mathbf{x}_t^\top \end{bmatrix}\right] \\
&= \mathbb{E}\left[\begin{bmatrix} \delta^2\mathbf{x}_t\mathbf{x}_t^\top\mathbf{M}_{22}\mathbf{x}_t\mathbf{x}_t^\top & \delta q\mathbf{x}_t\mathbf{x}_t^\top\mathbf{M}_{22}\mathbf{x}_t\mathbf{x}_t^\top \\ \delta q\mathbf{x}_t\mathbf{x}_t^\top\mathbf{M}_{22}\mathbf{x}_t\mathbf{x}_t^\top & q^2\mathbf{x}_t\mathbf{x}_t^\top\mathbf{M}_{22}\mathbf{x}_t\mathbf{x}_t^\top \end{bmatrix}\right] \\
&= \begin{bmatrix} \delta^2 & \delta q \\ \delta q & q^2 \end{bmatrix} \otimes \mathbb{E}\left[\mathbf{x}_t\mathbf{x}_t^\top\mathbf{M}_{22}\mathbf{x}_t\mathbf{x}_t^\top\right] \\
&= \begin{bmatrix} \delta^2 & \delta q \\ \delta q & q^2 \end{bmatrix} \otimes (\mathcal{M} \circ \mathbf{M}_{22}) \succeq \mathbf{0},
\end{aligned}
$$

where the last inequality holds because $\mathbf{M}_{22} \succeq \mathbf{0}$, $\mathcal{M}$ is a PSD operator and $\begin{bmatrix} \delta^2 & \delta q \\ \delta q & q^2 \end{bmatrix} \succeq \mathbf{0}$.
In a similar way,

$$
\begin{aligned}
(\mathbf{V}_2 \otimes \mathbf{V}_2) \circ \mathbf{M} &= \mathbf{V}_2 \mathbf{M} \mathbf{V}_2^\top \\
&= \begin{bmatrix} \mathbf{0} & -\delta\mathbf{H} \\ \mathbf{0} & -q\mathbf{H} \end{bmatrix} \begin{bmatrix} \mathbf{M}_{11} & \mathbf{M}_{12} \\ \mathbf{M}_{21} & \mathbf{M}_{22} \end{bmatrix} \begin{bmatrix} \mathbf{0} & \mathbf{0} \\ -\delta\mathbf{H} & -q\mathbf{H} \end{bmatrix} \\
&= \begin{bmatrix} \delta^2\mathbf{H}\mathbf{M}_{22}\mathbf{H} & \delta q\mathbf{H}\mathbf{M}_{22}\mathbf{H} \\ \delta q\mathbf{H}\mathbf{M}_{22}\mathbf{H} & q^2\mathbf{H}\mathbf{M}_{22}\mathbf{H} \end{bmatrix} \\
&= \begin{bmatrix} \delta^2 & \delta q \\ \delta q & q^2 \end{bmatrix} \otimes \mathbf{H}\mathbf{M}_{22}\mathbf{H} \\
&= \begin{bmatrix} \delta^2 & \delta q \\ \delta q & q^2 \end{bmatrix} \otimes (\widetilde{\mathcal{M}} \circ \mathbf{M}_{22}) \succeq \mathbf{0},
\end{aligned}
$$

where the inequality holds because $\mathbf{M}_{22} \succeq \mathbf{0}$, $\widetilde{\mathcal{M}}$ is a PSD operator, and $\begin{bmatrix} \delta^2 & \delta q \\ \delta q & q^2 \end{bmatrix} \succeq \mathbf{0}$.

(c) We have
$$
\mathcal{B} \circ \mathbf{M} = \mathbb{E}[\widehat{\mathbf{A}}_t \mathbf{M} \widehat{\mathbf{A}}_t^\top], \quad \widetilde{\mathcal{B}} \circ \mathbf{M} = \mathbf{A}\mathbf{M}\mathbf{A}^\top,
$$
so both $\mathcal{B}$ and $\widetilde{\mathcal{B}}$ are PSD operators.

(d) Note that $\widehat{\mathbf{A}}_t = \mathbf{V}_1 + \widehat{\mathbf{V}}_2$, and $\mathbf{A} = \mathbf{V}_1 + \mathbf{V}_2$, so

$$
\begin{aligned}
(\mathcal{B} - \widetilde{\mathcal{B}}) \circ \mathbf{M} &= (\mathbb{E}[(\mathbf{V}_1 + \widehat{\mathbf{V}}_2) \otimes (\mathbf{V}_1 + \widehat{\mathbf{V}}_2)] - (\mathbf{V}_1 + \mathbf{V}_2) \otimes (\mathbf{V}_1 + \mathbf{V}_2)) \circ \mathbf{M} \\
&= (\mathbb{E}[\widehat{\mathbf{V}}_2 \otimes \widehat{\mathbf{V}}_2] - \mathbf{V}_2 \otimes \mathbf{V}_2) \circ \mathbf{M} \\
&= \begin{bmatrix} \delta^2 & \delta q \\ \delta q & q^2 \end{bmatrix} \otimes ((\mathcal{M} - \widetilde{\mathcal{M}}) \circ \mathbf{M}_{22}) \succeq \mathbf{0},
\end{aligned}
$$

where the second inequality holds because because $\mathbb{E}[\mathbf{V}_1 \otimes \widehat{\mathbf{V}}_2] = \mathbf{V}_1 \otimes \mathbf{V}_2$ and $\mathbb{E}[\widehat{\mathbf{V}}_2 \otimes \mathbf{V}_1] = \mathbf{V}_2 \otimes \mathbf{V}_1$, the third inequality follows from part (b), and the inequality holds because $\mathbf{M}_{22} \succeq \mathbf{0}$, $\mathcal{M} - \widetilde{\mathcal{M}}$ is a PSD operator, and $\begin{bmatrix} \delta^2 & \delta q \\ \delta q & q^2 \end{bmatrix} \succeq \mathbf{0}$.

$\square$

## F.2 ANALYSIS OF FOURTH MOMENT

In this section, we study the difference of operators $\mathcal{B}$ and $\widetilde{\mathcal{B}}$ (due to the fourth moment) when they are operated on PSD matrix $\mathbf{M}$. Specifically, we are interested in bounding the inner product

$$
\left\langle \begin{bmatrix} \mathbf{0} & \mathbf{0} \\ \mathbf{0} & \mathbf{H} \end{bmatrix}, \sum_{j=0}^{t-1} \mathcal{B}^j \circ \mathbf{M} \right\rangle. \tag{F.5}
$$

The following lemma is the starting point of the analysis of fourth moment:

**Lemma F.3.** For any PSD matrix $\mathbf{M}$, we have

$$(\mathcal{B} - \widetilde{\mathcal{B}}) \circ \mathbf{M} \preceq \mathbb{E}[\widehat{\mathbf{V}}_2 \otimes \widehat{\mathbf{V}}_2] \circ \mathbf{M},$$

where

$$\mathbb{E}[\widehat{\mathbf{V}}_2 \otimes \widehat{\mathbf{V}}_2] \circ \mathbf{M} \preceq \psi \left\langle \begin{bmatrix} \mathbf{0} & \mathbf{0} \\ \mathbf{0} & \mathbf{H} \end{bmatrix}, \mathbf{M} \right\rangle \cdot \begin{bmatrix} \delta^2 & \delta q \\ \delta q & q^2 \end{bmatrix} \otimes \mathbf{H}.$$

*Proof.* By Lemma F.2(d), we have

$$(\mathcal{B} - \widetilde{\mathcal{B}}) \circ \mathbf{M} = \left( \mathbb{E}[\widehat{\mathbf{V}}_2 \otimes \widehat{\mathbf{V}}_2] - \mathbf{V}_2 \otimes \mathbf{V}_2 \right) \circ \mathbf{M} \preceq \mathbb{E}[\widehat{\mathbf{V}}_2 \otimes \widehat{\mathbf{V}}_2] \circ \mathbf{M},$$

where the inequality holds due to Lemma F.2(b).

Let $\mathbf{M}_{22}$ be the matrix that contains the last $d$ rows and $d$ columns of $\mathbf{M}$. By definition of $\widehat{\mathbf{V}}_2$ in (F.2), we have

$$\mathbb{E}\left[ \widehat{\mathbf{V}}_2 \otimes \widehat{\mathbf{V}}_2 \right] \circ \mathbf{M} = \begin{bmatrix} \delta^2 & \delta q \\ \delta q & q^2 \end{bmatrix} \otimes (\mathcal{M} \circ \mathbf{M})$$

$$\preceq \psi \operatorname{tr}(\mathbf{H} \mathbf{M}_{22}) \cdot \begin{bmatrix} \delta^2 & \delta q \\ \delta q & q^2 \end{bmatrix} \otimes \mathbf{H}$$

$$= \psi \left\langle \begin{bmatrix} \mathbf{0} & \mathbf{0} \\ \mathbf{0} & \mathbf{H} \end{bmatrix}, \mathbf{M} \right\rangle \cdot \begin{bmatrix} \delta^2 & \delta q \\ \delta q & q^2 \end{bmatrix} \otimes \mathbf{H},$$

where first eqality holds due to Lemma F.2(b), and the inequality holds due to Assumption 3.2. □

The operators $(\mathcal{I} - \mathcal{B})^{-1}$ and $(\mathcal{I} - \mathcal{B})^{-1}$ are of special interest in the analysis of fourth moment. We first show the existence $(\mathcal{I} - \widetilde{\mathcal{B}})^{-1}$.

**Lemma F.4.** With the parameters in (4.2), $(\mathcal{I} - \widetilde{\mathcal{B}})^{-1}$ exists and is a PSD operator.

*Proof.* It suffices to show that the property holds for any rank-one matrix $\mathbf{x}\mathbf{x}^\top$. We have

$$(\mathcal{I} - \widetilde{\mathcal{B}})^{-1} \circ (\mathbf{x}\mathbf{x}^\top) = \sum_{k=0}^{\infty} \widetilde{\mathcal{B}}^k \circ (\mathbf{x}\mathbf{x}^\top) = \sum_{k=0}^{\infty} \mathbf{A}^k (\mathbf{x}\mathbf{x}^\top)(\mathbf{A}^k)^\top = \sum_{k=0}^{\infty} (\mathbf{A}^k \mathbf{x})(\mathbf{A}^k \mathbf{x})^\top.$$

Thus, the $ij$-entry of $(\mathcal{I} - \mathcal{B})^{-1} \circ (\mathbf{x}\mathbf{x}^\top)$ is

$$\sum_{k=0}^{\infty} (\mathbf{A}^k \mathbf{x})_i (\mathbf{A}^k \mathbf{x})_j \leq \sum_{k=0}^{\infty} |(\mathbf{A}^k \mathbf{x})_i| \cdot |(\mathbf{A}^k \mathbf{x})_j| \infty.$$

The series converges because all eigenvalues of $\mathbf{A}$, i.e., eigenvalues of all $\mathbf{A}_i$, have magnitudes strictly smaller than 1. □

We then define operator $\mathcal{T}$ as

$$\mathcal{T} := \mathcal{I} - \mathbf{V}_1 \otimes \mathbf{V}_1 - \mathbf{V}_1 \otimes \mathbf{V}_2 - \mathbf{V}_2 \otimes \mathbf{V}_1 = \mathcal{I} - \widetilde{\mathcal{B}} + \mathbf{V}_2 \otimes \mathbf{V}_2. \tag{F.6}$$

Since $\mathcal{I} - \widetilde{\mathcal{B}}$ is invertible and $(\mathcal{I} - \widetilde{\mathcal{B}})^{-1}$ is a PSD operator, $\mathcal{T}$ is also invertible, and $\mathcal{T}^{-1}$ is a PSD operator. We can thus define matrix $\mathbf{U}$ as

$$\mathbf{U} := \mathcal{T}^{-1} \circ \left( \begin{bmatrix} \delta^2 & \delta q \\ \delta q & q^2 \end{bmatrix} \otimes \mathbf{H} \right). \tag{F.7}$$

The following lemma charantizes a key property of $\mathbf{U}$:

**Lemma F.5** (Modified from Jain et al. (2018)). With the choice of parameters in (4.2), the inner product $\left\langle \begin{bmatrix} \mathbf{0} & \mathbf{0} \\ \mathbf{0} & \mathbf{H} \end{bmatrix}, \mathbf{U} \right\rangle$ is upper bounded by $l$, where

$$l := \frac{\delta \operatorname{tr}(\mathbf{H})}{2} + \frac{1}{2\psi} + \frac{\gamma}{4} \sum_{i > \widetilde{\kappa}} \lambda_i. \tag{F.8}$$

Specifically for SHB where $\delta = 0$, we have

$$\left\langle \begin{bmatrix} \mathbf{0} & \mathbf{0} \\ \mathbf{0} & \mathbf{H} \end{bmatrix}, \mathbf{U} \right\rangle \leq \frac{q \operatorname{tr}(\mathbf{H})}{2(1 - c)}.$$

*Proof.* Denote $\mathbf{U}_i \in \mathbb{R}^{2 \times 2}$ as the $i$-th block of the block-diagonal matrix $\mathbf{U}$. By Equation (56) of Jain et al. (2018), we have

$$(\mathbf{U}_i)_{22} = \frac{(1 + c - c\delta\lambda_i)(q - c\delta) + 2cq\delta\lambda_i}{2(1 - c^2 + c\lambda_i(q + c\delta))} = \frac{\delta}{2} + \frac{(1+c)(q-\delta)}{2(1 - c^2 + c\lambda_i(q + c\delta))}. \tag{F.9}$$

On the one hand, $(\mathbf{U}_i)_{22}$ is bounded by

$$(\mathbf{U}_i)_{22} \leq \frac{\delta}{2} + \frac{(1+c)(q-\delta)}{2((1 - c^2)\delta\lambda_i + c\lambda_i(q + c\delta))} = \frac{\delta}{2} + \frac{(1+c)(q-\delta)}{2(cq + \delta)\lambda_i}$$

$$\leq \frac{\delta}{2} + \frac{(1+c)(q-\delta)}{2(1+c)\delta\lambda_i} = \frac{\delta}{2} + \frac{q-\delta}{1-c} \cdot \frac{1-c}{2\delta\lambda_i},$$

$$= \frac{\delta}{2} + \frac{\gamma - \delta}{2} \cdot \frac{2\alpha\beta}{2\delta\lambda_i} \leq \frac{\delta}{2} + \frac{\gamma}{2} \cdot \frac{\beta}{\delta\lambda_i} = \frac{\delta}{2} + \frac{1}{2\psi\widetilde{\kappa}\lambda_i}, \tag{F.10}$$

where the first inequality holds because $\delta\lambda_i \leq 1$, and the second inequality holds because $q \geq \delta$, and the third inequality holds because $\gamma - \delta \leq \gamma$ and $\alpha\beta \leq \beta$. On the other hand, $(\mathbf{U}_i)_{22}$ can also be bounded by

$$(\mathbf{U}_i)_{22} \leq \frac{\delta}{2} + \frac{(1+c)(q-\delta)}{2(1-c^2)} = \frac{\delta}{2} + \frac{q-\delta}{2(1-c)} = \frac{\delta}{2} + \frac{\gamma - \delta}{4} \leq \frac{\delta}{2} + \frac{\gamma}{4}, \tag{F.11}$$

where the first inequality holds because $1 - c^2 + (q - c\delta)\lambda_i \geq 1 - c^2$, and the second inequality holds because $(\gamma - \delta)/4 \leq \gamma/4$. Thus, we have

$$\left\langle \begin{bmatrix} \mathbf{0} & \mathbf{0} \\ \mathbf{0} & \mathbf{H} \end{bmatrix}, \mathbf{U} \right\rangle = \sum_{i=1}^{d} \lambda_i(\mathbf{U}_i)_{22} \leq \frac{\delta}{2}\sum_{i=1}^{d}\lambda_i + \sum_{i=1}^{\widetilde{\kappa}}\frac{1}{2\psi\widetilde{\kappa}} + \sum_{i>\widetilde{\kappa}}\frac{\gamma\lambda_i}{4} = \frac{\delta\,\mathrm{tr}(\mathbf{H})}{2} + \frac{1}{2\psi} + \frac{\gamma}{4}\sum_{i>\widetilde{\kappa}}\lambda_i,$$

where the inequality holds due to (F.10) for $i \leq \widetilde{\kappa}$ and (F.11) for $i > \widetilde{\kappa}$.
Specifically for SHB, we have

$$(\mathbf{U}_i)_{22} = \frac{(1+c)q}{2((1-c^2) + cq\lambda_i)} \leq \frac{(1+c)q}{2(1-c^2)} = \frac{q}{2(1-c)},$$

where the inequality holds because $2((1-c^2) + cq\lambda_i) \geq 2(1-c^2)$. We thus have

$$\left\langle \begin{bmatrix} \mathbf{0} & \mathbf{0} \\ \mathbf{0} & \mathbf{H} \end{bmatrix}, \mathbf{U} \right\rangle = \sum_{i=1}^{d}\lambda_i(\mathbf{U}_i)_{22} \leq \frac{q\,\mathrm{tr}(\mathbf{H})}{2(1-c)}.$$

$\square$

The following lemma charaterizes $(\mathcal{I} - \mathcal{B})^{-1}$ in terms of $\mathcal{T}$ and $\widehat{\mathbf{V}}_2$:

**Lemma F.6.** The operator $(\mathcal{I} - \mathcal{B})^{-1}$ can be written in the form of geometric series

$$(\mathcal{I} - \mathcal{B})^{-1} = \sum_{k=0}^{\infty}(\mathcal{T}^{-1}\mathbb{E}[\widehat{\mathbf{V}}_2 \otimes \widehat{\mathbf{V}}_2])^k \circ \mathcal{T}^{-1}.$$

*Proof.* According to the definition of $\mathcal{B}$,

$$\mathcal{B} = \mathbb{E}\left[\widehat{\mathbf{A}}_t \otimes \widehat{\mathbf{A}}_t\right] = \mathbb{E}\left[(\mathbf{V}_1 + \widehat{\mathbf{V}}_2) \otimes (\mathbf{V}_1 + \widehat{\mathbf{V}}_2)\right]$$

$$= \mathbf{V}_1 \otimes \mathbf{V}_1 + \mathbf{V}_1 \otimes \mathbf{V}_2 + \mathbf{V}_2 \otimes \mathbf{V}_1 + \mathbb{E}\left[\widehat{\mathbf{V}}_2 \otimes \widehat{\mathbf{V}}_2\right],$$

where the last equality holds because $\mathbb{E}[\widehat{\mathbf{V}}_2] = \mathbf{V}_2$. We thus have

$$(\mathcal{I} - \mathcal{B})^{-1} = \left(\mathcal{T} - \mathbb{E}[\widehat{\mathbf{V}}_2 \otimes \widehat{\mathbf{V}}_2]\right)^{-1}$$

$$= \left\{\mathcal{T}\left[\mathcal{I} - \mathcal{T}^{-1}\mathbb{E}[\widehat{\mathbf{V}}_2 \otimes \widehat{\mathbf{V}}_2]\right]\right\}^{-1}$$

$$= \left[\mathcal{I} - \mathcal{T}^{-1}\mathbb{E}[\widehat{\mathbf{V}}_2 \otimes \widehat{\mathbf{V}}_2]\right]^{-1}\mathcal{T}^{-1}$$

$$= \sum_{k=0}^{\infty}(\mathcal{T}^{-1}\mathbb{E}[\widehat{\mathbf{V}}_2 \otimes \widehat{\mathbf{V}}_2])^k \circ \mathcal{T}^{-1},$$

where the last inequality holds due to geometric series of linear operators.

$\square$

We now show that $(\mathcal{I} - \mathcal{B})^{-1}$ exists and is a PSD operator.

**Lemma F.7.** With the parameters in (4.2), for any PSD matrix $\mathbf{M}$, $(\mathcal{I} - \mathcal{B})^{-1} \circ \mathbf{M}$ exists and is a PSD matrix. Moreover, if we define $\mathbf{Q} := \mathcal{T}^{-1} \circ \mathbf{M}$, then we have

$$(\mathcal{I} - \mathcal{B})^{-1} \circ \mathbf{M} = \mathbf{Q} + \frac{\psi}{1 - \psi l} \left\langle \begin{bmatrix} \mathbf{0} & \mathbf{0} \\ \mathbf{0} & \mathbf{H} \end{bmatrix}, \mathbf{Q} \right\rangle \cdot \mathbf{U}.$$

*Proof.* With Lemma F.6, we have

$$(\mathcal{I} - \mathcal{B})^{-1} \circ \mathbf{M} = \sum_{k=0}^{\infty} (\mathcal{T}^{-1} \mathbb{E}[\widehat{\mathbf{V}}_2 \otimes \widehat{\mathbf{V}}_2])^k \circ \mathbf{Q}.$$

Note that by Lemma F.3

$$\mathbb{E}[\widehat{\mathbf{V}}_2 \otimes \widehat{\mathbf{V}}_2] \circ \mathbf{Q} \preceq \psi \left\langle \begin{bmatrix} \mathbf{0} & \mathbf{0} \\ \mathbf{0} & \mathbf{H} \end{bmatrix}, \mathbf{Q} \right\rangle \cdot \begin{bmatrix} \delta^2 & \delta q \\ \delta q & q^2 \end{bmatrix} \otimes \mathbf{H},$$

and by definition of $\mathbf{U}$ in (F.7), we have

$$\mathcal{T}^{-1} \mathbb{E}[\widehat{\mathbf{V}}_2 \otimes \widehat{\mathbf{V}}_2] \circ \mathbf{Q} \preceq \psi \left\langle \begin{bmatrix} \mathbf{0} & \mathbf{0} \\ \mathbf{0} & \mathbf{H} \end{bmatrix}, \mathbf{Q} \right\rangle \cdot \mathbf{U}.$$

Then, applying Lemma F.5 and the definition of $\mathbf{U}$ recursively, we have for all $k \geq 1$,

$$(\mathcal{T}^{-1} \mathbb{E}[\widehat{\mathbf{V}}_2 \otimes \widehat{\mathbf{V}}_2])^k \circ \mathbf{Q} \preceq \psi^k l^{k-1} \left\langle \begin{bmatrix} \mathbf{0} & \mathbf{0} \\ \mathbf{0} & \mathbf{H} \end{bmatrix}, \mathbf{Q} \right\rangle \cdot \mathbf{U}. \tag{F.12}$$

Summing (F.12), considering the special case of $k = 0$, we have

$$(\mathcal{I} - \mathcal{B})^{-1} \circ \mathbf{M} \preceq \mathbf{Q} + \sum_{k=1}^{\infty} \psi^k l^{k-1} \left\langle \begin{bmatrix} \mathbf{0} & \mathbf{0} \\ \mathbf{0} & \mathbf{H} \end{bmatrix}, \mathbf{Q} \right\rangle \cdot \mathbf{U} = \mathbf{Q} + \frac{\psi}{1 - \psi l} \left\langle \begin{bmatrix} \mathbf{0} & \mathbf{0} \\ \mathbf{0} & \mathbf{H} \end{bmatrix}, \mathbf{Q} \right\rangle \cdot \mathbf{U}.$$

Therefore, $(\mathcal{I} - \mathcal{B})^{-1}$ exists and is a PSD operator. $\qquad \square$

The following result shows that the inner product (F.5) is different by only a constant if all $\mathcal{B}$ operators are replaced with $\widetilde{\mathcal{B}}$.

**Lemma F.8.** For any PSD matrix $\mathbf{M} \in \mathbb{R}^{2d \times 2d}$, define the partial sum

$$\mathbf{R}_t = \sum_{k=0}^{t-1} \mathcal{B}^k \circ \mathbf{M}.$$

Then we have

$$\mathbf{R}_t \preceq \sum_{k=0}^{t-1} \widetilde{\mathcal{B}}^k \circ \mathbf{M} + \frac{\psi}{1 - \psi l} \left\langle \begin{bmatrix} \mathbf{0} & \mathbf{0} \\ \mathbf{0} & \mathbf{H} \end{bmatrix}, \sum_{k=0}^{t-1} \widetilde{\mathcal{B}}^k \circ \mathbf{M} \right\rangle \cdot \mathbf{U}$$

and

$$\left\langle \begin{bmatrix} \mathbf{0} & \mathbf{0} \\ \mathbf{0} & \mathbf{H} \end{bmatrix}, \mathbf{R}_t \right\rangle \leq r \left\langle \begin{bmatrix} \mathbf{0} & \mathbf{0} \\ \mathbf{0} & \mathbf{H} \end{bmatrix}, \sum_{j=0}^{t-1} \widetilde{\mathcal{B}}^j \circ \mathbf{M} \right\rangle,$$

where $r = (1 - \psi l)^{-1}$.

*Proof.* By definition of $\mathbf{R}_t$, we have

$$\begin{aligned} \mathbf{R}_t &= (\mathcal{I} - \mathcal{B})^{-1} (\mathcal{I} - \mathcal{B}^t) \circ \mathbf{M} \\ &\preceq (\mathcal{I} - \mathcal{B})^{-1} (\mathcal{I} - \widetilde{\mathcal{B}}^t) \circ \mathbf{M} \\ &= (\mathcal{I} - \mathcal{B})^{-1} (\mathcal{I} - \widetilde{\mathcal{B}}) \sum_{k=0}^{t-1} \widetilde{\mathcal{B}}^k \circ \mathbf{M}, \end{aligned} \tag{F.13}$$

where the inequality holds because $\widetilde{\mathcal{B}} \preceq \mathcal{B}$. Note that by definition of $\widetilde{\mathcal{B}}$, we have

$$\mathcal{I} - \widetilde{\mathcal{B}} = \mathcal{I} - (\mathbf{V}_1 + \mathbf{V}_2) \otimes (\mathbf{V}_1 + \mathbf{V}_2) \preceq \mathcal{I} - \mathbf{V}_1 \otimes \mathbf{V}_1 - \mathbf{V}_1 \otimes \mathbf{V}_2 - \mathbf{V}_2 \otimes \mathbf{V}_1 = \mathcal{T}, \tag{F.14}$$

where the inequality holds because $\mathbf{V}_2 \otimes \mathbf{V}_2$ is a PSD operator. $\mathbf{R}_t$ can thus be further bounded as

$$\mathbf{R}_t \preceq (\mathcal{I} - \mathcal{B})^{-1} \mathcal{T} \circ \left( \sum_{k=0}^{t-1} \widetilde{\mathcal{B}}^k \circ \mathbf{M} \right)$$

$$\preceq \sum_{k=0}^{t-1} \widetilde{\mathcal{B}}^k \circ \mathbf{M} + \frac{\psi}{1 - \psi l} \left\langle \begin{bmatrix} \mathbf{0} & \mathbf{0} \\ \mathbf{0} & \mathbf{H} \end{bmatrix}, \sum_{k=0}^{t-1} \widetilde{\mathcal{B}}^k \circ \mathbf{M} \right\rangle \cdot \mathbf{U},$$

where the first inequality holds due to (F.14), and the second inequality holds due to Lemma F.6. Therefore, taking inner product with $\begin{bmatrix} \mathbf{0} & \mathbf{0} \\ \mathbf{0} & \mathbf{H} \end{bmatrix}$, we have

$$\left\langle \begin{bmatrix} \mathbf{0} & \mathbf{0} \\ \mathbf{0} & \mathbf{H} \end{bmatrix}, \mathbf{R}_t \right\rangle$$

$$\leq \left\langle \begin{bmatrix} \mathbf{0} & \mathbf{0} \\ \mathbf{0} & \mathbf{H} \end{bmatrix}, \sum_{k=0}^{t-1} \widetilde{\mathcal{B}}^k \circ \mathbf{M} \right\rangle + \frac{\psi}{1 - \psi l} \left\langle \begin{bmatrix} \mathbf{0} & \mathbf{0} \\ \mathbf{0} & \mathbf{H} \end{bmatrix}, \sum_{k=0}^{t-1} \widetilde{\mathcal{B}}^k \circ \mathbf{M} \right\rangle \cdot \left\langle \begin{bmatrix} \mathbf{0} & \mathbf{0} \\ \mathbf{0} & \mathbf{H} \end{bmatrix}, \mathbf{U} \right\rangle$$

$$\leq \left\langle \begin{bmatrix} \mathbf{0} & \mathbf{0} \\ \mathbf{0} & \mathbf{H} \end{bmatrix}, \sum_{k=0}^{t-1} \widetilde{\mathcal{B}}^k \circ \mathbf{M} \right\rangle + \frac{\psi l}{1 - \psi l} \left\langle \begin{bmatrix} \mathbf{0} & \mathbf{0} \\ \mathbf{0} & \mathbf{H} \end{bmatrix}, \sum_{k=0}^{t-1} \widetilde{\mathcal{B}}^k \circ \mathbf{M} \right\rangle$$

$$= \frac{1}{1 - \psi l} \left\langle \begin{bmatrix} \mathbf{0} & \mathbf{0} \\ \mathbf{0} & \mathbf{H} \end{bmatrix}, \sum_{k=0}^{t-1} \widetilde{\mathcal{B}}^k \circ \mathbf{M} \right\rangle,$$

where the second inequality holds due to Lemma F.5. $\qquad \square$

## G    VARIANCE UPPER BOUND

### G.1    PROOF OF LEMMA D.3

In this subsection, we prove Lemma D.3. We need the following lemmas. The first lemma characterizes the recursive formula of $\mathbf{C}_t$:

**Lemma G.1** (Section E.2 of Jain et al. (2018)). Define

$$\widehat{\boldsymbol{\Sigma}} := \mathbb{E}[\boldsymbol{\zeta}_t \otimes \boldsymbol{\zeta}_t], \tag{G.1}$$

then the covariance matrix $\mathbf{C}_t$ satisfies

$$\mathbf{C}_t = \mathcal{B} \circ \mathbf{C}_{t-1} + \widehat{\boldsymbol{\Sigma}}.$$

Combining Lemma G.1 with Lemma K.3, we immediate know that $\mathbf{C}_t$ is an increasing sequence with

$$\mathbf{C}_t = \sum_{k=0}^{t-1} \mathcal{B}^k \circ \widehat{\boldsymbol{\Sigma}}. \tag{G.2}$$

The following lemmas provide upper bounds for $\mathbf{M}_1$ and $\mathbf{M}_2$, respectively:

**Lemma G.2.** With the choice of parameters as in (4.2), we have

$$\left\langle \begin{bmatrix} \mathbf{H} & \mathbf{0} \\ \mathbf{0} & \mathbf{0} \end{bmatrix}, \mathbf{M}_1 \right\rangle \leq \sigma^2 r \left[ \frac{9k^*}{N} + \frac{36N(q - c\delta)^2}{(1 - c)^2} \sum_{i > k^*} \lambda_i^2 \right].$$

**Lemma G.3.** With the choice of parameters as in (4.2), we have

$$\left\langle \begin{bmatrix} \mathbf{H} & \mathbf{0} \\ \mathbf{0} & \mathbf{0} \end{bmatrix}, \mathbf{M}_2 \right\rangle \leq \sigma^2 r \left[ \frac{18k^*}{N} + \frac{36s(q - c\delta)^2}{(1 - c)^2} \sum_{i > k^*} \lambda_i^2 \right].$$

We now prove Lemma D.3.

*Proof of Lemma D.3.* By Lemma D.2,

$$\text{Variance} \leq \frac{1}{2} \left\langle \begin{bmatrix} \mathbf{0} & \mathbf{0} \\ \mathbf{0} & \mathbf{0} \end{bmatrix}, \mathbf{M}_1 + \mathbf{M}_2 \right\rangle$$

$$\leq \frac{\sigma^2 r}{2}\left[\frac{9k^*}{N} + \frac{36N(q - c\delta)^2}{(1 - c)^2}\sum_{i>k^*}\lambda_i^2\right] + \frac{\sigma^2 r}{2}\left[\frac{18k^*}{N} + \frac{36s(q - c\delta)^2}{(1 - c)^2}\sum_{i>k^*}\lambda_i^2\right]$$

$$= \sigma^2 r\left[\frac{27k^*}{2N} + \frac{18(s + N)(q - c\delta)^2}{(1 - c)^2}\sum_{i>k^*}\lambda_i^2\right],$$

where the second inequality holds due to Lemma G.2 and Lemma G.3. $\qquad\square$

We remark that due to Lemma K.1, we have $\frac{q - c\delta}{1 - c} = \frac{\gamma + \delta}{2} \leq \gamma$. Additionally, the constants in this proof are smaller than those given in Theorem 4.1. Therefore, the variance bound in Theorem 4.1 can be fully covered by the result provided in this proof.

### G.2 Proof of Lemma G.3

We start with an upper bound for $\widehat{\Sigma}$:

**Lemma G.4.** Let $\widehat{\Sigma}$ be defined in (G.1). Then

$$\widehat{\Sigma} \preceq \sigma^2 \begin{bmatrix} \delta^2 & \delta q \\ \delta q & q^2 \end{bmatrix} \otimes \mathbf{H}.$$

*Proof.* By definition of $\widehat{\Sigma}$ in (G.1), we have

$$\widehat{\Sigma} = \mathbb{E}[\boldsymbol{\zeta}_t \otimes \boldsymbol{\zeta}_t] = \mathbb{E}\left[\begin{bmatrix} \delta^2 \cdot \epsilon_t^2 \mathbf{x}_t \mathbf{x}_t^\top & \delta q \cdot \epsilon_t^2 \mathbf{x}_t \mathbf{x}_t^\top \\ \delta q \cdot \epsilon_t^2 \mathbf{x}_t \mathbf{x}_t^\top & q^2 \cdot \epsilon_t^2 \mathbf{x}_t \mathbf{x}_t^\top \end{bmatrix}\right] = \begin{bmatrix} \delta^2 & \delta q \\ \delta q & q^2 \end{bmatrix} \otimes \boldsymbol{\Sigma}. \qquad\text{(G.3)}$$

By Assumption 3.3, we have $\sigma^2 = \|\mathbf{H}^{-1/2}\boldsymbol{\Sigma}\mathbf{H}^{-1/2}\|$, so $\mathbf{H}^{-1/2}\boldsymbol{\Sigma}\mathbf{H}^{-1/2} \preceq \sigma^2\mathbf{I}$, and

$$\boldsymbol{\Sigma} \preceq \sigma^2\mathbf{H}. \qquad\text{(G.4)}$$

Combining (G.3) with (G.4), we complete the proof. $\qquad\square$

We then provide an upper bound for the limiting matrix $\mathbf{C}_\infty$.

**Lemma G.5.** Let $\mathbf{C}_\infty$ be defined as

$$\mathbf{C}_\infty := (\mathcal{I} - \mathcal{B})^{-1} \circ \widehat{\Sigma} = \sum_{k=0}^{\infty} \mathcal{B}^k \circ \widehat{\Sigma}, \qquad\text{(G.5)}$$

Then

$$\left\langle \begin{bmatrix} \mathbf{0} & \mathbf{0} \\ \mathbf{0} & \mathbf{H} \end{bmatrix}, \mathbf{C}_\infty \right\rangle \leq \frac{\sigma^2 l}{1 - \psi l},$$

where $l$ is defined in (F.8).

*Proof.* By definition of $\mathbf{C}_\infty$, we have

$$\mathbf{C}_\infty = (\mathcal{I} - \mathcal{B})^{-1} \circ \widehat{\Sigma} \preceq \sigma^2 (\mathcal{I} - \mathcal{B})^{-1} \circ \left(\begin{bmatrix} \delta^2 & \delta q \\ \delta q & q^2 \end{bmatrix} \otimes \mathbf{H}\right)$$

$$\preceq \sigma^2 \left(\mathbf{U} + \frac{\psi}{1 - \psi l}\left\langle \begin{bmatrix} \mathbf{0} & \mathbf{0} \\ \mathbf{0} & \mathbf{H} \end{bmatrix}, \mathbf{U}\right\rangle \cdot \mathbf{U}\right)$$

$$\preceq \sigma^2 \left(\mathbf{U} + \frac{\psi l}{1 - \psi l} \cdot \mathbf{U}\right) = \frac{\sigma^2}{1 - \psi l} \cdot \mathbf{U},$$

where the first inequality holds due to Lemma F.6, and the second inequality holds due to Lemma F.5. Therefore, the inner product is bounded by

$$\left\langle \begin{bmatrix} \mathbf{0} & \mathbf{0} \\ \mathbf{0} & \mathbf{H} \end{bmatrix}, \mathbf{C}_\infty \right\rangle \leq \frac{\sigma^2}{1 - \psi l}\left\langle \begin{bmatrix} \mathbf{0} & \mathbf{0} \\ \mathbf{0} & \mathbf{H} \end{bmatrix}, \mathbf{U}\right\rangle \leq \frac{\sigma^2 l}{1 - \psi l},$$

where the second inequality holds due to Lemma F.5. $\qquad\square$

We now prove Lemma G.3. For the matrix $\mathbf{M}_2$, we have the following upper bound:

$$
\mathbf{M}_2 = \frac{1}{N^2} \sum_{t=1}^{N-1} \left[ \sum_{k=0}^{N-t-1} \mathbf{A}^k \right] \left[ (\mathcal{B} - \widetilde{\mathcal{B}}) \circ \mathbf{C}_{s+t-1} + \widehat{\mathbf{\Sigma}} \right] \left[ \sum_{k=0}^{N-t-1} \mathbf{A}^k \right]^\top
$$

$$
\preceq \frac{1}{N^2} \sum_{t=1}^{N-1} \left[ \sum_{k=0}^{N-t-1} \mathbf{A}^k \right] \left[ \left( \psi \left\langle \begin{bmatrix} \mathbf{0} & \mathbf{0} \\ \mathbf{0} & \mathbf{H} \end{bmatrix}, \mathbf{C}_{s+t-1} \right\rangle + \sigma^2 \right) \begin{bmatrix} \delta^2 & \delta q \\ \delta q & q^2 \end{bmatrix} \otimes \mathbf{H} \right] \left[ \sum_{k=0}^{N-t-1} \mathbf{A}^k \right]^\top
$$

$$
\preceq \frac{1}{N^2} \sum_{t=1}^{N-1} \left[ \sum_{k=0}^{N-t-1} \mathbf{A}^k \right] \left[ \left( \psi \left\langle \begin{bmatrix} \mathbf{0} & \mathbf{0} \\ \mathbf{0} & \mathbf{H} \end{bmatrix}, \mathbf{C}_\infty \right\rangle + \sigma^2 \right) \begin{bmatrix} \delta^2 & \delta q \\ \delta q & q^2 \end{bmatrix} \otimes \mathbf{H} \right] \left[ \sum_{k=0}^{N-t-1} \mathbf{A}^k \right]^\top
$$

$$
\preceq \frac{1}{N^2} \sum_{t=1}^{N-1} \left[ \sum_{k=0}^{N-t-1} \mathbf{A}^k \right] \left[ \left( \frac{\sigma^2 \psi l}{1 - \psi l} + \sigma^2 \right) \begin{bmatrix} \delta^2 & \delta q \\ \delta q & q^2 \end{bmatrix} \otimes \mathbf{H} \right] \left[ \sum_{k=0}^{N-t-1} \mathbf{A}^k \right]^\top
$$

$$
= \frac{\sigma^2 r}{N^2} \sum_{t=1}^{N-1} \left[ \sum_{k=0}^{N-t-1} \mathbf{A}^k \right] \left( \begin{bmatrix} \delta^2 & \delta q \\ \delta q & q^2 \end{bmatrix} \otimes \mathbf{H} \right) \left[ \sum_{k=0}^{N-t-1} \mathbf{A}^k \right]^\top, \tag{G.6}
$$

where the first equality holds due to Lemma G.1, the first inequality holds due to Lemma G.4 and Lemma F.3, the second inequality holds because $\mathbf{C}_t$ is increasing, and the third inequality holds due to Lemma G.5. As $\mathbf{A}$ is block diagonal and $\mathbf{H}$ is diagonal, plugging (G.6) into the inner product, we have

$$
\left\langle \begin{bmatrix} \mathbf{H} & \mathbf{0} \\ \mathbf{0} & \mathbf{0} \end{bmatrix}, \mathbf{M}_2 \right\rangle \leq \frac{\sigma^2 r}{N^2} \sum_{i=1}^{d} \lambda_i^2 \sum_{t=1}^{N-1} \left( \left[ \sum_{k=0}^{N-t-1} \mathbf{A}_i^k \right] \begin{bmatrix} \delta \\ q \end{bmatrix} \right)_1^2
$$

$$
= \frac{\sigma^2 r}{N^2} \sum_{t=0}^{N-1} \sum_{i=1}^{d} \lambda_i^2 \left( \left[ \sum_{k=0}^{t-1} \mathbf{A}_i^k \right] \begin{bmatrix} \delta \\ q \end{bmatrix} \right)_1^2
$$

$$
\leq \frac{\sigma^2 r}{N^2} \sum_{t=0}^{N-1} \left[ 9k^* + \frac{36N^2(q - c\delta)^2}{(1 - c)^2} \sum_{i > k^*} \lambda_i^2 \right]
$$

$$
= \sigma^2 r \left[ \frac{9k^*}{N} + \frac{36N(q - c\delta)^2}{(1 - c)^2} \sum_{i > k^*} \lambda_i^2 \right],
$$

where the second inequality holds due to Corollary K.7.

### G.3 PROOF OF LEMMA G.2

The following lemma provides an upper bound on $\mathbf{C}_t$ by its update rule.

**Lemma G.6.** For any $t > 0$, $\mathbf{C}_t$ can be upper bounded by

$$
\mathbf{C}_t \preceq \sigma^2 r \sum_{k=0}^{t-1} \widetilde{\mathcal{B}}^k \circ \left( \begin{bmatrix} \delta^2 & \delta q \\ \delta q & q^2 \end{bmatrix} \otimes \mathbf{H} \right).
$$

*Proof.* By the recursive formula (7.5), we have the following,

$$
\mathbf{C}_t = \mathcal{B} \circ \mathbf{C}_{t-1} + \widehat{\mathbf{\Sigma}} = \widetilde{\mathcal{B}} \circ \mathbf{C}_{t-1} + (\mathcal{B} - \widetilde{\mathcal{B}}) \circ \mathbf{C}_{t-1} + \widehat{\mathbf{\Sigma}}
$$

$$
\preceq \widetilde{\mathcal{B}} \circ \mathbf{C}_{t-1} + \psi \left\langle \begin{bmatrix} \mathbf{0} & \mathbf{0} \\ \mathbf{0} & \mathbf{H} \end{bmatrix}, \mathbf{C}_{t-1} \right\rangle \cdot \begin{bmatrix} \delta^2 & \delta q \\ \delta q & q^2 \end{bmatrix} \otimes \mathbf{H} + \sigma^2 \begin{bmatrix} \delta^2 & \delta q \\ \delta q & q^2 \end{bmatrix} \otimes \mathbf{H}
$$

$$
\preceq \widetilde{\mathcal{B}} \circ \mathbf{C}_{t-1} + \psi \left\langle \begin{bmatrix} \mathbf{0} & \mathbf{0} \\ \mathbf{0} & \mathbf{H} \end{bmatrix}, \mathbf{C}_\infty \right\rangle \cdot \begin{bmatrix} \delta^2 & \delta q \\ \delta q & q^2 \end{bmatrix} \otimes \mathbf{H} + \sigma^2 \begin{bmatrix} \delta^2 & \delta q \\ \delta q & q^2 \end{bmatrix} \otimes \mathbf{H}
$$

$$
\preceq \widetilde{\mathcal{B}} \circ \mathbf{C}_{t-1} + \frac{\sigma^2 \psi l}{1 - \psi l} \cdot \begin{bmatrix} \delta^2 & \delta q \\ \delta q & q^2 \end{bmatrix} \otimes \mathbf{H} + \sigma^2 \begin{bmatrix} \delta^2 & \delta q \\ \delta q & q^2 \end{bmatrix} \otimes \mathbf{H}
$$

$$
= \widetilde{\mathcal{B}} \circ \mathbf{C}_{t-1} + \sigma^2 r \begin{bmatrix} \delta^2 & \delta q \\ \delta q & q^2 \end{bmatrix} \otimes \mathbf{H}, \tag{G.7}
$$

where the first inequality holds due to Lemma F.3 and Lemma G.4, the second inequality holds due to holds because $\mathbf{C}_t$ is increasing, and the last inequality holds due to Lemma G.5. Applying (G.7) recursively, we have for all $t > 0$,

$$\mathbf{C}_t \preceq \sigma^2 r \sum_{k=0}^{t-1} \widetilde{\mathcal{B}}^k \circ \begin{bmatrix} \delta^2 & \delta q \\ \delta q & q^2 \end{bmatrix} \otimes \mathbf{H}.$$

$\square$

We are now ready to prove Lemma G.2. With Lemma G.6, we have

$$\mathbf{M}_1 = \frac{1}{N^2} \left[ \sum_{k=0}^{N-1} \mathbf{A}^k \right] \mathbf{C}_s \left[ \sum_{k=0}^{N-1} \mathbf{A}^k \right]^\top$$

$$\preceq \frac{\sigma^2 r}{N^2} \sum_{j=0}^{s-1} \left[ \sum_{k=0}^{N-1} \mathbf{A}^k \right] \left[ \widetilde{\mathcal{B}}^j \circ \begin{bmatrix} \delta^2 & \delta q \\ \delta q & q^2 \end{bmatrix} \otimes \mathbf{H} \right] \left[ \sum_{k=0}^{N-1} \mathbf{A}^k \right]^\top$$

$$= \frac{\sigma^2 r}{N^2} \sum_{j=0}^{s-1} \left[ \sum_{k=0}^{N-1} \mathbf{A}^{j+k} \right] \left( \begin{bmatrix} \delta^2 & \delta q \\ \delta q & q^2 \end{bmatrix} \otimes \mathbf{H} \right) \left[ \sum_{k=0}^{N-1} \mathbf{A}^{j+k} \right]^\top. \quad (G.8)$$

As $\mathbf{A}$ is block-diagonal and $\mathbf{H}$ is diagonal, plugging (G.8) into the inner product, we have

$$\left\langle \begin{bmatrix} \mathbf{H} & \mathbf{0} \\ \mathbf{0} & \mathbf{0} \end{bmatrix}, \mathbf{M}_1 \right\rangle \leq \frac{\sigma^2 r}{N^2} \sum_{i=1}^{d} \lambda_i^2 \sum_{j=0}^{s-1} \left( \sum_{k=0}^{N-1} \mathbf{A}_i^{j+k} \begin{bmatrix} \delta \\ q \end{bmatrix} \right)_1^2 \quad (G.9)$$

$$\leq \frac{\sigma^2 r}{N^2} \left[ 18 N k^* + \frac{36 s N^2 (q - c\delta)^2}{(1-c)^2} \sum_{i > k^*} \lambda_i^2 \right]$$

$$= \sigma^2 r \left[ \frac{18 k^*}{N} + \frac{36 s (q - c\delta)^2}{(1-c)^2} \sum_{i > k^*} \lambda_i^2 \right],$$

where the second inequality holds due to Corollary K.6.

# H  BIAS UPPER BOUND

## H.1  PROOF OF LEMMA D.4

In this subsection, we prove Lemma D.4. We first need the following lemma for $\mathbf{B}_t$:

**Lemma H.1.** With $\mathbf{B}_t$ as defined in (7.4), we have

$$\mathbf{B}_t = \mathcal{B} \circ \mathbf{B}_{t-1},$$

and

$$\mathbf{B}_t \preceq \widetilde{\mathcal{B}}^t \circ \mathbf{B}_0 + \psi \sum_{k=0}^{t-1} \left\langle \begin{bmatrix} \mathbf{0} & \mathbf{0} \\ \mathbf{0} & \mathbf{H} \end{bmatrix}, \mathbf{B}_{t-1-k} \right\rangle \cdot \widetilde{\mathcal{B}}^k \circ \begin{bmatrix} \delta^2 & \delta q \\ \delta q & q^2 \end{bmatrix} \otimes \mathbf{H}.$$

We also have the following lemma for the partial sum of $\mathbf{B}_t$:

**Lemma H.2.** Let $\mathbf{B}_t$ defined in (7.4). Then we have

$$\sum_{k=0}^{t-1} \left\langle \begin{bmatrix} \mathbf{0} & \mathbf{0} \\ \mathbf{0} & \mathbf{H} \end{bmatrix}, \mathbf{B}_k \right\rangle \leq r \left[ \frac{14}{\delta} \|\mathbf{w}_0 - \mathbf{w}^*\|_{\mathbf{I}_{0:\hat{k}}}^2 + \frac{10}{1-c} \|\mathbf{w}_0 - \mathbf{w}^*\|_{\mathbf{H}_{\hat{k}:k^\dagger}}^2 \right.$$

$$\left. + \frac{1-c}{q-c\delta} \|\mathbf{w}_0 - \mathbf{w}^*\|_{\mathbf{I}_{k^\dagger:k^*}}^2 + 4t \|\mathbf{w}_0 - \mathbf{w}^*\|_{\mathbf{H}_{k^*:\infty}}^2 \right].$$

We are now ready prove Lemma D.4.

*Proof of Lemma D.4.* By Lemma D.2, it suffices to bound the inner produce of $\begin{bmatrix} \mathbf{H} & \mathbf{0} \\ \mathbf{0} & \mathbf{0} \end{bmatrix}$ with $\mathbf{M}_3$ and $\mathbf{M}_4$ separately. For $\mathbf{M}_3$, by Lemma H.1, we have

$$
\begin{aligned}
\mathbf{M}_3 &= \frac{1}{N^2} \left[ \sum_{k=0}^{N-1} \mathbf{A}^k \right] \mathbf{B}_s \left[ \sum_{k=0}^{N-1} \mathbf{A}^k \right]^\top \\
&\preceq \frac{1}{N^2} \left[ \sum_{k=0}^{N-1} \mathbf{A}^{k+s} \right] \mathbf{B}_0 \left[ \sum_{k=0}^{N-1} \mathbf{A}^{k+s} \right]^\top \\
&\quad + \frac{\psi}{N^2} \sum_{t=0}^{s-1} \left\langle \begin{bmatrix} \mathbf{0} & \mathbf{0} \\ \mathbf{0} & \mathbf{H} \end{bmatrix}, \mathbf{B}_{s-1-t} \right\rangle \left[ \sum_{k=0}^{N-1} \mathbf{A}^{k+t} \right] \left( \begin{bmatrix} \delta^2 & \delta q \\ \delta q & q^2 \end{bmatrix} \otimes \mathbf{H} \right) \left[ \sum_{k=0}^{N-1} \mathbf{A}^{k+t} \right]^\top . \quad \text{(H.1)}
\end{aligned}
$$

We also note that

$$
\mathbf{B}_0 = \begin{bmatrix} (\mathbf{w}_0 - \mathbf{w}^*)(\mathbf{w}_0 - \mathbf{w}^*)^\top & (\mathbf{w}_0 - \mathbf{w}^*)(\mathbf{w}_0 - \mathbf{w}^*)^\top \\ (\mathbf{w}_0 - \mathbf{w}^*)(\mathbf{w}_0 - \mathbf{w}^*)^\top & (\mathbf{w}_0 - \mathbf{w}^*)(\mathbf{w}_0 - \mathbf{w}^*)^\top \end{bmatrix} = \begin{bmatrix} 1 & 1 \\ 1 & 1 \end{bmatrix} \otimes [(\mathbf{w}_0 - \mathbf{w}^*)(\mathbf{w}_0 - \mathbf{w}^*)^\top]. \tag{H.2}
$$

$\mathbf{H}$ is diagonal and $\mathbf{A}$ is block diagonal, so plugging (H.1) and (H.2) into the inner product, we have

$$
\begin{aligned}
\left\langle \begin{bmatrix} \mathbf{H} & \mathbf{0} \\ \mathbf{0} & \mathbf{0} \end{bmatrix}, \mathbf{M}_3 \right\rangle &\leq \frac{1}{N^2} \sum_{i=1}^d \lambda_i w_i^2 \left( \sum_{k=0}^{N-1} \mathbf{A}_i^{k+s} \begin{bmatrix} 1 \\ 1 \end{bmatrix} \right)_1^2 \\
&\quad + \underbrace{\frac{\psi}{N^2} \sum_{t=0}^{s-1} \left\langle \begin{bmatrix} \mathbf{0} & \mathbf{0} \\ \mathbf{0} & \mathbf{H} \end{bmatrix}, \mathbf{B}_{s-1-t} \right\rangle \cdot \sum_{i=1}^d \lambda_i^2 \left( \sum_{k=0}^{N-1} \mathbf{A}_i^{k+t} \begin{bmatrix} \delta \\ q \end{bmatrix} \right)_1^2}_{\text{K}}. \tag{H.3}
\end{aligned}
$$

By Corollary K.9, we have

$$
\begin{aligned}
\text{Effective Bias} &:= \frac{1}{2N^2} \sum_{i=1}^d \lambda_i w_i^2 \left( \sum_{k=0}^{N-1} \mathbf{A}_i^{k+s} \begin{bmatrix} 1 \\ 1 \end{bmatrix} \right)_1^2 \\
&\leq \frac{8(c\delta/q)^{2s}}{N^2 \delta^2} \|\mathbf{w}_0 - \mathbf{w}^*\|_{\mathbf{H}_{0:k\ddagger}^{-1}}^2 + \frac{4s^2}{N^2} c^s \|(\mathbf{I} - \delta \mathbf{H})^{s/2} (\mathbf{w}_0 - \mathbf{w}^*)\|_{\mathbf{H}_{k\ddagger:k\dagger}}^2 \\
&\quad + \frac{16c^s}{N^2 \delta^2} \|(\mathbf{I} - \delta \mathbf{H})^{s/2} (\mathbf{w}_0 - \mathbf{w}^*)\|_{\mathbf{H}_{k\dagger:\hat{k}}^{-1}}^2 + \frac{100c^s}{N^2(1-c)^2} \|(\mathbf{I} - \delta \mathbf{H})^s (\mathbf{w}_0 - \mathbf{w}^*)\|_{\mathbf{H}_{k\ddagger:\hat{k}}}^2 \\
&\quad + \frac{9(1-c)^2}{2N^2(q-c\delta)^2} \left\| \left( \mathbf{I} - \frac{q-c\delta}{1-c} \mathbf{H} \right)^s (\mathbf{w}_0 - \mathbf{w}^*) \right\|_{\mathbf{H}_{k\dagger:k^*}^{-1}}^2 \\
&\quad + 18 \left\| \left( \mathbf{I} - \frac{q-c\delta}{1-c} \mathbf{H} \right)^s (\mathbf{w}_0 - \mathbf{w}^*) \right\|_{\mathbf{H}_{k^*:\infty}}^2 .
\end{aligned}
$$

K can be bounded as

$$
\begin{aligned}
\text{K} &= \frac{\psi}{N^2} \sum_{t=0}^{s-1} \left\langle \begin{bmatrix} \mathbf{0} & \mathbf{0} \\ \mathbf{0} & \mathbf{H} \end{bmatrix}, \mathbf{B}_{s-1-t} \right\rangle \cdot \sum_{i=1}^d \lambda_i^2 \left( \sum_{k=0}^{N-1} \mathbf{A}_i^{k+t} \begin{bmatrix} \delta \\ q \end{bmatrix} \right)_1^2 \\
&\leq \frac{\psi}{N^2} \left[ 9k^* + \frac{36(q-c\delta)^2 N^2}{(1-c)^2} \sum_{i>k^*} \lambda_i^2 \right] \sum_{t=0}^{s-1} \left\langle \begin{bmatrix} \mathbf{0} & \mathbf{0} \\ \mathbf{0} & \mathbf{H} \end{bmatrix}, \mathbf{B}_{s-1-t} \right\rangle \\
&= \frac{\psi}{N^2} \left[ 9k^* + \frac{36(q-c\delta)^2 N^2}{(1-c)^2} \sum_{i>k^*} \lambda_i^2 \right] \sum_{t=0}^{s-1} \left\langle \begin{bmatrix} \mathbf{0} & \mathbf{0} \\ \mathbf{0} & \mathbf{H} \end{bmatrix}, \mathbf{B}_t \right\rangle \\
&\leq \frac{\psi r}{N} \left[ \frac{9k^*}{N} + \frac{36N(q-c\delta)^2}{(1-c)^2} \sum_{i>k^*} \lambda_i^2 \right] \cdot \left[ \frac{14}{\delta} \|\mathbf{w}_0 - \mathbf{w}^*\|_{\mathbf{I}_{0:\hat{k}}}^2 \right.
\end{aligned}
$$

$$+ \frac{10}{1-c}\|\mathbf{w}_0 - \mathbf{w}^*\|^2_{\mathbf{H}_{\hat{k}:k^\dagger}} + \frac{1-c}{q-c\delta}\|\mathbf{w}_0 - \mathbf{w}^*\|^2_{\mathbf{I}_{k^\dagger:k^*}} + 4s\|\mathbf{w}_0 - \mathbf{w}^*\|^2_{\mathbf{H}_{k^*:\infty}}\Bigg], \qquad \text{(H.4)}$$

where the first inequality holds due to Corollary K.6, and the second inequality holds due to Lemma H.2.

For $\mathbf{M}_4$, we have

$$\mathbf{M}_4 = \frac{1}{N^2}\sum_{t=1}^{N-1}\left[\sum_{k=0}^{N-t-1}\mathbf{A}^k\right]\left((\mathcal{B} - \widetilde{\mathcal{B}}) \circ \mathbf{B}_{s+t-1}\right)\left[\sum_{k=0}^{N-t-1}\mathbf{A}^k\right]^\top$$

$$\preceq \frac{\psi}{N^2}\sum_{t=1}^{N-1}\left\langle\begin{bmatrix}\mathbf{0} & \mathbf{0}\\ \mathbf{0} & \mathbf{H}\end{bmatrix}, \mathbf{B}_{s+t-1}\right\rangle\left[\sum_{k=0}^{N-t-1}\mathbf{A}^k\right]\left(\begin{bmatrix}\delta^2 & \delta q\\ \delta q & q^2\end{bmatrix}\otimes\mathbf{H}\right)\left[\sum_{k=0}^{N-t-1}\mathbf{A}^k\right]^\top, \qquad \text{(H.5)}$$

where the first equality holds beause $\mathcal{B} \circ \mathbf{B}_{s+t-1} = \mathbf{B}_{s+t}$, and the inequality holds due to Lemma F.3. $\mathbf{H}$ is diagonal and $\mathbf{A}$ is block-diagonal, so plugging (H.5) into the inner product, we have

$$\left\langle\begin{bmatrix}\mathbf{H} & \mathbf{0}\\ \mathbf{0} & \mathbf{0}\end{bmatrix}, \mathbf{M}_4\right\rangle \leq \frac{\psi}{N^2}\sum_{t=1}^{N-1}\left\langle\begin{bmatrix}\mathbf{0} & \mathbf{0}\\ \mathbf{0} & \mathbf{H}\end{bmatrix}, \mathbf{B}_{s+t-1}\right\rangle \cdot \sum_{i=1}^{d}\lambda_i^2\left(\sum_{k=0}^{N-t-1}\mathbf{A}_i^k\begin{bmatrix}\delta\\ q\end{bmatrix}\right)_1^2$$

$$= \frac{\psi}{N^2}\sum_{t=0}^{N-1}\left\langle\begin{bmatrix}\mathbf{0} & \mathbf{0}\\ \mathbf{0} & \mathbf{H}\end{bmatrix}, \mathbf{B}_{s+N-t-1}\right\rangle\sum_{i=1}^{d}\lambda_i^2\left(\sum_{k=0}^{t-1}\mathbf{A}_i^k\begin{bmatrix}\delta\\ q\end{bmatrix}\right)_1^2$$

$$\leq \frac{\psi}{N}\left[\frac{9k^*}{N} + \frac{36N(q-c\delta)^2}{(1-c)^2}\sum_{i>k^*}\lambda_i^2\right]\sum_{t=0}^{N-1}\left\langle\begin{bmatrix}\mathbf{0} & \mathbf{0}\\ \mathbf{0} & \mathbf{H}\end{bmatrix}, \mathbf{B}_{s+N-t-1}\right\rangle$$

$$= \frac{\psi}{N}\left[\frac{9k^*}{N} + \frac{36N(q-c\delta)^2}{(1-c)^2}\sum_{i>k^*}\lambda_i^2\right]\sum_{t=0}^{N-1}\left\langle\begin{bmatrix}\mathbf{0} & \mathbf{0}\\ \mathbf{0} & \mathbf{H}\end{bmatrix}, \mathbf{B}_{s+t}\right\rangle, \qquad \text{(H.6)}$$

where the second inequality holds due to Corollary K.7. Note that

$$\sum_{t=0}^{N-1}\left\langle\begin{bmatrix}\mathbf{0} & \mathbf{0}\\ \mathbf{0} & \mathbf{H}\end{bmatrix}, \mathbf{B}_{s+t}\right\rangle \leq \sum_{t=0}^{s+N-1}\left\langle\begin{bmatrix}\mathbf{0} & \mathbf{0}\\ \mathbf{0} & \mathbf{H}\end{bmatrix}, \mathbf{B}_t\right\rangle$$

$$\leq r\left[\frac{14}{\delta}\|\mathbf{w}_0 - \mathbf{w}^*\|^2_{\mathbf{I}_{0:\hat{k}}} + \frac{10}{1-c}\|\mathbf{w}_0 - \mathbf{w}^*\|^2_{\mathbf{H}_{\hat{k}:k^\dagger}}\right.$$

$$\left. + \frac{1-c}{q-c\delta}\|\mathbf{w}_0 - \mathbf{w}^*\|^2_{\mathbf{I}_{k^\dagger:k^*}} + 4(s+N)\|\mathbf{w}_0 - \mathbf{w}^*\|^2_{\mathbf{H}_{k^*:\infty}}\right], \qquad \text{(H.7)}$$

where the first inequality holds because $\mathbf{B}_t \succeq 0$, and the second inequality holds due to Lemma H.2. Plugging (H.7) into (H.6), combining the result with (H.4), we have

$$\text{Bias} = \frac{1}{2}\left\langle\begin{bmatrix}\mathbf{H} & \mathbf{0}\\ \mathbf{0} & \mathbf{0}\end{bmatrix}, \mathbf{M}_3 + \mathbf{M}_4\right\rangle$$

$$\leq \text{Effective Bias} + \frac{\psi r}{2N}\left[\frac{9k^*}{N} + \frac{36N(q-c\delta)^2}{(1-c)^2}\sum_{i>k^*}\lambda_i^2\right] \cdot \left[\frac{14}{\delta}\|\mathbf{w}_0 - \mathbf{w}^*\|^2_{\mathbf{I}_{0:\hat{k}}}\right.$$

$$\left. + \frac{10}{1-c}\|\mathbf{w}_0 - \mathbf{w}^*\|^2_{\mathbf{H}_{\hat{k}:k^\dagger}} + \frac{1-c}{q-c\delta}\|\mathbf{w}_0 - \mathbf{w}^*\|^2_{\mathbf{I}_{k^\dagger:k^*}} + 4s\|\mathbf{w}_0 - \mathbf{w}^*\|^2_{\mathbf{H}_{k^*:\infty}}\right]$$

$$+ \frac{\psi r}{2N}\left[\frac{9k^*}{N} + \frac{36N(q-c\delta)^2}{(1-c)^2}\sum_{i>k^*}\lambda_i^2\right] \cdot \left[\frac{14}{\delta}\|\mathbf{w}_0 - \mathbf{w}^*\|^2_{\mathbf{I}_{0:\hat{k}}} + \frac{10}{1-c}\|\mathbf{w}_0 - \mathbf{w}^*\|^2_{\mathbf{H}_{\hat{k}:k^\dagger}}\right.$$

$$\left. + \frac{1-c}{q-c\delta}\|\mathbf{w}_0 - \mathbf{w}^*\|^2_{\mathbf{I}_{k^\dagger:k^*}} + 4(s+N)\|\mathbf{w}_0 - \mathbf{w}^*\|^2_{\mathbf{H}_{k^*:\infty}}\right]$$

$$\leq \text{Effective Bias} + \frac{\psi r}{N}\left[\frac{9k^*}{N} + \frac{36N(q-c\delta)^2}{(1-c)^2}\sum_{i>k^*}\lambda_i^2\right] \cdot \left[\frac{14}{\delta}\|\mathbf{w}_0 - \mathbf{w}^*\|_{\mathbf{I}_{0:\widehat{k}}}^2\right.$$

$$\left. + \frac{10}{1-c}\|\mathbf{w}_0 - \mathbf{w}^*\|_{\mathbf{H}_{\widehat{k}:k\dagger}}^2 + \frac{1-c}{q-c\delta}\|\mathbf{w}_0 - \mathbf{w}^*\|_{\mathbf{I}_{k\dagger:k^*}}^2 + 4(s+N)\|\mathbf{w}_0 - \mathbf{w}^*\|_{\mathbf{H}_{k^*:\infty}}^2\right],$$

where the second inequality holds because $4s\|\mathbf{w}_0 - \mathbf{w}^*\|_{\mathbf{H}_{k:\infty}}^2 \leq 4(s+N)\|\mathbf{w}_0 - \mathbf{w}^*\|_{\mathbf{H}_{k:\infty}}^2$. $\qquad\square$

We remark that due to Lemma K.1, we have $\frac{q-c\delta}{1-c} = \frac{\gamma+\delta}{2} \leq \gamma$. Additionally, the constants in this proof are smaller than those given in Theorem 4.1. Therefore, the bias bound in Theorem 4.1 can be fully covered by the result provided in this proof.

### H.2   PROOF OF LEMMA H.1

The recursive formula $\mathbf{B}_t = \mathcal{B} \circ \mathbf{B}_{t-1}$ is proven in Section B.2 of Jain et al. (2018). We further have

$$\mathbf{B}_t = \mathcal{B} \circ \mathbf{B}_{t-1} = \widetilde{\mathcal{B}} \circ \mathbf{B}_{t-1} + (\mathcal{B} - \widetilde{\mathcal{B}}) \circ \mathbf{B}_{t-1}$$

$$\preceq \widetilde{\mathcal{B}} \circ \mathbf{B}_{t-1} + \psi \left\langle \begin{bmatrix} \mathbf{0} & \mathbf{0} \\ \mathbf{0} & \mathbf{H} \end{bmatrix}, \mathbf{B}_{t-1} \right\rangle \cdot \begin{bmatrix} \delta^2 & \delta q \\ \delta q & q^2 \end{bmatrix} \otimes \mathbf{H}$$

$$\preceq \widetilde{\mathcal{B}}^t \circ \mathbf{B}_0 + \psi \sum_{k=0}^{t-1} \left\langle \begin{bmatrix} \mathbf{0} & \mathbf{0} \\ \mathbf{0} & \mathbf{H} \end{bmatrix}, \mathbf{B}_k \right\rangle \cdot \widetilde{\mathcal{B}}^{t-1-k} \circ \begin{bmatrix} \delta^2 & \delta q \\ \delta q & q^2 \end{bmatrix} \otimes \mathbf{H}$$

$$= \widetilde{\mathcal{B}}^t \circ \mathbf{B}_0 + \psi \sum_{k=0}^{t-1} \left\langle \begin{bmatrix} \mathbf{0} & \mathbf{0} \\ \mathbf{0} & \mathbf{H} \end{bmatrix}, \mathbf{B}_{t-1-k} \right\rangle \cdot \widetilde{\mathcal{B}}^{k} \circ \begin{bmatrix} \delta^2 & \delta q \\ \delta q & q^2 \end{bmatrix} \otimes \mathbf{H},$$

where the first inequality holds due to Lemma F.3, and the second inequality holds by recursively applying the bound.

### H.3   PROOF OF LEMMA H.2

Note that $\mathbf{B}_t = \mathcal{B}^t \circ \mathbf{B}_0$ by (7.4). By Lemma F.8, we have

$$\sum_{k=0}^{t-1} \left\langle \begin{bmatrix} \mathbf{0} & \mathbf{0} \\ \mathbf{0} & \mathbf{H} \end{bmatrix}, \mathbf{B}_k \right\rangle \leq r\sum_{k=0}^{t-1} \left\langle \begin{bmatrix} \mathbf{0} & \mathbf{0} \\ \mathbf{0} & \mathbf{H} \end{bmatrix}, \widetilde{\mathcal{B}}^k \circ \mathbf{B}_0 \right\rangle = r\sum_{k=0}^{t-1} \left\langle \begin{bmatrix} \mathbf{0} & \mathbf{0} \\ \mathbf{0} & \mathbf{H} \end{bmatrix}, \mathbf{A}^k \mathbf{B}_0 (\mathbf{A}^k)^\top \right\rangle. \quad \text{(H.8)}$$

$\mathbf{H}$ is a diagonal matrix, and $\mathbf{A}$ is a block-diagonal matrix with each block being $\mathbf{A}_i$, so (H.8) can be further bounded by

$$\sum_{k=0}^{t-1} \left\langle \begin{bmatrix} \mathbf{0} & \mathbf{0} \\ \mathbf{0} & \mathbf{H} \end{bmatrix}, \mathbf{B}_k \right\rangle \leq r\sum_i \lambda_i w_i^2 \sum_{k=0}^{t-1} \left(\mathbf{A}_i^k \begin{bmatrix} 1 \\ 1 \end{bmatrix}\right)_2^2$$

$$\leq r\left[\frac{14}{\delta}\|\mathbf{w}_0 - \mathbf{w}^*\|_{\mathbf{I}_{0:\widehat{k}}}^2 + \frac{10}{1-c}\|\mathbf{w}_0 - \mathbf{w}^*\|_{\mathbf{H}_{\widehat{k}:k\dagger}}^2\right.$$

$$\left. + \frac{1-c}{q-c\delta}\|\mathbf{w}_0 - \mathbf{w}^*\|_{\mathbf{I}_{k\dagger:k^*}}^2 + 4t\|\mathbf{w}_0 - \mathbf{w}^*\|_{\mathbf{H}_{k^*:\infty}}^2\right],$$

where the second inequality holds due to Corollary K.11.

## I   PROOF FOR THE CLASSICAL SETTING

In this section, we prove results for the case of finite dimensions. Before we prove the theorem, we first note that with the parameter choice in (4.4) and $\widetilde{\kappa} = d$,

$$1 - \psi l = 1 - \frac{\psi\delta\,\text{tr}(\mathbf{H})}{2} - \frac{1}{2} = \frac{1}{4},$$

so $r = 4$. We also note that with $\gamma\mu = 2\beta$, we have

$$\frac{(1-c)^2}{(\sqrt{q-c\delta} + \sqrt{c(q-\delta)})^2} = \frac{(1-c)^2}{(1+c)q - 2c\delta + 2\sqrt{c(q-\delta)(q-c\delta)}} \leq \frac{(1-c)^2}{(1+c)q - 2c\delta}$$

$$= \frac{(2(1-\alpha))^2}{2\alpha(\alpha\delta + (1-\alpha)\gamma) - 2(2\alpha-1)\delta} = \frac{2(1-\alpha)}{(1-\alpha)\delta + \alpha\gamma} \leq \frac{2(1-\alpha)}{\alpha\gamma} = \frac{2\beta}{\gamma} = \mu,$$

where the first equality holds because $2\sqrt{c(q-\delta)(q-c\delta)} \geq 0$, the second equality holds because $c = 2\alpha - 1$ and $q = \alpha\delta + (1-\alpha)\gamma$, and the second inequality holds because $(1-\alpha)\delta \geq 0$. That is to say, there is no eigenvalue in the region of $i > k^\dagger$.

The main idea of the proof is similar to that of Theorem 4.1. We decompose the excess risk into variance and bias, and then characterize $\mathbf{M}_1, \mathbf{M}_2, \mathbf{M}_3$ and $\mathbf{M}_4$. The following lemmas provide upper bounds for the inner product of $\begin{bmatrix} \mathbf{H} & \mathbf{0} \\ \mathbf{0} & \mathbf{0} \end{bmatrix}$ with $\mathbf{M}_1, \mathbf{M}_2, \mathbf{M}_3$ and $\mathbf{M}_4$.

**Lemma I.1** (Modified from Lemma G.2). With $\mathbf{M}_1$ defined in (D.2), we have

$$\left\langle \begin{bmatrix} \mathbf{H} & \mathbf{0} \\ \mathbf{0} & \mathbf{0} \end{bmatrix}, \mathbf{M}_1 \right\rangle \leq \frac{128\sigma^2 d}{N^2(1-c)}.$$

**Lemma I.2** (Modified from Lemma G.3). With $\mathbf{M}_2$ defined in (D.3), we have

$$\left\langle \begin{bmatrix} \mathbf{H} & \mathbf{0} \\ \mathbf{0} & \mathbf{0} \end{bmatrix}, \mathbf{M}_2 \right\rangle \leq \frac{9\sigma^2 rd}{N} = \frac{36\sigma^2 d}{N}.$$

**Lemma I.3.** With $\mathbf{M}_3$ defined in (D.4), we have

$$\left\langle \begin{bmatrix} \mathbf{H} & \mathbf{0} \\ \mathbf{0} & \mathbf{0} \end{bmatrix}, \mathbf{M}_3 \right\rangle \leq \frac{100}{N^2(1-c)^2} \exp\left(-\frac{(1-c)s}{2}\right) \cdot \|\mathbf{w}_0 - \mathbf{w}^*\|_\mathbf{H}^2 + \frac{504\psi d}{N^2(1-c)}\|\mathbf{w}_0 - \mathbf{w}^*\|_\mathbf{H}^2.$$

**Lemma I.4.** With $\mathbf{M}_4$ defined in (D.5), we have

$$\left\langle \begin{bmatrix} \mathbf{H} & \mathbf{0} \\ \mathbf{0} & \mathbf{0} \end{bmatrix}, \mathbf{M}_4 \right\rangle \leq \frac{504\psi d}{N^2(1-c)}\|\mathbf{w}_0 - \mathbf{w}^*\|_\mathbf{H}^2.$$

With the lemmas above, we can prove the upper bound of excess risk in the classical setting.

**Theorem I.5** (Restatement of Corollary 4.4). Under Assumptions 3.1, 3.2 and 3.3, with the parameter choice in (4.4), we have

$$\mathbb{E}[L(\overline{\mathbf{w}}_{s:s+N})] - L(\mathbf{w}^*) \leq \frac{100}{N^2(1-c)^2} \exp\left(-\frac{(1-c)s}{2}\right) \cdot \|\mathbf{w}_0 - \mathbf{w}^*\|_\mathbf{H}^2$$
$$+ \frac{36\sigma^2 d}{N} + \frac{128}{N^2(1-c)} + \frac{1008\psi d}{N^2(1-c)}\|\mathbf{w}_0 - \mathbf{w}^*\|_\mathbf{H}^2.$$

*Proof.* By Lemma D.1 and Lemma D.2, we have

$$\mathbb{E}[L(\overline{\mathbf{w}}_{s:s+N})] - L(\mathbf{w}^*) \leq \left\langle \begin{bmatrix} \mathbf{H} & \mathbf{0} \\ \mathbf{0} & \mathbf{0} \end{bmatrix}, \mathbf{M}_1 + \mathbf{M}_2 + \mathbf{M}_3 + \mathbf{M}_4 \right\rangle. \tag{I.1}$$

Substituting the results of Lemma I.1, Lemma I.2, Lemma I.3 and Lemma I.4 into (I.1), we get the desired result. $\square$

We remark that due to Lemma K.1, we have $1 - c \geq \beta$. Additionally, the constants in Theorem I.5 are smaller than those in Corollary 4.4. Therefore, Theorem I.5 can fully recover Corollary 4.4.

## I.1 VARIANCE UPPER BOUND

The proof for Lemma I.2 is straightforward given Lemma G.3 and the fact that there is no eigenvalue in the region of $i > k^\dagger$. Below we provide the proof for Lemma I.1.

*Proof of Lemma I.1.* According to (G.9) in the proof of Lemma G.2, we have

$$\left\langle \begin{bmatrix} \mathbf{H} & \mathbf{0} \\ \mathbf{0} & \mathbf{0} \end{bmatrix}, \mathbf{M}_1 \right\rangle \leq \frac{\sigma^2 r}{N^2} \sum_{i=1}^d \lambda_i^2 \sum_{j=0}^{s-1} \left( \sum_{k=0}^{N-1} \mathbf{A}_i^{j+k} \begin{bmatrix} \delta \\ q \end{bmatrix} \right)_1^2. \tag{I.2}$$

Similar to the proof of Corollary K.6, we have

(a) When $i \leq k^{\ddagger}$,

$$\sum_{j=0}^{s-1} \left( \sum_{k=0}^{N-1} \mathbf{A}_i^{j+k} \begin{bmatrix} \delta \\ q \end{bmatrix} \right)_1^2 \leq \frac{4}{(1-c)\lambda_i^2}.$$

(b) By (K.24) and (K.26), when $k^{\ddagger} < i \leq k^{\dagger}$,

$$\sum_{j=0}^{s-1} \left( \sum_{k=0}^{N-1} \mathbf{A}_i^{j+k} \begin{bmatrix} \delta \\ q \end{bmatrix} \right)_1^2 \leq \frac{32}{(1-c)\lambda_i^2}.$$

(I.2) can thus be bounded by

$$\left\langle \begin{bmatrix} \mathbf{H} & \mathbf{0} \\ \mathbf{0} & \mathbf{0} \end{bmatrix}, \mathbf{M}_1 \right\rangle \leq \frac{4\sigma^2}{N} \sum_{i=1}^{d} \lambda_i^2 \sum_{j=0}^{s-1} \left( \sum_{k=0}^{N-1} \mathbf{A}_i^{j+k} \begin{bmatrix} \delta \\ q \end{bmatrix} \right)_1^2$$

$$\leq \frac{4\sigma^2}{N^2} \left[ \sum_{i \leq k^{\ddagger}} \lambda_i^2 \cdot \frac{4}{(1-c)\lambda_i^2} + \sum_{i > k^{\ddagger}} \lambda_i^2 \cdot \frac{32}{(1-c)\lambda_i^2} \right]$$

$$= \frac{\sigma^2}{N^2(1-c)} \left[ 16k^{\ddagger} + 128(d - k^{\ddagger}) \right]$$

$$\leq \frac{128\sigma^2 d}{N^2(1-c)},$$

where first inequality holds because $r = 4$ and due to (I.2), and the last inequality holds because the coefficient $16 < 128$. $\qquad \square$

## I.2 BIAS UPPER BOUND

We first provide a list of lemmas modified by considering only eigenvalues $\lambda_i$ with $i \leq k^{\dagger}$:

**Lemma I.6** (Modified from Corollary K.6). Let $\mathbf{A}_i$ be defined in (E.1). Then for all $j \geq 0$,

$$\sum_{i=1}^{d} \lambda_i^2 \left( \sum_{k=0}^{t-1} \mathbf{A}_i^{j+k} \begin{bmatrix} \delta \\ q \end{bmatrix} \right)_1^2 \leq 9d.$$

Lemma I.6 follows directly from the corresponding results in the overparameterized regime, and we do not provide the proof here.

**Lemma I.7** (Modified from Corollary K.9). Let $\mathbf{A}_i$ be defined in (E.1). Then we have

$$\sum_{i=1}^{d} \lambda_i w_i^2 \left( \sum_{k=0}^{N-1} \mathbf{A}_i^{s+k} \begin{bmatrix} 1 \\ 1 \end{bmatrix} \right)_1^2 \leq \frac{100}{(1-c)^2} \exp\left( -\frac{(1-c)s}{2} \right) \cdot \|\mathbf{w}_0 - \mathbf{w}^*\|_{\mathbf{H}}^2.$$

*Proof.* By Lemma K.8,

(a) For all $i \leq k^{\ddagger}$, we have

$$\left( \sum_{k=0}^{N-1} \mathbf{A}_i^{s+k} \begin{bmatrix} 1 \\ 1 \end{bmatrix} \right)_1^2 \leq \frac{4}{\lambda_i^2} (c\delta/q)^{2s} \leq \frac{4}{\lambda_i^2} c^{s/2},$$

where the second inequality holds because $(c\delta/q)^2 \leq c^2 \leq \sqrt{c}$.

(b) For all $k^{\ddagger} < i \leq \widehat{k}$, we have

$$2s[c(1 - \delta\lambda_i)]^{s/2} + \frac{4}{\delta\lambda_i}[c(1 - \delta\lambda_i)]^{s/2}$$

$$\leq 2 \sum_{j=0}^{s-1} [c(1 - \delta\lambda_i)]^{(s+j)/4} + \frac{4}{\delta\lambda_i}[c(1 - \delta\lambda_i)]^{s/2}$$

$$= 2 \frac{[c(1 - \delta\lambda_i)]^{s/4} - [c(1 - \delta\lambda_i)]^{s/2}}{1 - [c(1 - \delta\lambda_i)]^{1/4}} + \frac{4}{\delta\lambda_i}[c(1 - \delta\lambda_i)]^{s/2}$$

$$\leq 2\frac{[c(1-\delta\lambda_i)]^{s/4}-[c(1-\delta\lambda_i)]^{s/2}}{\delta\lambda_i/4}+\frac{4}{\delta\lambda_i}[c(1-\delta\lambda_i)]^{s/2}$$

$$=\frac{8[c(1-\delta\lambda_i)]^{s/4}-4[c(1-\delta\lambda_i)]^{s/2}}{\delta\lambda_i}\leq\frac{8}{\delta\lambda_i}[c(1-\delta\lambda_i)]^{s/4}, \tag{I.3}$$

where the first inequality holds because $[c(1-\delta\lambda_i)]^{s/2}\leq[c(1-\delta\lambda_i)]^{(s+j)/4}$, the second inequality holds because $1-[c(1-\delta\lambda_i)]^{1/4}\geq 1-(1-\delta\lambda_i)^{1/4}\geq\delta\lambda_i/4$, and the last inequality holds because $[c(1-\delta\lambda_i)]^{s/2}\geq 0$. We thus have

$$\left(\sum_{k=0}^{N-1}\mathbf{A}_i^{s+k}\begin{bmatrix}1\\1\end{bmatrix}\right)_1^2\leq\left(2s[c(1-\delta\lambda_i)]^{s/2}+\frac{4}{\delta\lambda_i}[c(1-\delta\lambda_i)]^{s/2}\right)^2$$

$$\leq\frac{64}{\delta^2\lambda_i^2}[c(1-\delta\lambda_i)]^{s/2}\leq\frac{64}{\delta^2\lambda_i^2}\cdot c^{s/2},$$

where the first inequality holds due to Lemma K.8, the second inequality holds due to (I.3), and the last inequality holds because $c(1-\delta\lambda_i)\leq c$.

(c) For $\widehat{k}<i\leq k^\dagger$, we have

$$2s[c(1-\delta\lambda_i)]^{s/2}+\frac{10}{1-c}[c(1-\delta\lambda_i)]^{s/2}$$

$$\leq 2\sum_{j=0}^{s-1}[c(1-\delta\lambda_i)]^{(s+j)/4}+\frac{10}{1-c}[c(1-\delta\lambda_i)]^{s/2}$$

$$=2\frac{[c(1-\delta\lambda_i)]^{s/4}-[c(1-\delta\lambda_i)]^{s/2}}{1-[c(1-\delta\lambda_i)]^{1/4}}+\frac{10}{1-c}[c(1-\delta\lambda_i)]^{s/2}$$

$$\leq 2\frac{[c(1-\delta\lambda_i)]^{s/4}-[c(1-\delta\lambda_i)]^{s/2}}{(1-c)/4}+\frac{10}{1-c}[c(1-\delta\lambda_i)]^{s/2}$$

$$=\frac{8[c(1-\delta\lambda_i)]^{s/4}+2[c(1-\delta\lambda_i)]^{s/2}}{1-c}\leq\frac{10}{1-c}[c(1-\delta\lambda_i)]^{s/4}, \tag{I.4}$$

where the first inequality holds because $[c(1-\delta\lambda_i)]^{s/2}\leq[c(1-\delta\lambda_i)]^{(s+j)/4}$, the second inequality holds because $1-[c(1-\delta\lambda_i)]^{1/4}\geq 1-c^{1/4}\geq(1-c)/4$, and the last inequality holds because $[c(1-\delta\lambda_i)]^{s/2}\leq[c(1-\delta\lambda_i)]^{s/4}$. We thus have

$$\left(\sum_{k=0}^{N-1}\mathbf{A}_i^{s+k}\begin{bmatrix}1\\1\end{bmatrix}\right)_1^2\leq\left(2s[c(1-\delta\lambda_i)]^{s/2}+\frac{10}{1-c}[c(1-\delta\lambda_i)]^{s/2}\right)^2$$

$$\leq\frac{100}{(1-c)^2}[c(1-\delta\lambda_i)]^{s/2}\leq\frac{100}{(1-c)^2}\cdot c^{s/2},$$

where the first inequality holds due to Lemma K.8, the second inequality holds due to (I.4), and the last inequality holds because $c(1-\delta\lambda_i)\leq c$.

Concluding all the above,

$$\sum_{i=1}^d\lambda_i w_i^2\left(\sum_{k=0}^{N-1}\mathbf{A}_i^{s+k}\begin{bmatrix}1\\1\end{bmatrix}\right)_1^2$$

$$\leq\sum_{i\leq k^\ddagger}\lambda_i w_i^2\cdot\frac{4}{\delta^2\lambda_i^2}\cdot c^{s/2}+\sum_{k^\ddagger<i\leq\widehat{k}}\lambda_i w_i^2\cdot\frac{64}{\delta^2\lambda_i^2}\cdot c^{s/2}+\sum_{i>\widehat{k}}\lambda_i w_i^2\cdot\frac{100}{(1-c)^2}\cdot c^{s/2}$$

$$\leq\sum_{i\leq k^\ddagger}\lambda_i w_i^2\cdot\frac{4}{(1-c)^2}\cdot c^{s/2}+\sum_{k^\ddagger<i\leq\widehat{k}}\lambda_i w_i^2\cdot\frac{64}{(1-c)^2}\cdot c^{s/2}+\sum_{i>\widehat{k}}\lambda_i w_i^2\cdot\frac{100}{(1-c)^2}\cdot c^{s/2}$$

$$\leq\frac{100c^{s/2}}{(1-c)^2}\sum_{i=1}^d\lambda_i w_i^2=\frac{100c^{s/2}}{(1-c)^2}\cdot\|\mathbf{w}_0-\mathbf{w}^*\|_{\mathbf{H}}^2$$

$$\leq \frac{100}{(1-c)^2} \exp\left(-\frac{(1-c)s}{2}\right) \cdot \|\mathbf{w}_0 - \mathbf{w}^*\|_{\mathbf{H}}^2,$$

where the second inequality holds because $\delta\lambda_i \geq 1-c$ for $i \leq \widehat{k}$, the third inequality holds because the coefficients $4, 64, 100$ are bounded by $100$, and the last inequality holds because $c^{s/2} \leq \exp(-(1-c)s/2)$. $\square$

**Lemma I.8.** With $\mathbf{B}_t$ defined in (7.4), we have

$$\sum_{k=0}^{t-1} \left\langle \begin{bmatrix} \mathbf{0} & \mathbf{0} \\ \mathbf{0} & \mathbf{H} \end{bmatrix}, \mathbf{B}_k \right\rangle \leq \frac{56}{1-c}\|\mathbf{w}_0 - \mathbf{w}^*\|_{\mathbf{H}}^2.$$

*Proof.* By Lemma H.2, taking $r = 4$, we have

$$\sum_{k=0}^{t-1} \left\langle \begin{bmatrix} \mathbf{0} & \mathbf{0} \\ \mathbf{0} & \mathbf{H} \end{bmatrix}, \mathbf{B}_k \right\rangle \leq \frac{56}{\delta\lambda_i} \sum_{i \leq k^\ddagger} \lambda_i w_i^2 + \frac{40}{1-c} \sum_{k^\ddagger < i \leq k^\dagger} \lambda_i w_i^2$$

$$\leq \frac{56}{1-c} \sum_{i \leq k^\ddagger} \cdot \lambda_i w_i^2 + \frac{40}{1-c} \sum_{k^\ddagger < i \leq k^\dagger} \lambda_i w_i^2$$

$$\leq \frac{56}{1-c}\|\mathbf{w}_0 - \mathbf{w}^*\|_{\mathbf{H}}^2,$$

where the first inequality holds because $\delta\lambda_i \geq 1-c$ for $i \leq k^\ddagger$, and the second inequality holds because the coefficients $40, 50$ can be bounded by $56$. $\square$

We are now ready to bound the inner product of $\begin{bmatrix} \mathbf{H} & \mathbf{0} \\ \mathbf{0} & \mathbf{0} \end{bmatrix}$ with $\mathbf{M}_3$ and $\mathbf{M}_4$.

*Proof of Lemma I.3.* Similar to the proof of Lemma D.4, we have

$$\left\langle \begin{bmatrix} \mathbf{H} & \mathbf{0} \\ \mathbf{0} & \mathbf{0} \end{bmatrix}, \mathbf{M}_3 \right\rangle$$

$$\leq \frac{1}{N^2} \sum_{i=1}^{d} \lambda_i w_i^2 \left(\sum_{k=0}^{N-1} \mathbf{A}_i^{j+k} \begin{bmatrix} 1 \\ 1 \end{bmatrix}\right)_1^2$$

$$+ \frac{\psi}{N^2} \sum_{t=0}^{s-1} \left\langle \begin{bmatrix} \mathbf{0} & \mathbf{0} \\ \mathbf{0} & \mathbf{H} \end{bmatrix}, \mathbf{B}_{s-1-t} \right\rangle \sum_{i=1}^{d} \lambda_i^2 \left(\sum_{k=0}^{N-1} \mathbf{A}_i^{k+t} \begin{bmatrix} \delta \\ q \end{bmatrix}\right)_1^2$$

$$\leq \frac{100}{N^2(1-c)^2} \exp\left(-\frac{(1-c)s}{2}\right) \cdot \|\mathbf{w}_0 - \mathbf{w}^*\|_{\mathbf{H}}^2 + \frac{\psi}{N^2} \sum_{t=0}^{s-1} \left\langle \begin{bmatrix} \mathbf{0} & \mathbf{0} \\ \mathbf{0} & \mathbf{H} \end{bmatrix}, \mathbf{B}_{s-1-t} \right\rangle \cdot 9d$$

$$\leq \frac{100}{N^2(1-c)^2} \exp\left(-\frac{(1-c)s}{2}\right) \cdot \|\mathbf{w}_0 - \mathbf{w}^*\|_{\mathbf{H}}^2 + \frac{9\psi d}{N^2} \cdot \frac{56}{1-c}\|\mathbf{w}_0 - \mathbf{w}^*\|_{\mathbf{H}}^2$$

$$= \frac{100}{N^2(1-c)^2} \exp\left(-\frac{(1-c)s}{2}\right) \cdot \|\mathbf{w}_0 - \mathbf{w}^*\|_{\mathbf{H}}^2 + \frac{504\psi d}{N^2(1-c)}\|\mathbf{w}_0 - \mathbf{w}^*\|_{\mathbf{H}}^2,$$

where the second inequality holds due to Lemma I.7 and Lemma I.6, and the third inequality holds due to Lemma I.8. $\square$

*Proof of Lemma I.4.* Similar to the proof of Lemma D.4, we have

$$\left\langle \begin{bmatrix} \mathbf{H} & \mathbf{0} \\ \mathbf{0} & \mathbf{0} \end{bmatrix}, \mathbf{M}_4 \right\rangle \leq \frac{\psi}{N^2} \sum_{t=0}^{N-1} \left\langle \begin{bmatrix} \mathbf{0} & \mathbf{0} \\ \mathbf{0} & \mathbf{H} \end{bmatrix}, \mathbf{B}_{s+N-t-1} \right\rangle \sum_{i=1}^{d} \lambda_i^2 \left(\sum_{k=0}^{t-1} \mathbf{A}_i^k \begin{bmatrix} \delta \\ q \end{bmatrix}\right)_1^2$$

$$\leq \frac{9d\psi}{N^2} \sum_{t=0}^{N-1} \left\langle \begin{bmatrix} \mathbf{0} & \mathbf{0} \\ \mathbf{0} & \mathbf{H} \end{bmatrix}, \mathbf{B}_{s+N-t-1} \right\rangle$$

$$\leq \frac{9\psi d}{N^2} \cdot \sum_{t=0}^{s+N-1} \left\langle \begin{bmatrix} \mathbf{0} & \mathbf{0} \\ \mathbf{0} & \mathbf{H} \end{bmatrix}, \mathbf{B}_t \right\rangle$$

$$\leq \frac{9\psi d}{N^2} \cdot \frac{56}{1-c} \|\mathbf{w}_0 - \mathbf{w}^*\|_{\mathbf{H}}^2 = \frac{504\psi d}{N^2(1-c)} \|\mathbf{w}_0 - \mathbf{w}^*\|_{\mathbf{H}}^2,$$

where the second inequality holds due to Lemma I.6, the second inequality holds because $\mathbf{B}_t \succeq 0$, and the last inequality holds due to Lemma I.8. $\qquad\square$

## J  PROOF FOR THE ONE-HOT DISTRIBUTION SETTING

The choice of parameters is as follows:

$$\gamma \in (0,1), \ \delta \in (0,\gamma], \ \beta \in (0,1), \ \alpha = \frac{1}{1+\beta}. \tag{J.1}$$

We now present the excess risk bound:

**Theorem J.1.** Under Assumptions 3.1, 3.3 and 3.2, with the parameter choice of (J.1), assuming $N(1-c) \geq 2$, we have the following upper bound for the excess risk:

$$\mathbb{E}[L(\bar{\mathbf{w}}_{s:s+N})] - L(\mathbf{w}^*) \leq 2 \cdot \text{EffectiveVar} + 2 \cdot \text{EffectiveBias},$$

where effective variance is bounded by

$$\text{EffectiveVar} \leq \sigma^2 r \left[ \frac{27k^*}{2N} + 18(s+N)\gamma^2 \sum_{i>k^*} \lambda_i^2 \right] + \frac{r}{N^2} \left[ \frac{126}{\delta} \|\mathbf{w}_0 - \mathbf{w}^*\|_{\mathbf{H}_{0:\hat{k}}^{-1}}^2 \right.$$
$$\left. + \frac{90}{1-c} \|\mathbf{w}_0 - \mathbf{w}^*\|_{\mathbf{I}_{\hat{k}:k^\dagger}}^2 + \frac{18}{\gamma} \|\mathbf{w}_0 - \mathbf{w}^*\|_{\mathbf{H}_{k^\dagger:k^*}^{-1}}^2 + 36\gamma^2 N^2 s \|\mathbf{w}_0 - \mathbf{w}^*\|_{\mathbf{H}_{k^*:\infty}^2}^2 \right],$$

and effective bias is bounded in the same way as Theorem 4.1.

The constant $r$ is formally defined as

$$r = \frac{1}{1 - \max_{1 \leq i \leq d}(\mathbf{U}_i)_{22}}, \tag{J.2}$$

Note that

$$(\mathbf{U}_i)_{22} \leq \frac{q - c\delta}{2(1-c)} \leq \frac{\gamma}{2},$$

so the upper bound of $r$ is given by

$$r \leq \frac{1}{1 - \gamma/2}.$$

The proof of Theorem J.1 depends on the following lemmas:

**Lemma J.2** (Modified from Lemma G.2). Let $r$ be defined in (J.2). Then we have

$$\left\langle \begin{bmatrix} \mathbf{H} & \mathbf{0} \\ \mathbf{0} & \mathbf{0} \end{bmatrix}, \mathbf{M}_1 \right\rangle \leq \sigma^2 r \left[ \frac{18k^*}{N} + \frac{36s(q-c\delta)^2}{(1-c)^2} \sum_{i>k^*} \lambda_i^2 \right].$$

**Lemma J.3** (Modified from Lemma G.3). Let $r$ be defined in (J.2). Then we have

$$\left\langle \begin{bmatrix} \mathbf{H} & \mathbf{0} \\ \mathbf{0} & \mathbf{0} \end{bmatrix}, \mathbf{M}_2 \right\rangle \leq \sigma^2 r \left[ \frac{9k^*}{N} + \frac{36N(q-c\delta)^2}{(1-c)^2} \sum_{i>k^*} \lambda_i^2 \right].$$

**Lemma J.4.** Let $r$ be defined in (J.2). Then we have

$$\left\langle \begin{bmatrix} \mathbf{H} & \mathbf{0} \\ \mathbf{0} & \mathbf{0} \end{bmatrix}, \mathbf{M}_3 \right\rangle \leq \frac{r}{N^2} \left[ \frac{126}{\delta} \|\mathbf{w}_0 - \mathbf{w}^*\|_{\mathbf{H}_{0:\hat{k}}^{-1}}^2 + \frac{90}{1-c} \|\mathbf{w}_0 - \mathbf{w}^*\|_{\mathbf{I}_{\hat{k}:k^\dagger}}^2 + \frac{9(1-c)}{q-c\delta} \|\mathbf{w}_0 - \mathbf{w}^*\|_{\mathbf{H}_{k^\dagger:k^*}^{-1}}^2 \right.$$
$$\left. + \frac{36(q-c\delta)^2 N^2 s}{(1-c)^2} \|\mathbf{w}_0 - \mathbf{w}^*\|_{\mathbf{H}_{k^*:\infty}^2}^2 \right] + \text{EffectiveBias},$$

where EffectiveBias is the same as one in Theorem 4.1.

**Lemma J.5.** Let $r$ be defined in (J.2). Then we have

$$\left\langle \begin{bmatrix} \mathbf{H} & \mathbf{0} \\ \mathbf{0} & \mathbf{0} \end{bmatrix}, \mathbf{M}_4 \right\rangle \leq \frac{r}{N^2} \left[ \frac{126}{\delta} \|\mathbf{w}_0 - \mathbf{w}^*\|_{\mathbf{H}_{0:\widehat{k}}^{-1}}^2 + \frac{90}{1-c} \|\mathbf{w}_0 - \mathbf{w}^*\|_{\widehat{\mathbf{I}}_{\widehat{k}:k^\dagger}}^2 + \frac{9(1-c)}{q-c\delta} \|\mathbf{w}_0 - \mathbf{w}^*\|_{\mathbf{H}_{k^\dagger:k^*}^{-1}}^2 \right. $$
$$\left. + \frac{36(q-c\delta)^2 N^2(s+N)}{(1-c)^2} \|\mathbf{w}_0 - \mathbf{w}^*\|_{\mathbf{H}_{k^*:\infty}^2}^2 \right].$$

*Proof of Theorem J.1.* Note that the excess risk is

$$\left\langle \begin{bmatrix} \mathbf{H} & \mathbf{0} \\ \mathbf{0} & \mathbf{0} \end{bmatrix}, \mathbf{M}_1 + \mathbf{M}_2 + \mathbf{M}_3 + \mathbf{M}_4 \right\rangle,$$

so the upper bound can be obtained by combining Lemmas J.2, J.3, J.4 and J.5. $\qquad\square$

**Notations.** In this section, for any matrix $\mathbf{M} \in \mathbb{R}^{2d \times 2d}$, denote

$$\mathbf{M} = \begin{bmatrix} \mathbf{M}_{11} & \mathbf{M}_{12} \\ \mathbf{M}_{21} & \mathbf{M}_{22} \end{bmatrix} \in \mathbb{R}^{2d \times 2d},$$

where $\mathbf{M}_{ij} \in \mathbb{R}^{d \times d}$.

## J.1 ANALYSIS OF FOURTH MOMENT

In this setting, for any matrix $\mathbf{M} \in \mathbb{R}^{2d \times 2d}$, we have

$$\mathbb{E}[\widehat{\mathbf{V}}_2 \times \widehat{\mathbf{V}}_2] \circ \mathbf{M} = \begin{bmatrix} \delta^2 & \delta q \\ \delta q & q^2 \end{bmatrix} \otimes (\mathbf{H} \odot \mathbf{M}_{22}) = \begin{bmatrix} \delta^2 & \delta q \\ \delta q & q^2 \end{bmatrix} \otimes \operatorname{diag}(\lambda_1(\mathbf{M}_{22})_{11}, \dots, \lambda_d(\mathbf{M}_{22})_{dd}).$$

**Lemma J.6** (Modified from Lemma F.7). For any PSD matrix $\mathbf{M} \in \mathbb{R}^{2d \times 2d}$ define $\mathbf{Q} := \mathcal{T}^{-1} \circ \mathbf{M}$. Then

$$(\mathcal{I} - \mathcal{B})^{-1} \circ \mathbf{M} = \mathbf{Q} + \operatorname{diag}\left( \frac{(\mathbf{Q}_{22})_{11}}{1 - (\mathbf{U}_1)_{22}} \mathbf{U}_1, \dots, \frac{(\mathbf{Q}_{22})_{dd}}{1 - (\mathbf{U}_d)_{22}} \mathbf{U}_d \right).$$

*Proof.* By Lemma F.6, we have

$$(\mathcal{I} - \mathcal{B})^{-1} \circ \mathbf{M} = \sum_{k=0}^{\infty} (\mathcal{T}^{-1} \mathbb{E}[\widehat{\mathbf{V}}_2 \otimes \widehat{\mathbf{V}}_2])^k \circ \mathbf{Q}.$$

Note that

$$\mathbb{E}[\widehat{\mathbf{V}}_2 \otimes \widehat{\mathbf{V}}_2] \circ \mathbf{Q} = \begin{bmatrix} \delta^2 & \delta q \\ \delta q & q^2 \end{bmatrix} \otimes \operatorname{diag}(\lambda_1(\mathbf{Q}_{22})_{11}, \dots, \lambda_d(\mathbf{Q}_{22})_{dd}),$$

so by definition of $\mathcal{T}$,

$$\mathcal{T}^{-1} \mathbb{E}[\widehat{\mathbf{V}}_2 \otimes \widehat{\mathbf{V}}_2] \circ \mathbf{Q} = \operatorname{diag}((\mathbf{Q}_{22})_{11} \mathbf{U}_1, \dots, (\mathbf{Q}_{22})_{dd} \mathbf{U}_d).$$

We can similarly prove that for all $k \geq 1$,

$$(\mathcal{T}^{-1} \mathbb{E}[\widehat{\mathbf{V}}_2 \otimes \widehat{\mathbf{V}}_2])^k \circ \mathbf{Q} = \operatorname{diag}((\mathbf{Q}_{22})_{11}(\mathbf{U}_1)_{22}^{k-1} \mathbf{U}_1, \dots, (\mathbf{Q}_{22})_{11}(\mathbf{U}_d)_{22}^{k-1} \mathbf{U}_d).$$

Summing the above, we have

$$(\mathcal{I} - \mathcal{B})^{-1} \circ \mathbf{M} = \mathbf{Q} + \operatorname{diag}\left( \frac{(\mathbf{Q}_{22})_{11}}{1 - (\mathbf{U}_1)_{22}} \mathbf{U}_1, \dots, \frac{(\mathbf{Q}_{22})_{dd}}{1 - (\mathbf{U}_d)_{22}} \mathbf{U}_d \right).$$

$\qquad\square$

**Lemma J.7** (Modified from Lemma F.8). For any PSD matrix $\mathbf{M} \in \mathbb{R}^{2d \times 2d}$, define the partial sum

$$\mathbf{R}_t = \sum_{k=0}^{t-1} \mathcal{B}^k \circ \mathbf{M}.$$

Then we have

$$\mathbf{R}_t \preceq \sum_{k=0}^{t-1} \widetilde{\mathcal{B}}^k \circ \mathbf{M} + \sum_{k=0}^{t-1} \operatorname{diag}\left( \frac{((\widetilde{\mathcal{B}}^k \circ \mathbf{M})_{22})_{11}}{1 - (\mathbf{U}_1)_{22}} \mathbf{U}_1, \dots, \frac{((\widetilde{\mathcal{B}}^k \circ \mathbf{M})_{22})_{dd}}{1 - (\mathbf{U}_d)_{22}} \mathbf{U}_d \right).$$

and

$$\mathbb{E}[\widehat{\mathbf{V}}_2 \otimes \widehat{\mathbf{V}}_2] \preceq \begin{bmatrix} \delta^2 & \delta q \\ \delta q & q^2 \end{bmatrix} \otimes \operatorname{diag}\left( \frac{\lambda_1((\widetilde{\mathcal{B}}^k \circ \mathbf{M})_{22})_{11}}{1 - (\mathbf{U}_1)_{22}}, \dots, \frac{\lambda_d((\widetilde{\mathcal{B}}^k \circ \mathbf{M})_{22})_{dd}}{1 - (\mathbf{U}_d)_{22}} \right).$$

*Proof.* Similar to the proof of Lemma F.8, we have

$$\sum_{k=0}^{t-1} \mathcal{B}^k \circ \mathbf{M} \preceq (\mathcal{I} - \mathcal{B})^{-1} \mathcal{T} \circ \left( \sum_{k=0}^{t-1} \widetilde{\mathcal{B}}^k \circ \mathbf{M} \right)$$

$$= \sum_{k=0}^{t-1} \widetilde{\mathcal{B}}^k \circ \mathbf{M} + \sum_{k=0}^{t-1} \mathrm{diag} \left( \frac{((\widetilde{\mathcal{B}}^k \circ \mathbf{M})_{22})_{11}}{1 - (\mathbf{U}_1)_{22}} \mathbf{U}_1, \dots, \frac{((\widetilde{\mathcal{B}}^k \circ \mathbf{M})_{22})_{dd}}{1 - (\mathbf{U}_d)_{22}} \mathbf{U}_d \right),$$

where the equality holds due to Lemma J.6. We thus have

$$\mathbb{E}[\widehat{\mathbf{V}}_2 \otimes \widehat{\mathbf{V}}_2] \circ \left( \sum_{k=0}^{t-1} \mathcal{B}^k \circ \mathbf{M} \right) \preceq \begin{bmatrix} \delta^2 & \delta q \\ \delta q & q^2 \end{bmatrix} \otimes \left[ \sum_{k=0}^{t-1} \mathrm{diag} \left( \lambda_1((\widetilde{\mathcal{B}}^k \circ \mathbf{M})_{22})_{11}, \dots, \lambda_d((\widetilde{\mathcal{B}}^k \circ \mathbf{M})_{22})_{dd} \right) \right.$$

$$+ \sum_{k=0}^{t-1} \mathrm{diag} \left( \frac{\lambda_1(\mathbf{U}_1)_{22}}{1 - (\mathbf{U}_1)_{22}} ((\widetilde{\mathcal{B}}^k \circ \mathbf{M})_{22})_{11}, \dots, \frac{\lambda_d(\mathbf{U}_d)_{22}}{1 - (\mathbf{U}_d)_{22}} ((\widetilde{\mathcal{B}}^k \circ \mathbf{M})_{22})_{dd} \right) \Bigg]$$

$$= \begin{bmatrix} \delta^2 & \delta q \\ \delta q & q^2 \end{bmatrix} \otimes \mathrm{diag} \left( \frac{\lambda_1((\widetilde{\mathcal{B}}^k \circ \mathbf{M})_{22})_{11}}{1 - (\mathbf{U}_1)_{22}}, \dots, \frac{\lambda_d((\widetilde{\mathcal{B}}^k \circ \mathbf{M})_{22})_{dd}}{1 - (\mathbf{U}_d)_{22}} \right).$$

$\square$

## J.2 VARIANCE UPPER BOUND

We now provide the proof of Lemma J.3.

*Proof of Lemma J.3.* Note that

$$\mathbf{C}_\infty = (\mathcal{I} - \mathcal{B})^{-1} \circ \widehat{\boldsymbol{\Sigma}} \preceq \sigma^2 (\mathcal{I} - \mathcal{B})^{-1} \circ \left( \begin{bmatrix} \delta^2 & \delta q \\ \delta q & q^2 \end{bmatrix} \otimes \mathbf{H} \right)$$

$$= \sigma^2 \left[ \mathbf{U} + \mathrm{diag} \left( \frac{(\mathbf{U}_1)_{22}}{1 - (\mathbf{U}_1)_{22}} \mathbf{U}_1, \dots, \frac{(\mathbf{U}_d)_{22}}{1 - (\mathbf{U}_d)_{22}} \mathbf{U}_d \right) \right]$$

$$= \sigma^2 \mathrm{diag} \left( \frac{\mathbf{U}_1}{1 - (\mathbf{U}_1)_{22}}, \dots, \frac{\mathbf{U}_d}{1 - (\mathbf{U}_d)_{22}} \right),$$

where the second equality holds due to Lemma J.6. We thus have

$$\mathbb{E}[\widehat{\mathbf{V}}_2 \otimes \widehat{\mathbf{V}}_2] \circ \mathbf{C}_\infty \preceq \sigma^2 \begin{bmatrix} \delta^2 & \delta q \\ \delta q & q^2 \end{bmatrix} \otimes \mathrm{diag} \left( \frac{\lambda_1(\mathbf{U}_1)_{22}}{1 - (\mathbf{U}_1)_{22}}, \dots, \frac{\lambda_d(\mathbf{U}_d)_{22}}{1 - (\mathbf{U}_d)_{22}} \right). \tag{J.3}$$

Therefore, $\mathbf{M}_2$ can be bounded by

$$\mathbf{M}_2 = \frac{1}{N^2} \sum_{t=1}^{N-1} \left[ \sum_{k=0}^{N-t-1} \mathbf{A}^k \right] \left[ (\mathcal{B} - \widetilde{\mathcal{B}}) \circ \mathbf{C}_{s+t-1} + \widehat{\boldsymbol{\Sigma}} \right] \left[ \sum_{k=0}^{N-t-1} \mathbf{A}^k \right]^\top$$

$$\preceq \frac{1}{N^2} \sum_{t=1}^{N-1} \left[ \sum_{k=0}^{N-t-1} \mathbf{A}^k \right] \left[ \mathbb{E}[\widehat{\mathbf{V}}_2 \otimes \widehat{\mathbf{V}}_2] \circ \mathbf{C}_\infty + \widehat{\boldsymbol{\Sigma}} \right] \left[ \sum_{k=0}^{N-t-1} \mathbf{A}^k \right]^\top$$

$$\preceq \frac{1}{N^2} \sum_{t=1}^{N-1} \left[ \sum_{k=0}^{N-t-1} \mathbf{A}^k \right] \left[ \sigma^2 \begin{bmatrix} \delta^2 & \delta q \\ \delta q & q^2 \end{bmatrix} \otimes \left( \mathrm{diag} \left( \frac{\lambda_1(\mathbf{U}_1)_{22}}{1 - (\mathbf{U}_1)_{22}}, \dots, \frac{\lambda_d(\mathbf{U}_d)_{22}}{1 - (\mathbf{U}_d)_{22}} \right) + \mathbf{H} \right) \right] \left[ \sum_{k=0}^{N-t-1} \mathbf{A}^k \right]^\top$$

$$= \frac{\sigma^2}{N^2} \sum_{t=1}^{N-1} \left[ \sum_{k=0}^{N-t-1} \mathbf{A}^k \right] \left[ \begin{bmatrix} \delta^2 & \delta q \\ \delta q & q^2 \end{bmatrix} \otimes \mathrm{diag} \left( \frac{\lambda_1}{1 - (\mathbf{U}_1)_{22}}, \dots, \frac{\lambda_d}{1 - (\mathbf{U}_d)_{22}} \right) \right] \left[ \sum_{k=0}^{N-t-1} \mathbf{A}^k \right]^\top$$

$$\preceq \frac{\sigma^2 r}{N^2} \sum_{t=1}^{N-1} \left[ \sum_{k=0}^{N-t-1} \mathbf{A}^k \right] \left[ \begin{bmatrix} \delta^2 & \delta q \\ \delta q & q^2 \end{bmatrix} \otimes \mathbf{H} \right] \left[ \sum_{k=0}^{N-t-1} \mathbf{A}^k \right]^\top, \tag{J.4}$$

where the first inequality holds because $\mathcal{B} - \widetilde{\mathcal{B}} = \mathbb{E}[\widehat{\mathbf{V}}_2 \otimes \widehat{\mathbf{V}}_2] - \mathbf{V}_2 \otimes \mathbf{V}_2 \preceq \mathbb{E}[\widehat{\mathbf{V}}_2 \otimes \widehat{\mathbf{V}}_2]$ and $\mathcal{B}_{s+t-1} \preceq \mathbf{C}_\infty$, the second inequality holds due to (J.3), and the last inequality holds due to definition of $r$. The inner product of $\mathbf{M}_2$ and $\begin{bmatrix} \mathbf{H} & \mathbf{0} \\ \mathbf{0} & \mathbf{0} \end{bmatrix}$ can thus be bounded by

$$\left\langle \begin{bmatrix} \mathbf{H} & \mathbf{0} \\ \mathbf{0} & \mathbf{0} \end{bmatrix}, \mathbf{M}_2 \right\rangle \leq \frac{\sigma^2 r}{N^2} \sum_{i=1}^{d} \lambda_i^2 \sum_{t=1}^{N-1} \left( \sum_{k=0}^{N-t-1} \mathbf{A}_i^k \begin{bmatrix} \delta \\ q \end{bmatrix} \right)_2^2$$

$$\leq \sigma^2 r \left[ \frac{9k^*}{N} + \frac{36N(q-c\delta)^2}{(1-c)^2} \sum_{i>k^*} \lambda_i^2 \right],$$

where the second inequality holds by deduction similar to that of Lemma G.3. $\qquad \square$

**Lemma J.8** (Modified from Lemma G.6). For any $t > 0$, $\mathbf{C}_t$ can be upper bounded by

$$\mathbf{C}_t \preceq \sigma^2 r \sum_{k=0}^{t-1} \widetilde{\mathcal{B}}^k \circ \left( \begin{bmatrix} \delta^2 & \delta q \\ \delta q & q^2 \end{bmatrix} \otimes \mathbf{H} \right).$$

*Proof.* By the iteration formula $\mathbf{C}_t = \mathcal{B} \circ \mathbf{C}_{t-1} + \widehat{\mathbf{\Sigma}}$, we have

$$\mathbf{C}_t = \widetilde{\mathcal{B}} \circ \mathbf{C}_{t-1} + (\mathcal{B} - \widetilde{\mathcal{B}}) \circ \mathbf{C}_{t-1} + \widehat{\mathbf{\Sigma}}$$

$$\preceq \widetilde{\mathcal{B}} \circ \mathbf{C}_{t-1} + \mathbb{E}[\widehat{\mathbf{V}}_2 \otimes \widehat{\mathbf{V}}_2] \circ \mathbf{C}_{t-1} + \widehat{\mathbf{\Sigma}}$$

$$\preceq \widetilde{\mathcal{B}} \circ \mathbf{C}_{t-1} + \mathbb{E}[\widehat{\mathbf{V}}_2 \otimes \widehat{\mathbf{V}}_2] \circ \mathbf{C}_\infty + \sigma^2 \begin{bmatrix} \delta^2 & \delta q \\ \delta q & q^2 \end{bmatrix} \otimes \mathbf{H}$$

$$\preceq \widetilde{\mathcal{B}} \circ \mathbf{C}_{t-1} + \sigma^2 \begin{bmatrix} \delta^2 & \delta q \\ \delta q & q^2 \end{bmatrix} \otimes \mathrm{diag}\left( \frac{\lambda_1}{1-(\mathbf{U}_1)_{22}}, \cdots, \frac{\lambda_d}{1-(\mathbf{U}_d)_{22}} \right)$$

$$\leq \widetilde{\mathcal{B}} \circ \mathbf{C}_{t-1} + \sigma^2 r \begin{bmatrix} \delta^2 & \delta q \\ \delta q & q^2 \end{bmatrix} \otimes \mathbf{H},$$

where the first inequality holds because $\mathcal{B} - \widetilde{\mathcal{B}} \preceq \mathbb{E}[\widehat{\mathbf{V}}_2 \otimes \widehat{\mathbf{V}}_2]$, the second inequality holds because $\mathbf{C}_{t-1} \preceq \mathbf{C}_\infty$ and Lemma G.4, the third inequality holds due to (J.3), and the last inequality holds due to the definition of $r$. Iterating the inequality above, we have

$$\mathbf{C}_t \preceq \sigma^2 r \sum_{k=0}^{t-1} \widetilde{\mathcal{B}}^k \circ \left( \begin{bmatrix} \delta^2 & \delta q \\ \delta q & q^2 \end{bmatrix} \otimes \mathbf{H} \right).$$

$\qquad \square$

As the bound for $\mathbf{C}_t$ is exactly the same as the bound given in Lemma G.6, we can prove the Lemma J.2 in exactly the same way as Lemma G.2.

### J.3 BIAS UPPER BOUND

**Lemma J.9** (Modified from Lemma H.1). For any $t \geq 0$, $\mathbf{B}_t$ can be upper bounded by

$$\mathbf{B}_t \preceq \widetilde{\mathcal{B}}^t \circ \mathbf{B}_0 + \sum_{k=0}^{t-1} \widetilde{\mathcal{B}}^k \circ \left( \begin{bmatrix} \delta^2 & \delta q \\ \delta q & q^2 \end{bmatrix} \otimes (\mathbf{H} \odot (\mathbf{B}_{t-1-k})_{22}) \right).$$

*Proof.* By the iterative formula $\mathbf{B}_t = \mathcal{B} \circ \mathbf{B}_{t-1}$, we have

$$\mathbf{B}_t = \widetilde{\mathcal{B}} \circ \mathbf{B}_{t-1} + (\mathcal{B} - \widetilde{\mathcal{B}}) \circ \mathbf{B}_{t-1}$$

$$\preceq \widetilde{\mathcal{B}} \circ \mathbf{B}_{t-1} + \mathbb{E}[\widehat{\mathbf{V}}_2 \otimes \widehat{\mathbf{V}}_2] \circ \mathbf{B}_{t-1}$$

$$= \widetilde{\mathcal{B}} \circ \mathbf{B}_{t-1} + \begin{bmatrix} \delta^2 & \delta q \\ \delta q & q^2 \end{bmatrix} \otimes (\mathbf{H} \odot (\mathbf{B}_{t-1})_{22})$$

$$\preceq \widetilde{\mathcal{B}}^t \circ \mathbf{B}_0 + \sum_{k=0}^{t-1} \widetilde{\mathcal{B}}^k \circ \left( \begin{bmatrix} \delta^2 & \delta q \\ \delta q & q^2 \end{bmatrix} \otimes (\mathbf{H} \odot (\mathbf{B}_{t-1-k})_{22}) \right),$$

where the first inequality holds because $\mathcal{B} - \widetilde{\mathcal{B}} \preceq \mathbb{E}[\widehat{\mathbf{V}}_2 \otimes \widehat{\mathbf{V}}_2]$, and the second inequality holds by iteratively applying the previous inequality. $\qquad \square$

**Lemma J.10.** We have

$$\sum_{t=0}^{s-1}((\mathbf{B}_t)_{22})_{ii} \le rw_i^2 \sum_{t=0}^{s-1}\left(\mathbf{A}_i^t\begin{bmatrix}1\\1\end{bmatrix}\right)_2^2.$$

*Proof.* Note that

$$
\begin{aligned}
\sum_{t=0}^{s-1}((\mathbf{B}_t)_{22})_{ii} &= \sum_{t=0}^{s-1}\left((\mathcal{B}^t\circ\mathbf{B}_0)_{22}\right)_{ii}\\
&\le \sum_{t=0}^{s-1}\left(\left(\widetilde{\mathcal{B}}^t\circ\mathbf{B}_0\right)_{22}\right)_{ii} + \sum_{t=0}^{s-1}\frac{((\widetilde{\mathcal{B}}^t\circ\mathbf{B}_0)_{22})_{ii}}{1-(\mathbf{U}_i)_{22}}(\mathbf{U}_i)_{22}\\
&= \frac{w_i^2}{1-(\mathbf{U}_i)_{22}}\sum_{t=0}^{s-1}\left(\mathbf{A}_i^t\begin{bmatrix}1\\1\end{bmatrix}\right)_2^2\\
&\le rw_i^2\sum_{t=0}^{s-1}\left(\mathbf{A}_i^t\begin{bmatrix}1\\1\end{bmatrix}\right)_2^2,
\end{aligned}
$$

where the first inequality holds due to Lemma J.7, and the second inequality holds due to the definition of $r$. $\qquad\square$

*Proof of Lemma J.4.* By the bound for $\mathbf{B}_s$, we have

$$
\begin{aligned}
\mathbf{M}_3 &= \frac{1}{N^2}\left[\sum_{k=0}^{N-1}\mathbf{A}^k\right]\mathbf{B}_s\left[\sum_{k=0}^{N-1}\mathbf{A}^k\right]^\top \preceq \frac{1}{N^2}\left[\sum_{k=0}^{N-1}\mathbf{A}^{k+s}\right]\mathbf{B}_0\left[\sum_{k=0}^{N-1}\mathbf{A}^{k+s}\right]^\top\\
&\quad + \frac{1}{N^2}\sum_{t=0}^{s-1}\left[\sum_{k=0}^{N-1}\mathbf{A}^{k+t}\right]\left(\begin{bmatrix}\delta^2 & \delta q\\\delta q & q^2\end{bmatrix}\otimes(\mathbf{H}\odot(\mathbf{B}_{s-1-t})_{22})\right)\left[\sum_{k=0}^{N-1}\mathbf{A}^{k+t}\right]^\top,
\end{aligned}
$$

so its inner product with $\begin{bmatrix}\mathbf{H} & \mathbf{0}\\\mathbf{0} & \mathbf{0}\end{bmatrix}$ is

$$
\begin{aligned}
\left\langle\begin{bmatrix}\mathbf{H} & \mathbf{0}\\\mathbf{0} & \mathbf{0}\end{bmatrix},\mathbf{M}_3\right\rangle &\le \underbrace{\sum_{i=1}^d\lambda_i w_i^2\left(\sum_{k=0}^{N-1}\mathbf{A}_i^{k+s}\begin{bmatrix}1\\1\end{bmatrix}\right)_1^2}_{\text{Effective Bias}}\\
&\quad + \underbrace{\frac{1}{N^2}\sum_{i=1}^d\lambda_i^2\sum_{t=0}^{s-1}((\mathbf{B}_{s-1-t})_{22})_{ii}\left(\sum_{k=0}^{N-1}\mathbf{A}_i^{k+t}\begin{bmatrix}\delta\\q\end{bmatrix}\right)_1^2}_{\text{K}}.
\end{aligned}
$$

The Effective Bias is the same as the standard case. K can be bounded by

$$
\begin{aligned}
\mathrm{K} &\le \frac{1}{N^2}\left[\sum_{i=1}^{k^*}\lambda_i^2\sum_{t=0}^{s-1}((\mathbf{B}_{s-1-t})_{22})_{ii}\cdot\frac{9}{\lambda_i^2} + \sum_{i=k^*+1}^d\lambda_i^2\sum_{t=0}^{t-1}((\mathbf{B}_{s-1-t})_{22})_{ii}\cdot\frac{36(q-c\delta)^2N^2}{(1-c)^2}\right]\\
&= \frac{1}{N^2}\left[9\sum_{i=1}^{k^*}\sum_{t=0}^{s-1}((\mathbf{B}_{s-1-t})_{22})_{ii} + \sum_{i=k^*+1}^d\frac{36(q-c\delta)^2N^2\lambda_i^2}{(1-c)^2}\sum_{t=0}^{s-1}((\mathbf{B}_{s-1-t})_{22})_{ii}\right]\\
&\le \frac{r}{N^2}\left[\sum_{i\le\widehat{k}}\frac{126w_i^2}{\delta\lambda_i} + \sum_{\widehat{k}<i\le k^\dagger}\frac{90w_i^2}{1-c} + \sum_{k^\dagger<i\le k^*}\frac{9(1-c)w_i^2}{(q-c\delta)\lambda_i} + \sum_{i>k^*}\frac{36(q-c\delta)^2N^2s\lambda_i^2w_i^2}{(1-c)^2}\right]\\
&= \frac{r}{N^2}\left[\frac{126}{\delta}\|\mathbf{w}_0-\mathbf{w}^*\|_{\mathbf{H}_{0:\widehat{k}}^{-1}}^2 + \frac{90}{1-c}\|\mathbf{w}_0-\mathbf{w}^*\|_{\mathbf{I}_{\widehat{k}:k^\dagger}}^2 + \frac{9(1-c)}{q-c\delta}\|\mathbf{w}_0-\mathbf{w}^*\|_{\mathbf{H}_{k^\dagger:k^*}^{-1}}^2\right.\\
&\quad\left. + \frac{36(q-c\delta)^2N^2s}{(1-c)^2}\|\mathbf{w}_0-\mathbf{w}^*\|_{\mathbf{H}_{k^*:\infty}^2}^2\right],
\end{aligned}
$$

where the first inequality holds due to Corollary K.7, and the second inequality holds due to Lemma J.10. □

*Proof of Lemma J.5.* For $\mathbf{M}_4$, we have

$$
\begin{aligned}
\mathbf{M}_4 &= \frac{1}{N^2} \sum_{t=1}^{N-1} \left[ \sum_{k=0}^{N-t-1} \mathbf{A}^k \right] \left( (\mathcal{B} - \widetilde{\mathcal{B}}) \circ \mathbf{B}_{s+t-1} \right) \left[ \sum_{k=0}^{N-t-1} \mathbf{A}^k \right]^\top \\
&\preceq \frac{1}{N^2} \sum_{t=1}^{N-1} \left[ \sum_{k=0}^{N-t-1} \mathbf{A}^k \right] \left( \mathbb{E}[\widehat{\mathbf{V}}_2 \otimes \widehat{\mathbf{V}}_2] \circ \mathbf{B}_{s+t-1} \right) \left[ \sum_{k=0}^{N-t-1} \mathbf{A}^k \right]^\top \\
&= \frac{1}{N^2} \sum_{t=1}^{N-1} \left[ \sum_{k=0}^{N-t-1} \mathbf{A}^k \right] \left( \begin{bmatrix} \delta^2 & \delta q \\ \delta q & q^2 \end{bmatrix} \otimes (\mathbf{H} \odot (\mathbf{B}_{s+t-1})_{22}) \right) \left[ \sum_{k=0}^{N-t-1} \mathbf{A}^k \right]^\top,
\end{aligned}
$$

where the inequality holds because $\mathcal{B} - \widetilde{\mathcal{B}} \preceq \mathbb{E}[\widehat{\mathbf{V}}_2 \otimes \widehat{\mathbf{V}}_2]$. The inner produce of $\mathbf{M}_4$ and $\begin{bmatrix} \mathbf{H} & \mathbf{0} \\ \mathbf{0} & \mathbf{0} \end{bmatrix}$ is thus bounded by

$$
\begin{aligned}
&\left\langle \mathbf{M}_4, \begin{bmatrix} \mathbf{H} & \mathbf{0} \\ \mathbf{0} & \mathbf{0} \end{bmatrix} \right\rangle \\
&\leq \frac{1}{N^2} \sum_{i=1}^d \lambda_i^2 \sum_{t=1}^{N-1} ((\mathbf{B}_{s+t-1})_{22})_{ii} \left( \sum_{k=0}^{N-t-1} \mathbf{A}_i^k \begin{bmatrix} \delta \\ q \end{bmatrix} \right)_1^2 \\
&\leq \frac{1}{N^2} \left[ 9 \sum_{i=1}^{k^*} \sum_{t=1}^{N-1} ((\mathbf{B}_{s+t-1})_{22})_{ii} + \sum_{i=k^*+1}^d \frac{36(q-c\delta)^2 N^2 \lambda_i^2}{(1-c)^2} \sum_{t=1}^{N-1} ((\mathbf{B}_{s+t-1})_{22})_{ii} \right] \\
&\leq \frac{1}{N^2} \left[ 9 \sum_{i=1}^{k^*} \sum_{t=0}^{s+N-1} ((\mathbf{B}_t)_{22})_{ii} + \sum_{i=k^*+1}^d \frac{36(q-c\delta)^2 N^2 \lambda_i^2}{(1-c)^2} \sum_{t=0}^{s+N-1} ((\mathbf{B}_t)_{22})_{ii} \right] \\
&\leq \frac{r}{N^2} \left[ \frac{126}{\delta} \| \mathbf{w}_0 - \mathbf{w}^* \|_{\mathbf{H}_{0:\hat{k}}^{-1}}^2 + \frac{90}{1-c} \| \mathbf{w}_0 - \mathbf{w}^* \|_{\mathbf{I}_{\hat{k}:k^\dagger}}^2 + \frac{9(1-c)}{q-c\delta} \| \mathbf{w}_0 - \mathbf{w}^* \|_{\mathbf{H}_{k^\dagger:k^*}^{-1}}^2 \right. \\
&\quad \left. + \frac{36(q-c\delta)^2 N^2 (s+N)}{(1-c)^2} \| \mathbf{w}_0 - \mathbf{w}^* \|_{\mathbf{H}_{k^*:\infty}^2}^2 \right],
\end{aligned}
$$

where the second inequality holds due to Corollary K.7, the second inequality holds due to Corollary K.7, the third inequality holds because $\sum_{t=1}^{N-1} ((\mathbf{B}_{s+t-1})_{22})_{ii} \leq \sum_{t=0}^{s+N-1} ((\mathbf{B}_t)_{22})_{ii}$, and the last inequality holds due to Lemma J.10. □

# K  AUXILIARY LEMMAS

The following lemma summarizes properties of auxiliary parameters $q$ and $c$ in relation to model parameters $\alpha, \beta, \gamma$ and $\delta$.

**Lemma K.1.** We have the following properties regarding $q$ and $c$:

(a) We have $c = 2\alpha - 1$, and $0 < c < 1$. Moreover, $\beta \leq 1 - c = 2\alpha\beta \leq 2\beta$.

(b) We have $\delta \leq q \leq (1+c)\delta$. Thus, $q - \delta \leq c(q - c\delta)$.

(c) We have
$$
\frac{q - c\delta}{1 - c} = \frac{\gamma + \delta}{2}, \quad \frac{q - \delta}{1 - c} = \frac{\gamma - \delta}{2}.
$$

Thus,
$$
\delta \leq \frac{q - c\delta}{1 - c} \leq \gamma.
$$

*Proof.* We first recall that $c = \alpha(1 - \beta)$ and $q = \alpha\delta + (1 - \alpha)\gamma$.

(a) Substituting $\beta = (1 - \alpha)/\alpha$ into the definition of $c$, we have

$$c = \alpha \left( 1 - \frac{1 - \alpha}{\alpha} \right) = 2\alpha - 1.$$

Note that $\beta \in (0, 1)$, so $\alpha = 1/(1 + \beta) \in (1/2, 1)$. Therefore, $c = 2\alpha - 1 \in (0, 1)$. Moreover,

$$1 - c = 1 - \alpha(1 - \beta) = 1 - \alpha + \alpha\beta \geq (1 - \alpha)\beta + \alpha\beta = \beta,$$

where the equality holds because $\beta < 1$. We also have

$$1 - c = 2(1 - \alpha) = 2\alpha\beta \leq 2\beta,$$

where the inequality holds because $\alpha < 1$.

(b) we have

$$q - \delta = \alpha\delta + (1 - \alpha)\gamma - \delta = (1 - \alpha)(\gamma - \delta) \geq 0, \tag{K.1}$$

where the inequality holds because $\gamma \geq \delta$ and $\alpha \in (0, 1)$. We also have

$$q - (1 + c)\delta = \alpha\delta + (1 - \alpha)\gamma - 2\alpha\delta = (1 - \alpha)\gamma - \alpha\delta = \alpha(\beta\gamma - \delta) = \alpha \left( \frac{\delta}{\psi\widetilde{\kappa}} - \delta \right) \leq 0,$$

where the third equality holds because $1 - \alpha = \alpha\beta$, the fourth equality holds because $\beta = \delta/(\psi\widetilde{\kappa}\gamma)$, and the last inequality holds because $\psi\widetilde{\kappa} \geq 1$. We thus have

$$(q - \delta) - c(q - c\delta) = (1 - c)[q - (1 + c)\delta] \leq 0.$$

(c) We have

$$q - c\delta = \alpha\delta + (1 - \alpha)\gamma - (2\alpha - 1)\delta = (1 - \alpha)(\gamma + \delta). \tag{K.2}$$

Combining (K.1) and (K.2) with the fact that $1 - c = 2(1 - \alpha)$, we have

$$\frac{q - c\delta}{1 - c} = \frac{\gamma + \delta}{2}, \quad \frac{q - \delta}{1 - c} = \frac{\gamma - \delta}{2}.$$

Note that $\delta \leq \gamma$, so

$$\delta \leq \frac{q - c\delta}{1 - c} \leq \gamma.$$

$\square$

**Lemma K.2.** Let $x_1, x_2$ be defined in (E.2) and (E.3). Then we have

(a) $(1 - x_1)(1 - x_2) = (q - c\delta)\lambda_i$.
(b) $(c - x_1)(c - x_2) = c(q - \delta)\lambda_i$.
(c) $(1 + x_1)(1 + x_2) = 2(1 + c) - (q + c\delta)\lambda_i$.
(d) $(c\delta - qx_1)(c\delta - qx_2) = c(q - \delta)(q - c\delta)$.

*Proof.* In the proof, we will use the properties $x_1 + x_2 = 1 + c - q\lambda_i$ and $x_1 x_2 = c(1 - \delta\lambda_i)$ extensively, which follows from Veda's Theorem.

(a) We have

$$(1 - x_1)(1 - x_2) = 1 - (x_1 + x_2) - x_1 x_2 = 1 - (1 + c - q\lambda_i) - c(1 - \delta\lambda_i) = (q - c\delta)\lambda_i.$$

(b) We have

$$(c - x_1)(c - x_2) = c^2 - c(x_1 + x_2) + x_1 x_2 = c^2 - c(1 + c - q\lambda_i) + c(1 - \delta\lambda_i) = c(q - \delta)\lambda_i.$$

(c) We have

$$(1 + x_1)(1 + x_2) = 1 + (x_1 + x_2) + x_1 x_2 = 1 + (1 + c - q\lambda_i) + c(1 - \delta\lambda_i)$$
$$= 2(1 + c) - (q + c\delta)\lambda_i.$$

(d) We have

$$(c\delta - qx_1)(c\delta - qx_2) = c^2\delta^2 - c\delta q(x_1 + x_2) + q^2 x_1 x_2$$
$$= c^2\delta^2 - c\delta q(1 + c - q\lambda_i) + q^2 \cdot c(1 - \delta\lambda_i)$$
$$= c(q - \delta)(q - c\delta)$$

$\square$

**Lemma K.3.** For a given PSD matrix $\mathbf{M}$, we define the following sequence of matrices recursively: $\mathbf{R}_0 = \mathbf{0}$, and

$$\mathbf{R}_{t+1} = \mathcal{B} \circ \mathbf{R}_t + \mathbf{M}, \quad t \geq 0. \tag{K.3}$$

Then for all $t \geq 0$, we have

$$\mathbf{R}_t = \sum_{k=0}^{t-1} \mathcal{B}^k \circ \mathbf{M}. \tag{K.4}$$

Thus, $\mathbf{R}_t$ is an increasing sequence:

$$\mathbf{R}_0 \preceq \mathbf{R}_1 \preceq \cdots \preceq \mathbf{R}_\infty. \tag{K.5}$$

*Proof.* We prove (K.4) by induction. When $t = 0$, (K.4) holds trivially. Suppose that (K.4) holds for $t$. By the recursive formula K.3, we have

$$\mathbf{R}_{t+1} = \mathcal{B} \circ \mathbf{R}_t + \mathbf{M} = \mathcal{B} \circ \left( \sum_{k=0}^{t-1} \mathcal{B}^k \circ \mathbf{M} \right) + \mathbf{M} = \sum_{k=1}^{t} \mathcal{B}^k \circ \mathbf{M} + \mathbf{M} = \sum_{k=0}^{t} \mathcal{B}^k \circ \mathbf{M},$$

where the second equality holds due to the induction hypothesis. Thus, (K.4) holds for $t + 1$. By (K.4), note that

$$\mathbf{R}_{t+1} - \mathbf{R}_t = \sum_{k=0}^{t} \mathcal{B}^k \circ \mathbf{M} - \sum_{k=0}^{t-1} \mathcal{B}^k \circ \mathbf{M} = \mathcal{B}^t \circ \mathbf{M} \succeq 0,$$

where the inequality holds due to Lemma F.2(c). Therefore, $\mathbf{R}_t \preceq \mathbf{R}_{t+1}$. $\square$

**Lemma K.4.** Let $\{\mathbf{M}_t\}_{t \geq 1}$ be a sequence of PSD matrices and $s, N$ be positive integers. Then

$$\sum_{t=s}^{s+N-1} \left[ \sum_{k=t+1}^{s+N-1} \mathbf{A}^{k-t} \mathbf{M}_t + \mathbf{M}_t + \sum_{k=t+1}^{s+N-1} \mathbf{M}_t (\mathbf{A}^\top)^{k-t} \right]$$

$$= \left[ \sum_{k=0}^{N-1} \mathbf{A}^k \right] \mathbf{M}_s \left[ \sum_{k=0}^{N-1} \mathbf{A}^k \right]^\top + \sum_{t=1}^{N-1} \left[ \sum_{k=0}^{N-t-1} \mathbf{A}^k \right] (\mathbf{M}_{s+t} - \widetilde{\mathcal{B}} \circ \mathbf{M}_{s+t-1}) \left[ \sum_{k=0}^{N-t-1} \mathbf{A}^k \right]^\top.$$

*Proof.* For $t = s, s+1, \ldots, s+N-2$, we have

$$\left[ \sum_{j=0}^{s+N-t-1} \mathbf{A}^j \right] \mathbf{M}_t \left[ \sum_{k=0}^{s+N-t-1} \mathbf{A}^k \right]^\top - \left[ \sum_{j=0}^{s+N-t-2} \mathbf{A}^j \right] (\mathbf{A} \mathbf{M}_t \mathbf{A}^\top) \left[ \sum_{k=0}^{s+N-t-2} \mathbf{A}^k \right]^\top$$

$$= \sum_{j,k=0}^{s+N-t-1} \mathbf{A}^j \mathbf{M}_t (\mathbf{A}^k)^\top - \sum_{j,k=1}^{s+N-t-1} \mathbf{A}^j \mathbf{M}_t (\mathbf{A}^k)^\top$$

$$= \sum_{j=1}^{s+N-t-1} \mathbf{A}^j \mathbf{M}_t + \mathbf{M}_t + \sum_{k=1}^{s+N-t-1} (\mathbf{A}^\top)^k$$

$$= \sum_{k=t+1}^{s+N-1} \mathbf{A}^{k-t} \mathbf{M}_t + \mathbf{M}_t + \sum_{k=t+1}^{s+N-1} (\mathbf{A}^\top)^{k-t}.$$

Take the sum over $t$, and we have

$$\sum_{t=s}^{s+N-1} \left[ \sum_{k=t+1}^{s+N-1} \mathbf{A}^{k-t} \mathbf{M}_t + \mathbf{M}_t + \sum_{k=t+1}^{s+N-1} \mathbf{M}_t (\mathbf{A}^\top)^{k-t} \right]$$

$$= \mathbf{M}_{s+N-1} + \sum_{t=s}^{s+N-2} \left[ \sum_{k=0}^{s+N-t-1} \mathbf{A}^k \right] \mathbf{M}_t \left[ \sum_{k=0}^{s+N-t-1} \mathbf{A}^k \right]^\top$$

$$-\sum_{t=s}^{s+N-2}\left[\sum_{k=0}^{s+N-t-2}\mathbf{A}^k\right](\mathbf{A}\mathbf{M}_t\mathbf{A}^\top)\left[\sum_{k=0}^{s+N-t-2}\mathbf{A}^k\right]^\top$$

$$=\left[\sum_{k=0}^{N-1}\mathbf{A}^k\right]\mathbf{M}_s\left[\sum_{k=0}^{N-1}\mathbf{A}^k\right]^\top+\sum_{t=s+1}^{s+N-1}\left[\sum_{k=0}^{s+N-t-1}\mathbf{A}^k\right]\mathbf{M}_t\left[\sum_{k=0}^{s+N-t-1}\mathbf{A}^k\right]^\top$$

$$-\sum_{t=s+1}^{s+N-1}\left[\sum_{k=0}^{s+N-t-1}\mathbf{A}^k\right](\mathbf{A}\mathbf{M}_{t-1}\mathbf{A}^\top)\left[\sum_{k=0}^{s+N-t-1}\mathbf{A}^k\right]^\top$$

$$=\left[\sum_{k=0}^{N-1}\mathbf{A}^k\right]\mathbf{M}_s\left[\sum_{k=0}^{N-1}\mathbf{A}^k\right]^\top+\sum_{t=s+1}^{s+N-1}\left[\sum_{k=0}^{s+N-t-1}\mathbf{A}^k\right](\mathbf{M}_t-\widetilde{\mathcal{B}}\circ\mathbf{M}_{t-1})\left[\sum_{k=0}^{s+N-t-1}\mathbf{A}^k\right]^\top$$

$$=\left[\sum_{k=0}^{N-1}\mathbf{A}^k\right]\mathbf{M}_s\left[\sum_{k=0}^{N-1}\mathbf{A}^k\right]^\top+\sum_{t=1}^{N-1}\left[\sum_{k=0}^{N-t-1}\mathbf{A}^k\right](\mathbf{M}_{s+t}-\widetilde{\mathcal{B}}\circ\mathbf{M}_{s+t-1})\left[\sum_{k=0}^{N-t-1}\mathbf{A}^k\right]^\top,$$

where the second equality holds due to change of index, the third equality holds due to the definition of $\widetilde{\mathcal{B}}$, and the fourth equality holds also due to change of index. $\qquad\square$

**Lemma K.5.** With $\mathbf{A}_i$ defined in (E.1), let $x_1$ and $x_2$ be the eigenvalues of $\mathbf{A}_i$ defined in (E.2) and (E.3). Then

- For all $i \le k^\ddagger$, we have

$$-\frac{2}{\lambda_i}(c\delta/q)^j \le \left(\sum_{k=0}^{t-1}\mathbf{A}_i^{j+k}\begin{bmatrix}\delta\\q\end{bmatrix}\right)_1 \le \frac{2}{\lambda_i}(c\delta/q)^j.$$

- For all $k^\ddagger < i \le \widehat{k}$, we have

$$\left|\left(\sum_{k=0}^{t-1}\mathbf{A}_i^{j+k}\begin{bmatrix}\delta\\q\end{bmatrix}\right)_1\right| \le \frac{3}{\lambda_i}[c(1-\delta\lambda_i)]^{j/2}+\delta j[c(1-\delta\lambda_i)]^{(j-1)/2}$$

- For all $\widehat{k} < i \le k^\dagger$, we have

$$\left|\left(\sum_{k=0}^{t-1}\mathbf{A}_i^{j+k}\begin{bmatrix}\delta\\q\end{bmatrix}\right)_1\right| \le \frac{3}{\lambda_i}[c(1-\delta\lambda_i)]^{j/2}+\frac{1-c}{\lambda_i}\cdot j[c(1-\delta\lambda_i)]^{(j-1)/2}$$

- For all $i > k^\dagger$, we have

$$0 \le \left(\sum_{k=0}^{t-1}\mathbf{A}_i^{j+k}\begin{bmatrix}\delta\\q\end{bmatrix}\right)_1 \le \frac{3}{\lambda_i}\left[1-\left(1-2\frac{q-c\delta}{1-c}\lambda_i\right)^t\right]\left(1-\frac{q-c\delta}{1-c}\lambda_i\right)^j$$

*Proof.* Note that

$$\sum_{k=0}^{t-1}\mathbf{A}_i^{j+k}\begin{bmatrix}\delta\\q\end{bmatrix}=(\mathbf{A}_i^j-\mathbf{A}_i^{j+t})(\mathbf{I}-\mathbf{A}_i)^{-1}\begin{bmatrix}\delta\\q\end{bmatrix}=(\mathbf{A}_i^j-\mathbf{A}_i^{j+t})\cdot\frac{1}{\lambda_i}\begin{bmatrix}1\\1\end{bmatrix}$$

$$=\frac{1}{\lambda_i}\begin{bmatrix}(\mathbf{A}_i^j)_{11}+(\mathbf{A}_i^j)_{12}-(\mathbf{A}_i^{j+t})_{11}-(\mathbf{A}_i^{j+t})_{12}\\(\mathbf{A}_i^j)_{11}+(\mathbf{A}_i^j)_{12}-(\mathbf{A}_i^{j+t})_{11}-(\mathbf{A}_i^{j+t})_{12}\end{bmatrix}. \tag{K.6}$$

Combining Lemma E.3 with (K.6), we have

$$\left(\sum_{k=0}^{t-1}\mathbf{A}_i^{j+k}\begin{bmatrix}\delta\\q\end{bmatrix}\right)_1=\frac{1}{\lambda_i}((\mathbf{A}_i^j)_{11}+(\mathbf{A}_i^j)_{12}-(\mathbf{A}_i^{j+t})_{11}-(\mathbf{A}_i^{j+t})_{12})$$

$$=\frac{1}{\lambda_i}\left[-\frac{x_1x_2^j-x_2x_1^j}{x_2-x_1}+(1-\delta\lambda_i)\frac{x_2^j-x_1^j}{x_2-x_1}+\frac{x_1x_2^{j+t}-x_2x_1^{j+t}}{x_2-x_1}-(1-\delta\lambda_i)\frac{x_2^{j+t}-x_1^{j+t}}{x_2-x_1}\right]$$

$$= \frac{1}{\lambda_i} \cdot \frac{(1 - \delta\lambda_i - x_1)x_2^j(1 - x_2^t) - (1 - \delta\lambda_i - x_2)x_1^j(1 - x_1^t)}{x_2 - x_1}. \tag{K.7}$$

For $i \leq k^{\ddagger}$, note that $x_1 x_2 = c(1 - \delta\lambda_i)$ and $x_2 \leq c$ by Lemma E.2, so we have

$$1 - \delta\lambda_i \leq x_1 \leq x_2 \leq c. \tag{K.8}$$

Thus, the upper bound of (K.7) is given by

$$\left( \sum_{k=0}^{t-1} \mathbf{A}_i^{j+t} \begin{bmatrix} \delta \\ q \end{bmatrix} \right)_1 \leq \frac{1}{\lambda_i} \cdot \frac{-(\delta\lambda_i + x_1 - 1)x_1^j(1 - x_2^t) + (\delta\lambda_i + x_2 - 1)x_1^j(1 - x_1^t)}{x_2 - x_1}$$

$$= \frac{x_1^j}{\lambda_i} \left[ (\delta\lambda_i + x_1 - 1)\frac{x_2^t - x_1^t}{x_2 - x_1} + (1 - x_1^t) \right]$$

$$\leq \frac{x_1^j}{\lambda_i} \left[ (\delta\lambda_i + x_1 - 1)\frac{1 - x_1^t}{1 - x_1} + (1 - x_1^t) \right] = \frac{\delta x_1^j(1 - x_1^t)}{1 - x_1} \leq \frac{\delta x_1^j}{1 - x_1}, \tag{K.9}$$

where the first inequality holds because $\delta\lambda_i + x_1 - 1 \geq 0$ and $x_2 \geq x_1$, and the second inequality holds due to Lemma K.12. Note that

$$\frac{1}{1 - x_1} = \frac{1 - x_2}{(1 - x_2)(1 - x_1)} = \frac{1 - x_2}{(q - c\delta)\lambda_i}$$

$$\leq \frac{1 - \frac{c\delta - \sqrt{c(q - \delta)(q - c\delta)}}{q}}{(q - c\delta)\lambda_i} = \frac{1}{q\lambda_i} \left( 1 + \sqrt{\frac{c(q - \delta)}{q - c\delta}} \right) \leq \frac{2}{\delta\lambda_i}, \tag{K.10}$$

where the second equality holds by Lemma K.2(a), the first inequality holds due to Lemma E.2, and the second inequality holds because $c(q - \delta) \leq q - c\delta$ and $q \geq \delta$. Note that $x_1 \leq x_2 \leq c\delta/q$, so (K.9) can be further bounded by

$$\left( \sum_{k=0}^{t-1} \mathbf{A}_i^{j+t} \begin{bmatrix} \delta \\ q \end{bmatrix} \right)_1 \leq \frac{\delta}{1 - x_1}(c\delta/q)^j \leq \frac{2}{\lambda_i}(c\delta/q)^j,$$

where the second inequality holds due to (K.10).

The lower bound of (K.7) is given by

$$\left( \sum_{k=0}^{t-1} \mathbf{A}_i^{j+k} \begin{bmatrix} \delta \\ q \end{bmatrix} \right)_1 \geq \frac{1}{\lambda_i} \cdot \frac{-(\delta\lambda_i + x_1 - 1)x_2^j(1 - x_2^t) + (\delta\lambda_i + x_2 - 1)x_1^j(1 - x_2^t)}{x_2 - x_1}$$

$$= \frac{1 - x_2^t}{\lambda_i} \cdot \frac{-(\delta\lambda_i + x_1 - 1)x_2^j + (\delta\lambda_i + x_2 - 1)x_1^j}{x_2 - x_1}, \tag{K.11}$$

where the first inequality holds because $\delta\lambda_i + x_1 - 1 \geq 0$ and $x_1 \leq x_2$. If $j \geq 1$, then

$$\frac{-(\delta\lambda_i + x_1 - 1)x_2^j + (\delta\lambda_i + x_2 - 1)x_1^j}{x_2 - x_1}$$

$$= -(\delta\lambda_i + x_1 - 1)x_2 \frac{x_2^{j-1} - x_1^{j-1}}{x_2 - x_1} + (1 - \delta\lambda_i)x_1^{j-1}$$

$$\geq -(\delta\lambda_i + x_1 - 1)x_2 \cdot \frac{(c\delta/q)^{j-1}}{c\delta/q - x_1}$$

$$= -\frac{x_1 x_2 - (1 - \delta\lambda_i)x_2}{c\delta/q - x_1} \cdot (c\delta/q)^{j-1}$$

$$= -(1 - \delta\lambda_i)\frac{c - x_2}{c\delta/q - x_1}(c\delta/q)^{j-1}, \tag{K.12}$$

where the inequality holds due to Lemma K.12, and the last equality holds because $x_1 x_2 = c(1 - \delta\lambda_i)$. Note that

$$(1 - \delta\lambda_i)\frac{c - x_2}{c\delta/q - x_1} = (1 - \delta\lambda_i) \cdot \frac{(c - x_2)(c\delta/q - x_2)}{(c\delta/q - x_1)(c\delta/q - x_2)} = \frac{q^2(1 - \delta\lambda_i)(c - x_2)(c\delta/q - x_2)}{c(q - \delta)(q - c\delta)}$$

$$\leq \frac{q^2}{c(q-\delta)(q-c\delta)} \cdot \left(1 - \delta \cdot \frac{(\sqrt{q-c\delta} + \sqrt{c(q-\delta)})^2}{q^2}\right) \cdot \left(c - \frac{c\delta - \sqrt{c(q-\delta)(q-c\delta)}}{q}\right)$$

$$\cdot \frac{\sqrt{c(q-\delta)(q-c\delta)}}{q}$$

$$= \frac{(c\delta - \sqrt{c(q-\delta)(q-c\delta)})^2}{cq^2} \left(1 + \sqrt{\frac{c(q-\delta)}{q-c\delta}}\right)$$

$$\leq \frac{c^2\delta^2}{cq^2} \cdot 2 \leq 2\frac{c\delta}{q}, \tag{K.13}$$

where the second equality holds due to Lemma K.2(d), the first inequality holds due to (E.6) and Lemma E.2, the second inequality holds because $c\delta - \sqrt{c(q-\delta)(q-c\delta)} \leq c\delta$ and $c(q-\delta) \leq q - c\delta$, and the last inequality holds because $\delta \leq q$. Therefore, substituting (K.12) and (K.13) into (K.11), we have

$$\left(\sum_{k=0}^{t-1} \mathbf{A}_i^{j+k} \begin{bmatrix} \delta \\ q \end{bmatrix}\right)_1 \geq -\frac{1-x_2^t}{\lambda_i} \cdot 2(c\delta/q)^j \geq -\frac{2}{\lambda_i}(c\delta/q)^j,$$

where the second inequality holds because $1 - x_2^t \leq 1$. If $j = 0$, then

$$\frac{1-x_2^t}{\lambda_i} \cdot \frac{-(\delta\lambda_i + x_1 - 1) + (\delta\lambda_i + x_2 - 1)}{x_2 - x_1} = \frac{1-x_2^t}{\lambda_i} \geq 0,$$

so the upper bound holds trivially.
For $k^\ddagger < i \leq k^\dagger$, the upper bound of (K.7) is given by

$$\left|\left(\sum_{k=0}^{t-1} \mathbf{A}_i^{j+k} \begin{bmatrix} \delta \\ q \end{bmatrix}\right)_1\right| = \frac{1}{\lambda_i}\left|\frac{(1-\delta\lambda_i - x_1)x_2^j(1-x_2^t) - (1-\delta\lambda_i - x_2)x_1^j(1-x_1^t)}{x_2 - x_1}\right|$$

$$= \frac{1}{\lambda_i}\left|\frac{x_2^j(1-x_2^t) + x_1^j(1-x_1^t)}{2}\right.$$

$$\left. + \left(1 - \delta\lambda_i - \frac{x_1+x_2}{2}\right) \cdot \frac{x_2^j(1-x_2^t) - x_1^j(1-x_1^t)}{x_2 - x_1}\right|$$

$$= \frac{1}{\lambda_i}\left|\frac{x_2^j(1-x_2^t) + x_1^j(1-x_1^t)}{2} + \frac{1 - c - (2\delta - q)\lambda_i}{2}\right.$$

$$\left. \cdot \left[\frac{x_2^j - x_1^j}{x_2 - x_1} \cdot \frac{(1-x_1^t) + (1-x_2^t)}{2} - \frac{x_2^t - x_1^t}{x_2 - x_1} \cdot \frac{x_2^j + x_1^j}{2}\right]\right|$$

$$\leq \frac{|x_2^j||1-x_2^t| + |x_1^j||1-x_1^t|}{2\lambda_i} + \frac{|1 - c - (2\delta - q)\lambda_i|}{2\lambda_i}$$

$$\cdot \left[\left|\frac{x_2^j - x_1^j}{x_2 - x_1}\right| \cdot \frac{|1-x_1^t| + |1-x_2^t|}{2} + \left|\frac{x_2^t - x_1^t}{x_2 - x_1}\right| \cdot \frac{|x_2^j| + |x_1^j|}{2}\right], \tag{K.14}$$

where the second equality holds because $x_1 + x_2 = 1 + c - q\lambda_i$, and the inequality holds due to triangle inequality. Note that $|1 - x_1^t| = |1 - x_2^t| \leq 1 + |x_2^t| \leq 2$ because $|x_2^t| \leq 1$. We can thus bound (K.14) as

$$\left|\left(\sum_{k=0}^{t-1} \mathbf{A}_i^{j+k} \begin{bmatrix} \delta \\ q \end{bmatrix}\right)_1\right|$$

$$\leq \frac{|x_1^j| + |x_2^j|}{\lambda_i} + \frac{|1 - c - (2\delta - q)\lambda_i|}{2\lambda_i} \cdot \left[2\left|\frac{x_2^j - x_1^j}{x_2 - x_1}\right| + \left|\frac{x_2^t - x_1^t}{x_2 - x_1}\right| \cdot \frac{|x_2^j| + |x_1^j|}{2}\right]$$

$$\leq \frac{2}{\lambda_i}[c(1-\delta\lambda_i)]^{j/2} + \frac{|1 - c - (2\delta - q)\lambda_i|}{2\lambda_i} \cdot \left[2j[c(1-\delta\lambda_i)]^{(j-1)/2} + t[c(1-\delta\lambda_i)]^{(j+t-1)/2}\right]$$

$$
= j \cdot \frac{|1 - c - (2\delta - q)\lambda_i|}{\lambda_i} \cdot [c(1 - \delta\lambda_i)]^{(j-1)/2}
$$
$$
+ \left\{ \frac{2}{\lambda_i} + \frac{|1 - c - (2\delta - q)\lambda_i|}{2\lambda_i} \cdot t[c(1 - \delta\lambda_i)]^{(t-1)/2} \right\} \cdot [c(1 - \delta\lambda_i)]^{j/2}, \tag{K.15}
$$

where the second inequality holds due to Lemma K.13. For $k^{\ddagger} < i \le \widehat{k}$, we have

$$
1 - c - (2\delta - q)\lambda_i \le \delta\lambda_i - (2\delta - q)\lambda_i = (q - \delta)\lambda_i \le \delta\lambda_i,
$$

where the first inequality holds because $1 - c \le \delta\lambda_i$, and the second inequality holds because $q \le (1 + c)\delta \le 2\delta$. We also have

$$
1 - c - (2\delta - q)\lambda_i \ge (q - 2\delta)\lambda_i \ge (\delta - 2\delta)\lambda_i = -\delta\lambda_i,
$$

where the first inequality holds because $1 - c \ge 0$, and the second inequality holds because $q \ge \delta$. Therefore,

$$
|1 - c - (2\delta - q)\lambda_i| \le \delta\lambda_i. \tag{K.16}
$$

(K.15) can thus be bounded by

$$
\left| \left( \sum_{k=0}^{t-1} \mathbf{A}_i^{j+k} \begin{bmatrix} \delta \\ q \end{bmatrix} \right)_1 \right| \le \delta j [c(1 - \delta\lambda_i)]^{(j-1/2)} + \left( \frac{2}{\lambda_i} + \frac{\delta}{2} \cdot \frac{2}{\delta\lambda_i} \right) \cdot [c(1 - \delta\lambda_i)]^{j/2}
$$
$$
= \delta j [c(1 - \delta\lambda_i)]^{(j-1/2)} + \frac{3}{\lambda_i} [c(1 - \delta\lambda_i)]^{j/2},
$$

where the inequality holds due to (K.16) and Lemma K.14. For $\widehat{k} < i \le k^{\dagger}$, we have

$$
1 - c - (2\delta - q)\lambda_i \ge 1 - c - (1 - c)(2\delta - q)/\delta = \frac{(1 - c)(q - \delta)}{\delta} \ge 0,
$$

where the first inequality holds because $\lambda_i \le (1 - c)/\delta$, and the second inequality holds because $q \ge \delta$. We also have

$$
1 - c - (2\delta - q)\lambda_i \le 1 - c,
$$

where the inequality holds because $2\delta - q \ge 2\delta - (1 + c)\delta = (1 - c)\delta \ge 0$. Therefore,

$$
|1 - c - (2\delta - q)\lambda_i| \le 1 - c. \tag{K.17}
$$

(K.15) can thus be bounded as

$$
\left| \left( \sum_{k=0}^{t-1} \mathbf{A}_i^{j+k} \begin{bmatrix} \delta \\ q \end{bmatrix} \right)_1 \right| \le \frac{1 - c}{\lambda_i} \cdot j [c(1 - \delta\lambda_i)]^{(j-1)/2} + \left( \frac{2}{\lambda_i} + \frac{1 - c}{2\lambda_i} \cdot \frac{2}{1 - c} \right) \cdot [c(1 - \delta\lambda_i)]^{j/2}
$$
$$
= \frac{1 - c}{\lambda_i} \cdot j [c(1 - \delta\lambda_i)]^{(j-1)/2} + \frac{3}{\lambda_i} [c(1 - \delta\lambda_i)]^{j/2},
$$

where the inequality holds due to (K.17) and Lemma K.14.
For $i > k^{\dagger}$, note that

$$
1 - \delta\lambda_i - x_2 \ge (1 - \delta\lambda_i) - \left( 1 - \frac{q - c\delta}{1 - c}\lambda_i \right) = \frac{q - \delta}{1 - c}\lambda_i \ge 0,
$$

where the first inequality holds due to Lemma E.2, and the second inequality holds because $q \ge \delta$. The upper bound of (K.7) is thus given by

$$
\left( \sum_{k=0}^{t-1} \mathbf{A}_i^{j+k} \begin{bmatrix} \delta \\ q \end{bmatrix} \right)_1 \le \frac{1}{\lambda_i} \cdot \frac{(1 - \delta\lambda_i - x_1)x_2^j(1 - x_2^t) - (1 - \delta\lambda_i - x_2)x_1^j(1 - x_2^t)}{x_2 - x_1}
$$
$$
= \frac{1 - x_2^t}{\lambda_i} \left[ (1 - \delta\lambda_i - x_2)\frac{x_2^j - x_1^j}{x_2 - x_1} + x_2^j \right] \tag{K.18}
$$

where the inequality holds due to $x_1 < x_2$. If $j \ge 1$,

$$
(1 - \delta\lambda_i - x_2)\frac{x_2^j - x_1^j}{x_2 - x_1} + x_2^j = (1 - \delta\lambda_i - x_2)x_1 \frac{x_2^{j-1} - x_1^{j-1}}{x_2 - x_1} + (1 - \delta\lambda_i)x_2^{j-1}
$$

$$\leq \left(1 - \frac{q - c\delta}{1 - c}\lambda_i\right)^{j-1} \left[\frac{(1 - \delta\lambda_i - x_2)x_1}{1 - \frac{q-c\delta}{1-c}\lambda_i - x_1} + (1 - \delta\lambda_i)\right]$$

$$= \left(1 - \frac{q - c\delta}{1 - c}\lambda_i\right)^{j-1} \cdot \frac{(1 - \delta\lambda_i)\left(1 - \frac{q-c\delta}{1-c}\lambda_i\right) - c(1 - \delta\lambda_i)}{1 - \frac{q-c\delta}{1-c}\lambda_i - x_1}$$

$$= \left(1 - \frac{q - c\delta}{1 - c}\lambda_i\right)^{j} \frac{1 - \delta\lambda_i}{1 - \frac{q-c\delta}{1-c}\lambda_i} \cdot \frac{1 - \frac{q-c\delta}{1-c}\lambda_i - c}{1 - \frac{q-c\delta}{1-c}\lambda_i - x_1},$$

where the inequality holds due to Lemma K.12. Note that

$$\frac{1 - \delta\lambda_i}{1 - \frac{q-c\delta}{1-c}\lambda_i} \leq \frac{1 - \delta \cdot \frac{(1-c)^2}{(\sqrt{q-c\delta}+\sqrt{c(q-\delta)})^2}}{1 - \frac{q-c\delta}{1-c} \cdot \frac{(1-c)^2}{(\sqrt{q-c\delta}+\sqrt{c(q-\delta)})^2}}$$

$$= \frac{(1 + \sqrt{c(q - c\delta)/(q - \delta)})^2}{(1 + \sqrt{c(q - c\delta)/(q - \delta)})^2 - (1 - c)} \leq \frac{(1 + 1)^2}{(1 + 1)^2 - (1 - c)} = \frac{4}{3 + c}, \tag{K.19}$$

where the first inequality holds due to (E.6), and the second inequality holds because $q - \delta \leq c(q - c\delta)$. We also note that

$$\frac{1 - \frac{q-c\delta}{1-c}\lambda_i - c}{1 - \frac{q-c\delta}{1-c}\lambda_i - x_1} = \frac{1 - \frac{q-c\delta}{1-c}\lambda_i - c}{1 - \frac{q-c\delta}{1-c}\lambda_i - \frac{(1+c-q\lambda_i)-\sqrt{(1+c-q\lambda_i)^2 - 4c(1-\delta\lambda_i)}}{2}}$$

$$\leq 2 \cdot \frac{1 - c - \frac{q-c\delta}{1-c}\lambda_i}{1 - c - \frac{(1+c)q-2c\delta}{1-c}\lambda_i} \leq 2 \cdot \frac{1 - c - \frac{q-c\delta}{1-c} \cdot \frac{(1-c)^2}{(\sqrt{q-c\delta}+\sqrt{c(q-\delta)})^2}}{1 - c - \frac{(1+c)q-2c\delta}{1-c} \cdot \frac{(1-c)^2}{(\sqrt{q-c\delta}+\sqrt{c(q-\delta)})^2}}$$

$$= 2 + \sqrt{\frac{c(q - \delta)}{q - c\delta}} \leq 2 + c, \tag{K.20}$$

where the first inequality holds because $\sqrt{(1 + c - q\lambda_i)^2 - 4c(1 - \delta\lambda_i)} \geq 0$, the second inequality holds due to (E.6), and the third inequality holds because $q - \delta \leq c(q - c\delta)$. Combining (K.19) and (K.20), we have

$$\frac{1 - \delta\lambda_i}{1 - \frac{q-c\delta}{1-c}\lambda_i} \cdot \frac{1 - \frac{q-c\delta}{1-c}\lambda_i - c}{1 - \frac{q-c\delta}{1-c}\lambda_i - x_1} \leq \frac{4(2 + c)}{3 + c} \leq 3, \tag{K.21}$$

where the second inequality holds because $c \leq 1$. where the second inequality holds because $x_2 \geq 1 - 2\frac{q-c\delta}{1-c}\lambda_i$ due to Lemma E.2. Combining (K.18) with (K.21), we have

$$\left(\sum_{k=0}^{t-1} \mathbf{A}_i^{j+k} \begin{bmatrix} \delta \\ q \end{bmatrix}\right)_1 \leq \frac{3}{\lambda_i}\left(1 - \frac{q - c\delta}{1 - c}\lambda_i\right)^{j}(1 - x_2^t)$$

$$\leq \frac{3}{\lambda_i}\left(1 - \frac{q - c\delta}{1 - c}\lambda_i\right)^{j}\left[1 - \left(1 - 2\frac{q - c\delta}{1 - c}\lambda_i\right)^{t}\right],$$

where the second inequality holds due to Lemma E.2. If $j = 0$, then by (K.18)

$$\left(\sum_{k=0}^{t-1} \mathbf{A}_i^{j+k} \begin{bmatrix} \delta \\ q \end{bmatrix}\right)_1 \leq \frac{1 - x_2^t}{\lambda_i} \leq \frac{1}{\lambda_i}\left[1 - \left(1 - 2\frac{q - c\delta}{1 - c}\lambda_i\right)^{t}\right],$$

where the second inequality holds due to Lemma E.2. Thus, the upper bound also holds for $j = 0$. The lower bound of (K.7) is given by

$$\left(\sum_{k=0}^{t-1} \mathbf{A}_i^{j+k} \begin{bmatrix} \delta \\ q \end{bmatrix}\right)_1 = \frac{1}{\lambda_i}\left[(1 - \delta\lambda_i - x_2)\frac{x_2^j(1 - x_2^t) - x_1^j(1 - x_1^t)}{x_2 - x_1} + x_2^j(1 - x_2^t)\right]$$

$$
\begin{aligned}
&\geq \frac{1}{\lambda_i}\left[(1 - \delta\lambda_i - x_2)\frac{x_2^j(1 - x_2^t) - x_2^j(1 - x_1^t)}{x_2 - x_1} + x_2^j(1 - x_2^t)\right] \\
&= \frac{x_2^j}{\lambda_i}\left[-(1 - \delta\lambda_i - x_2)\frac{x_2^t - x_1^t}{x_2 - x_1} + (1 - x_2^t)\right] \\
&\geq \frac{x_2^j}{\lambda_i}\left[-(1 - \delta\lambda_i - x_2)\frac{1 - x_2^t}{1 - x_2} + (1 - x_2^t)\right] \\
&= \frac{\delta x_2^j(1 - x_2^t)}{1 - x_2} \geq 0,
\end{aligned}
$$

where the first inequality holds because $x_1 \leq x_2$, the second inequality holds due to Lemma K.12, and the third inequality holds because $0 < x_2 < 1$. $\qquad\square$

The following corollaries follow from Lemma K.5.

**Corollary K.6.** With $\mathbf{A}_i$ defined in (E.1), assuming that $N(1 - c) \geq 2$, we have

$$
\sum_{i=1}^d \lambda_i^2 \sum_{j=0}^{s-1}\left(\sum_{k=0}^{N-1}\mathbf{A}_i^{j+k}\begin{bmatrix}\delta \\ q\end{bmatrix}\right)_1^2 \leq 18Nk^* + \frac{36sN^2(q - c\delta)^2}{(1 - c)^2}\sum_{i>k^*}\lambda_i^2.
$$

*Proof.* By Lemma K.5, we have

(a) For $i \leq k^\ddagger$,

$$
\begin{aligned}
\sum_{j=0}^{s-1}\left(\sum_{k=0}^{N-1}\mathbf{A}_i^{j+k}\begin{bmatrix}\delta \\ q\end{bmatrix}\right)_1^2 &\leq \frac{4}{\lambda_i^2}\sum_{j=0}^{s-1}(c\delta/q)^{2j} \leq \frac{4}{\lambda_i^2}\sum_{j=0}^{s-1}c^j = \frac{4}{\lambda_i^2}\cdot\frac{1 - c^s}{1 - c} \\
&\leq \frac{4}{(1 - c)\lambda_i^2} \quad\quad\quad\quad\text{(K.22)} \\
&\leq \frac{2N}{\lambda_i^2},
\end{aligned}
$$

where the second inequality holds because $(c\delta/q)^2 \leq c^2 \leq c$, the third inequality holds because $1 - c^s \leq 1$, and the last inequality holds due to the assumption that $N(1 - c) \geq 2$.

(b) For $k^\ddagger < i \leq \widehat{k}$,

$$
\begin{aligned}
&\frac{3}{\lambda_i}[c(1 - \delta\lambda_i)]^{j/2} + \delta j[c(1 - \delta\lambda_i)]^{(j-1)/2} \\
&\leq \frac{3}{\lambda_i}[c(1 - \delta\lambda_i)]^{j/2} + 2\delta[c(1 - \delta\lambda_i)]^{j/4}\sum_{t=0}^{j/2-1}[c(1 - \delta\lambda_i)]^{t/2} \\
&= \frac{3}{\lambda_i}[c(1 - \delta\lambda_i)]^{j/2} + 2\delta\frac{[c(1 - \delta\lambda_i)]^{j/4} - [c(1 - \delta\lambda_i)]^{j/2}}{1 - \sqrt{c(1 - \delta\lambda_i)}} \\
&\leq \frac{3}{\lambda_i}[c(1 - \delta\lambda_i)]^{j/2} + 2\delta\frac{[c(1 - \delta\lambda_i)]^{j/4} - [c(1 - \delta\lambda_i)]^{j/2}}{\delta\lambda_i/2} \\
&= \frac{4[c(1 - \delta\lambda_i)]^{j/4} - [c(1 - \delta\lambda_i)]^{j/2}}{\lambda_i} \leq \frac{4}{\lambda_i}[c(1 - \delta\lambda_i)]^{j/4}, \quad\text{(K.23)}
\end{aligned}
$$

where the first inequality holds because $[c(1 - \delta\lambda_i)]^{j/4-1/2} \leq [c(1 - \delta\lambda_i)]^{t/2}$ for all $t \leq j/2 - 1$, the second inequality holds because $1 - \sqrt{c(1 - \delta\lambda_i)} \geq 1 - \sqrt{1 - \delta\lambda_i} \geq \delta\lambda_i/2$, and the last inequality holds because $[c(1 - \delta\lambda_i)]^{j/2} \geq 0$. We thus have

$$
\begin{aligned}
\sum_{j=0}^{s-1}\left(\sum_{k=0}^{N-1}\mathbf{A}_i^{j+k}\begin{bmatrix}\delta \\ q\end{bmatrix}\right)_1^2 &\leq \sum_{j=0}^{s-1}\left(\frac{3}{\lambda_i}[c(1 - \delta\lambda_i)]^{j/2} + \delta j[c(1 - \delta\lambda_i)]^{(j-1)/2}\right)^2 \\
&\leq \frac{16}{\lambda_i^2}\sum_{j=0}^{s-1}[c(1 - \delta\lambda_i)]^{j/2} = \frac{16}{\lambda_i^2}\cdot\frac{1 - [c(1 - \delta\lambda_i)]^{s/2}}{1 - \sqrt{c(1 - \delta\lambda_i)}}
\end{aligned}
$$

$$\leq \frac{16}{\lambda_i^2} \cdot \frac{1}{1 - \sqrt{c(1 - \delta\lambda_i)}} \leq \frac{16}{\lambda_i^2} \cdot \frac{1}{(1 - c)/2} = \frac{32}{(1 - c)\lambda_i^2} \tag{K.24}$$

$$\leq \frac{16N}{\lambda_i^2},$$

where the first inequality holds due to Lemma K.5, the second inequality holds due to (K.23), the third inequality holds because $1 - [c(1 - \delta\lambda_i)]^{s/2} \leq 1$, the fourth inequality holds because $1 - \sqrt{c(1 - \delta\lambda_i)} \geq 1 - \sqrt{c} \geq (1 - c)/2$, and the last inequality holds due to the assumption that $N(1 - c) \geq 2$.

(c) For $\widehat{k} < i \leq k^\dagger$,

$$\frac{3}{\lambda_i}[c(1 - \delta\lambda_i)]^{j/2} + \frac{1 - c}{\lambda_i} \cdot j[c(1 - \delta\lambda_i)]^{(j-1)/2}$$

$$\leq \frac{3}{\lambda_i}[c(1 - \delta\lambda_i)]^{j/2} + \frac{2(1 - c)}{\lambda_i}[c(1 - \delta\lambda_i)]^{j/4} \sum_{t=0}^{j/2-1} [c(1 - \delta\lambda_i)]^{t/2}$$

$$= \frac{3}{\lambda_i}[c(1 - \delta\lambda_i)]^{j/2} + \frac{2(1 - c)}{\lambda_i} \cdot \frac{[c(1 - \delta\lambda_i)]^{j/4} - [c(1 - \delta\lambda_i)]^{j/2}}{1 - \sqrt{c(1 - \delta\lambda_i)}}$$

$$\leq \frac{3}{\lambda_i}[c(1 - \delta\lambda_i)]^{j/2} + \frac{2(1 - c)}{\lambda_i} \cdot \frac{[c(1 - \delta\lambda_i)]^{j/4} - [c(1 - \delta\lambda_i)]^{j/2}}{(1 - c)/2}$$

$$= \frac{4[c(1 - \delta\lambda_i)]^{j/4} - [c(1 - \delta\lambda_i)]^{j/2}}{\lambda_i} \leq \frac{4}{\lambda_i}[c(1 - \delta\lambda_i)]^{j/4}, \tag{K.25}$$

where the first inequality holds because $[c(1 - \delta\lambda_i)]^{j/4-1/2} \leq [c(1 - \delta\lambda_i)]^{t/2}$ for all $t \leq j/2 - 1$, the second inequality holds because $1 - \sqrt{c(1 - \delta\lambda_i)} \geq 1 - \sqrt{c} \geq (1 - c)/2$, and the last inequality holds because $[c(1 - \delta\lambda_i)]^{j/2} \geq 0$. Due to the same deduction as that in part (b), we have

$$\sum_{j=0}^{s-1} \left( \sum_{k=0}^{t-1} \mathbf{A}_i^{j+k} \begin{bmatrix} \delta \\ q \end{bmatrix} \right)_1^2 \leq \frac{32}{(1 - c)\lambda_i} \tag{K.26}$$

$$\leq \frac{16N}{\lambda_i^2}.$$

(d) For $k^\dagger < i \leq k^*$,

$$\sum_{j=0}^{s-1} \left( \sum_{k=0}^{N-1} \mathbf{A}_i^{j+k} \begin{bmatrix} \delta \\ q \end{bmatrix} \right)_1^2$$

$$\leq \sum_{j=0}^{s-1} \frac{9}{\lambda_i^2} \left[ 1 - \left( 1 - 2\frac{q - c\delta}{1 - c}\lambda_i \right)^N \right]^2 \left( 1 - \frac{q - c\delta}{1 - c}\lambda_i \right)^{2j}$$

$$\leq \frac{9}{\lambda_i^2} \left[ 1 - \left( 1 - 2\frac{q - c\delta}{1 - c}\lambda_i \right)^N \right]^2 \sum_{j=0}^{s-1} \left( 1 - \frac{q - c\delta}{1 - c}\lambda_i \right)^j$$

$$= \frac{9(1 - c)}{(q - c\delta)\lambda_i^3} \left[ 1 - \left( 1 - 2\frac{q - c\delta}{1 - c}\lambda_i \right)^N \right]^2 \cdot \left[ 1 - \left( 1 - \frac{q - c\delta}{1 - c}\lambda_i \right)^s \right]$$

$$\leq \frac{9(1 - c)}{(q - c\delta)\lambda_i^3}$$

$$\leq \frac{9(1 - c)}{(q - c\delta)\lambda_i^2} \cdot \frac{2N(q - c\delta)}{1 - c} = \frac{18N}{\lambda_i^2},$$

where the second inequality holds because $\left(1 - \frac{q-c\delta}{1-c}\lambda_i\right)^{2j} \le \left(1 - \frac{q-c\delta}{1-c}\lambda_i\right)^j$, the third inequality holds because $1 - \left(1 - 2\frac{q-c\delta}{1-c}\lambda_i\right)^N \le 1$ and $1 - \left(1 - \frac{q-c\delta}{1-c}\lambda_i\right)^s \le 1$, and the last inequality holds because $\lambda_i \ge \frac{1-c}{2N(q-c\delta)}$ due to definition of $k^*$.

(e) For $i > k^*$,

$$
\sum_{j=0}^{s-1}\left(\sum_{k=0}^{N-1}\mathbf{A}_i^{j+k}\begin{bmatrix}\delta\\q\end{bmatrix}\right)_1^2 \le \sum_{j=0}^{s-1}\frac{9}{\lambda_i^2}\left[1-\left(1-2\frac{q-c\delta}{1-c}\lambda_i\right)^N\right]^2\left(1-\frac{q-c\delta}{1-c}\lambda_i\right)^{2j}
$$

$$
\le \frac{9}{\lambda_i^2}\cdot\left(2N\frac{q-c\delta}{1-c}\lambda_i\right)^2\sum_{j=0}^{s-1}1 = \frac{36sN^2(q-c\delta)^2}{(1-c)^2},
$$

where the second inequality holds because $1 - \left(1 - 2\frac{q-c\delta}{1-c}\lambda_i\right)^N \le 2N\frac{q-c\delta}{1-c}\lambda_i$ and $\left(1 - \frac{q-c\delta}{1-c}\lambda_i\right)^{2j} \le 1$.

Concluding all the above, we have

$$
\sum_{i=1}^{d}\lambda_i^2\sum_{j=0}^{s-1}\left(\sum_{k=0}^{N-1}\mathbf{A}_i^{j+k}\begin{bmatrix}\delta\\q\end{bmatrix}\right)_1^2
$$

$$
\le \sum_{i\le k^{\ddagger}}\lambda_i^2\cdot\frac{2N}{\lambda_i^2} + \sum_{k^{\ddagger}<i\le k^{\dagger}}\lambda_i^2\cdot\frac{16N}{\lambda_i^2} + \sum_{k^{\dagger}<i\le k^*}\lambda_i^2\cdot\frac{18N}{\lambda_i^2} + \sum_{i>k^*}\lambda_i^2\cdot\frac{36sN^2(q-c\delta)^2}{(1-c)^2}
$$

$$
= 2Nk^{\ddagger} + 16N(k^{\dagger}-k^{\ddagger}) + 18N(k^*-k^{\dagger}) + \frac{36sN^2(q-c\delta)^2}{(1-c)^2}\sum_{i>k^*}\lambda_i^2
$$

$$
\le 18Nk^* + \frac{36sN^2(q-c\delta)^2}{(1-c)^2}\sum_{i>k^*}\lambda_i^2,
$$

where the second inequality holds because all coefficients $2, 16, 18$ are bounded by $18$. □

**Corollary K.7.** With $\mathbf{A}_i$ defined in (E.1), we have for all $j \ge 0$,

$$
\sum_{i=1}^{d}\lambda_i^2\left(\sum_{k=0}^{N-1}\mathbf{A}_i^{j+k}\begin{bmatrix}\delta\\q\end{bmatrix}\right)_1^2 \le 9k^* + \frac{36(q-c\delta)^2}{(1-c)^2}\sum_{i>k^*}\lambda_i^2.
$$

*Proof.* By Lemma K.5, we have

(a) For $i \le k^{\ddagger}$,

$$
\left(\sum_{k=0}^{N-1}\mathbf{A}_i^{j+k}\begin{bmatrix}\delta\\q\end{bmatrix}\right)_1^2 \le \frac{4}{\lambda_i^2}(c\delta/q)^{2j} \le \frac{4}{\lambda_i^2},
$$

where the second inequality holds because $c\delta/q \le 1$.

(b) For $k^{\ddagger} < i \le \widehat{k}$,

$$
\frac{3}{\lambda_i}[c(1-\delta\lambda_i)]^{j/2} + \delta j[c(1-\delta\lambda_i)]^{(j-1)/2}
$$

$$
\le \frac{3}{\lambda_i}[c(1-\delta\lambda_i)]^{j/2} + \delta\cdot\sum_{t=0}^{j-1}[c(1-\delta\lambda_i)]^{t/2}
$$

$$
= \frac{3}{\lambda_i}[c(1-\delta\lambda_i)]^{j/2} + \delta\cdot\frac{1-[c(1-\delta\lambda_i)]^{j/2}}{1-\sqrt{c(1-\delta\lambda_i)}}
$$

$$
\le \frac{3}{\lambda_i}[c(1-\delta\lambda_i)]^{j/2} + \delta\cdot\frac{1-[c(1-\delta\lambda_i)]^{j/2}}{\delta\lambda_i/2}
$$

$$= \frac{2 + [c(1 - \delta\lambda_i)]^{j/2}}{\lambda_i} \leq \frac{3}{\lambda_i}, \tag{K.27}$$

where the first inequality holds because $[c(1 - \delta\lambda_i)]^{(j-1)/2} \leq [c(1 - \delta\lambda_i)]^{t/2}$ for $t \leq j-1$, the second inequality holds because $1 - \sqrt{c(1 - \delta\lambda_i)} \geq 1 - \sqrt{1 - \delta\lambda_i} \geq \delta\lambda_i/2$, and the last inequality holds because $[c(1 - \delta\lambda_i)]^{j/2} \leq 1$. Therefore,

$$\left( \sum_{k=0}^{N-1} \mathbf{A}_i^{j+k} \begin{bmatrix} \delta \\ q \end{bmatrix} \right)_1^2 \leq \left( \frac{3}{\lambda_i} [c(1 - \delta\lambda_i)]^{j/2} + \delta j [c(1 - \delta\lambda_i)]^{(j-1)/2} \right)^2 \leq \frac{9}{\lambda_i^2},$$

where the first inequality hold due to Lemma K.5, and the second inequality holds due to (K.27).

(c) For $\widehat{k} < i \leq k^\dagger$,

$$\frac{3}{\lambda_i} [c(1 - \delta\lambda_i)]^{j/2} + \frac{1-c}{\lambda_i} \cdot j [c(1 - \delta\lambda_i)]^{(j-1)/2}$$

$$\leq \frac{3}{\lambda_i} [c(1 - \delta\lambda_i)]^{j/2} + \frac{1-c}{\lambda_i} \cdot \sum_{t=0}^{j-1} [c(1 - \delta\lambda_i)]^{t/2}$$

$$= \frac{3}{\lambda_i} [c(1 - \delta\lambda_i)]^{j/2} + \frac{1-c}{\lambda_i} \cdot \frac{1 - [c(1 - \delta\lambda_i)]^{j/2}}{1 - \sqrt{c(1 - \delta\lambda_i)}}$$

$$\leq \frac{3}{\lambda_i} [c(1 - \delta\lambda_i)]^{j/2} + \frac{1-c}{\lambda_i} \cdot \frac{1 - [c(1 - \delta\lambda_i)]^{j/2}}{(1 - c)/2}$$

$$= \frac{2 + [c(1 - \delta\lambda_i)]^{j/2}}{\lambda_i} \leq \frac{3}{\lambda_i}, \tag{K.28}$$

where the first inequality holds because $[c(1 - \delta\lambda_i)]^{(j-1)/2} \leq [c(1 - \delta\lambda_i)]^{t/2}$ for $t \leq j-1$, the second inequality holds because $1 - \sqrt{c(1 - \delta\lambda_i)} \geq 1 - \sqrt{c} \geq (1 - c)/2$, and the last inequality holds because $[c(1 - \delta\lambda_i)]^{j/2} \leq 1$. Therefore

$$\left( \sum_{k=0}^{N-1} \mathbf{A}_i^{j+k} \begin{bmatrix} \delta \\ q \end{bmatrix} \right)_1^2 \leq \left( \frac{3}{\lambda_i} [c(1 - \delta\lambda_i)]^{j/2} + \frac{1-c}{\lambda_i} \cdot j [c(1 - \delta\lambda_i)]^{(j-1)/2} \right)^2 \leq \frac{9}{\lambda_i^2},$$

where the first inequality holds due to Lemma K.5, and the second inequality holds due to (K.28).

(d) For $i > k^*$, we have

$$\left( \sum_{k=0}^{N-1} \mathbf{A}_i^{j+k} \begin{bmatrix} \delta \\ q \end{bmatrix} \right)_1^2 \leq \frac{9}{\lambda_i^2} \left[ 1 - \left( 1 - 2\frac{q - c\delta}{1-c}\lambda_i \right)^N \right]^2 \left( 1 - \frac{q - c\delta}{1-c} \right)^{2j}$$

$$\leq \frac{9}{\lambda_i^2} \left[ 1 - \left( 1 - 2\frac{q - c\delta}{1-c}\lambda_i \right)^N \right]^2$$

$$\leq \min \left\{ \frac{9}{\lambda_i^2}, \frac{36N^2(q - c\delta)^2}{(1-c)^2} \right\},$$

where the second inequality holds because $1 - \frac{q - c\delta}{1-c}\lambda_i \leq 1$, and the last inequality holds because $1 - (1-r)^N \leq 1$ and $1 - (1-r)^N \leq rN$ for all $r \in (0, 1)$.

Combining all the above, we have

$$\sum_{i=1}^{d} \lambda_i^2 \left( \sum_{k=0}^{N-1} \mathbf{A}_i^{j+k} \begin{bmatrix} \delta \\ q \end{bmatrix} \right)_1^2$$

$$\leq \sum_{i < k^\ddagger} \lambda_i^2 \cdot \frac{4}{\lambda_i^2} + \sum_{k^\ddagger < i \leq k^\dagger} \lambda_i^2 \cdot \frac{9}{\lambda_i^2} + \sum_{k^\dagger < i \leq k^*} \lambda_i^2 \cdot \frac{9}{\lambda_i^2} + \sum_{i > k^*} \lambda_i^2 \cdot \frac{36N^2(q - c\delta)^2}{(1-c)^2}$$

$$= 4k^{\ddagger} + 9(k^{\dagger} - k^{\ddagger}) + 9(k^* - k^{\dagger}) + \frac{36N^2(q - c\delta)^2}{(1 - c)^2} \sum_{i > k^*} \lambda_i^2,$$

$$\leq 9k^* + \frac{36N^2(q - c\delta)^2}{(1 - c)^2} \sum_{i > k^*} \lambda_i^2,$$

where the first inequality holds because the bound $\frac{9}{\lambda_i^2}$ is applied for $k^{\dagger} < i \leq k^*$, while the upper bound $\frac{36N^2(q-c\delta)^2}{(1-c)^2}$ is applied for $i > k^*$, and the second inequality holds because coefficient $4, 9$ can be bounded by $9$. $\qquad\square$

**Lemma K.8.** With $\mathbf{A}_i$ defined as in (E.1), let $x_1$ and $x_2$ be the eigenvalues of $\mathbf{A}_i$ defined in (E.2) and (E.3). Then

- For all $i \leq k^{\ddagger}$, we have

$$-\frac{4}{\delta\lambda_i}(c\delta/q)^j \leq \left(\sum_{k=0}^{t-1} \mathbf{A}_i^{j+k} \begin{bmatrix} 1 \\ 1 \end{bmatrix}\right)_1 \leq \frac{2}{\delta\lambda_i}(c\delta/q)^j.$$

- For all $k^{\ddagger} < i \leq \widehat{k}$, we have

$$\left|\left(\sum_{k=0}^{t-1} \mathbf{A}_i^{j+k} \begin{bmatrix} 1 \\ 1 \end{bmatrix}\right)_1\right| \leq 2j[c(1 - \delta\lambda_i)]^{j/2} + \frac{4}{\delta\lambda_i}[c(1 - \delta\lambda_i)]^{j/2}.$$

- For all $\widehat{k} < i \leq k^{\dagger}$, we have

$$\left|\left(\sum_{k=0}^{t-1} \mathbf{A}_i^{j+k} \begin{bmatrix} 1 \\ 1 \end{bmatrix}\right)_1\right| \leq 2j[c(1 - \delta\lambda_i)]^{j/2} + \frac{10}{1-c}[c(1 - \delta\lambda_i)]^{j/2}.$$

- For all $i > k^{\dagger}$, we have

$$0 \leq \left(\sum_{k=0}^{t-1} \mathbf{A}_i^{j+k} \begin{bmatrix} 1 \\ 1 \end{bmatrix}\right)_1 \leq \frac{3(1-c)}{(q-c\delta)\lambda_i}\left[1 - \left(1 - 2\frac{q-c\delta}{1-c}\lambda_i\right)^t\right]\left(1 - \frac{q-c\delta}{1-c}\lambda_i\right)^j.$$

*Proof.* Note that

$$\sum_{k=0}^{t-1} \mathbf{A}_i^{j+k} \begin{bmatrix} 1 \\ 1 \end{bmatrix} = (\mathbf{A}_i^j - \mathbf{A}_i^{j+t})(\mathbf{I} - \mathbf{A}_i)^{-1} \begin{bmatrix} 1 \\ 1 \end{bmatrix} = (\mathbf{A}_i^j - \mathbf{A}_i^{j+t}) \cdot \frac{1}{(q-c\delta)\lambda_i}\begin{bmatrix} 1 - c + (q-\delta)\lambda_i \\ 1 - c \end{bmatrix}$$

$$= \frac{1}{(q-c\delta)\lambda_i}\begin{bmatrix} (1 - c + (q-\delta)\lambda_i)((\mathbf{A}_i^j)_{11} - (\mathbf{A}_i^{j+t})_{11}) + (1-c)(\mathbf{A}_i^j)_{12} - (\mathbf{A}_i^{j+t})_{12}) \\ (1 - c + (q-\delta)\lambda_i)((\mathbf{A}_i^j)_{21} - (\mathbf{A}_i^{j+t})_{21}) + (1-c)((\mathbf{A}_i^j)_{22} - (\mathbf{A}_i^{j+t})_{22}). \end{bmatrix}$$

$$\tag{K.29}$$

Combine (K.29) with Lemma E.3, and we have

$$\left(\sum_{k=0}^{t-1} \mathbf{A}_i^{j+k} \begin{bmatrix} 1 \\ 1 \end{bmatrix}\right)_1 = \frac{1}{(q-c\delta)\lambda_i}\left[-(1 - c + (q-\delta)\lambda_i) \cdot \frac{x_1 x_2^j(1 - x_2^t) - x_2 x_1^j(1 - x_1^t)}{x_2 - x_1}\right.$$

$$\left. + (1-c)(1 - \delta\lambda_i)\frac{x_2^j(1 - x_2^t) - x_1^j(1 - x_1^t)}{x_2 - x_1}\right]$$

$$= \frac{1}{(q-c\delta)\lambda_i}\left\{\frac{[(1-c)(1 - \delta\lambda_i) - (1 - c + (q-\delta)\lambda_i)x_1]x_2^j(1 - x_2^t)}{x_1 - x_2}\right.$$

$$\left. - \frac{[(1-c)(1 - \delta\lambda_i) - (1 - c + (q-\delta)\lambda_i)x_2]x_1^j(1 - x_1^t)}{x_2 - x_1}\right\}. \tag{K.30}$$

For $i \leq k^{\ddagger}$, note that

$$(1-c)(1 - \delta\lambda_i) - (1 - c + (q-\delta)\lambda_i)x_1 = -(1-c)(x_1 + \delta\lambda_i - 1) - (q-\delta)\lambda_i x_1 \leq 0$$

due to (K.8) and $q - \delta \geq 0$. The upper bound of (K.30) is thus given by

$$\left(\sum_{k=0}^{t-1} \mathbf{A}_i^{j+k} \begin{bmatrix} 1 \\ 1 \end{bmatrix}\right)_1 \leq \frac{1}{(q - c\delta)\lambda_i} \left\{ \frac{[(1-c)(1-\delta\lambda_i) - (1 - c + (q-\delta)\lambda_i)x_1]x_1^j(1 - x_2^t)}{x_1 - x_2} \right.$$

$$\left. - \frac{[(1-c)(1-\delta\lambda_i) - (1 - c + (q-\delta)\lambda_i)x_2]x_1^j(1 - x_1^t)}{x_2 - x_1} \right\}$$

$$= \frac{x_1^j}{(q - c\delta)\lambda_i} \left\{ (1 - c + (q - \delta)\lambda_i)(1 - x_1^t) \right.$$

$$\left. + [(1 - c + (q - \delta)\lambda_i)x_1 - (1 - c)(1 - \delta\lambda_i)] \cdot \frac{x_2^t - x_1^t}{x_2 - x_1} \right\}$$

$$\leq \frac{x_1^j}{(q - c\delta)\lambda_i} \left\{ (1 - c + (q - \delta)\lambda_i)(1 - x_1^t) \right.$$

$$\left. + [(1 - c + (q - \delta)\lambda_i)x_1 - (1 - c)(1 - \delta\lambda_i)] \cdot \frac{1 - x_1^t}{1 - x_1} \right\}$$

$$= \frac{x_1^j(1 - x_1^t)}{(q - c\delta)\lambda_i} \cdot \frac{(q - c\delta)\lambda_i}{1 - x_1} = \frac{x_1^j(1 - x_1^t)}{1 - x_1}$$

$$\leq \frac{2}{\delta\lambda_i}(c\delta/q)^j,$$

where the first inequality holds because $x_2 \geq x_1$, the second inequality holds due to Lemma K.12, and the last inequality holds because $x_1 \leq x_2 \leq c\delta/q$, $1 - x_1^t \leq 1$ and (K.10). The lower bound of (K.30) is given by

$$\left(\sum_{k=0}^{t-1} \mathbf{A}_i^{j+k} \begin{bmatrix} 1 \\ 1 \end{bmatrix}\right)_1 \geq \frac{1}{(q - c\delta)\lambda_i} \left\{ \frac{-[(1 - c + (q-\delta)\lambda_i)x_1 - (1-c)(1-\delta\lambda_i)]x_2^j(1 - x_2^t)}{x_1 - x_2} \right.$$

$$\left. - \frac{[(1 - c + (q-\delta)\lambda_i)x_2 - (1-c)(1-\delta\lambda_i)]x_1^j(1 - x_2^t)}{x_2 - x_1} \right\}$$

$$= \frac{1 - x_2^t}{(q - c\delta)\lambda_i} \left\{ (1 - c)(1 - \delta\lambda_i)x_1^{j-1} \right.$$

$$\left. - [(1 - c + (q - \delta)\lambda_i)x_1 - (1 - c)(1 - \delta\lambda_i)]x_2 \cdot \frac{x_2^{j-1} - x_1^{j-1}}{x_2 - x_1} \right\}, \quad \text{(K.31)}$$

where the inequality holds because $x_1 \leq x_2$. If $j \geq 1$, then

$$(1 - c)(1 - \delta\lambda_i)x_1^{j-1} - [(1 - c + (q - \delta)\lambda_i)x_1 - (1 - c)(1 - \delta\lambda_i)]x_2 \cdot \frac{x_2^{j-1} - x_1^{j-1}}{x_2 - x_1}$$

$$\geq \frac{(1 - c)(1 - \delta\lambda_i)x_2 - (1 - c + (q - \delta)\lambda_i)x_1 x_2}{c\delta/q - x_1} \cdot (c\delta/q)^{j-1}$$

$$= \frac{(1 - c)(1 - \delta\lambda_i)x_2 - (1 - c + (q - \delta)\lambda_i) \cdot c(1 - \delta\lambda_i)}{c\delta/q - x_1} \cdot (c\delta/q)^{j-1}$$

$$= -(1 - \delta\lambda_i)\frac{(1 - c)(c - x_2) + c(q - \delta)\lambda_i}{c\delta/q - x_1} \cdot (c\delta/q)^{j-1}, \quad \text{(K.32)}$$

where the ineuqality holds because $(1 - c)(1 - \delta\lambda_i)x_1^{j-1} \geq 0$ and due to Lemma K.12. Note that

$$(1 - \delta\lambda_i)\frac{(1 - c)(c - x_2) + c(q - \delta)\lambda_i}{c\delta/q - x_1}$$

$$= (1 - \delta\lambda_i)\frac{(1 - c)\frac{-1 + c + q\lambda_i - \sqrt{(1 + c - q\lambda_i)^2 - 4c(1 - \delta\lambda_i)}}{2} + c(q - \delta)\lambda_i}{c\delta/q - \frac{1 + c - q\lambda_i - \sqrt{(1 + c - q\lambda_i)^2 - 4c(1 - \delta\lambda_i)}}{2}}$$

$$\leq (1 - \delta\lambda_i)\frac{(1-c)\frac{-1+c+q\lambda_i}{2} + c(q-\delta)\lambda_i}{c\delta/q - \frac{1+c-q\lambda_i}{2}} = q(1-\delta\lambda_i)\frac{[(1+c)q - 2c\delta]\lambda_i - (1-c)^2}{q^2\lambda_i - [(1+c)q - 2c\delta]}$$

$$\leq q\left[1 - \delta \cdot \frac{(\sqrt{q-c\delta} + \sqrt{c(q-\delta)})^2}{q^2}\right] \cdot \frac{[(1+c)q - 2c\delta]\frac{(\sqrt{q-c\delta} + \sqrt{c(q-\delta)})^2}{q^2} - (1-c)^2}{(\sqrt{q-c\delta} + \sqrt{c(q-\delta)})^2 - [(1+c)q - 2c\delta]}$$

$$= \frac{(c\delta - \sqrt{c(q-\delta)(q-c\delta)})^2(\sqrt{q-c\delta} + \sqrt{c(q-\delta)})^2}{cq^3}$$

$$\leq \frac{(c\delta)^2 \cdot 4(q-c\delta)}{cq^3} \leq \frac{4c(q-c\delta)}{q}, \tag{K.33}$$

where the first inequality holds because $\sqrt{(1+c-q\lambda_i)^2 - 4c(1-\delta\lambda_i)} \geq 0$, the second inequality holds due to (E.6), the third inequality holds because $c\delta - \sqrt{c(q-\delta)(q-c\delta)} \leq c\delta$ and $c(q-\delta) \leq q - c\delta$, and the last inequality holds because $\delta \leq q$. Combining (K.33) with (K.32) and (K.31), we have

$$\left(\sum_{k=0}^{t-1}\mathbf{A}_i^{j+k}\begin{bmatrix}1\\1\end{bmatrix}\right)_1 \geq -\frac{1-x_2^t}{(q-c\delta)\lambda_i} \cdot \frac{4c(q-c\delta)}{q}(c\delta/q)^{j-1} \geq -\frac{4}{\delta\lambda_i}(c\delta/q)^j,$$

where the second inequality holds because $1 - x_2^t \leq 1$.

For $k^\ddagger < i \leq k^\dagger$, i.e., $\mathbf{A}_i$ has complex eigenvalues $x_1, x_2$, we have

$$\left|\left(\sum_{k=0}^{t-1}\mathbf{A}_i^{j+k}\begin{bmatrix}1\\1\end{bmatrix}\right)_1\right| = \frac{1}{(q-c\delta)\lambda_i}\left|(1-c)(1-\delta\lambda_i)x_2^{j-1}(1-x_2^t)\right.$$

$$+ \left.[(1-c)(1-\delta\lambda_i) - (1 - c + (q-\delta)\lambda_i)x_2] \cdot x_1 \cdot \frac{(x_2^{j-1} - x_1^{j-1})(1-x_2^t) - x_1^{j-1}(x_2^t - x_1^t)}{x_2 - x_1}\right|$$

$$\leq \frac{1}{(q-c\delta)\lambda_i}|(1-c)(1-\delta\lambda_i) - (1 - c + (q-\delta)\lambda_i)x_2| \cdot |x_1|$$

$$\cdot \left[\left|\frac{x_2^{j-1} - x_1^{j-1}}{x_2 - x_1}\right| \cdot |1 - x_2^t| + |x_1^{j-1}| \cdot \left|\frac{x_2^t - x_1^t}{x_2 - x_1}\right|\right] + \frac{(1-c)(1-\delta\lambda_i)}{(q-c\delta)\lambda_i} \cdot |x_2^{j-1}| \cdot |1 - x_2^t|, \tag{K.34}$$

where the inequality holds due to triangle inequality. Note that

$$|(1-c)(1-\delta\lambda_i) - (1 - c + (q-\delta)\lambda_i)x_2|$$
$$= \sqrt{[(1-c)(1-\delta\lambda_i) - (1 - c + (q-\delta)\lambda_i)x_2][(1-c)(1-\delta\lambda_i) - (1 - c + (q-\delta)\lambda_i)x_1]},$$

where

$$[(1-c)(1-\delta\lambda_i) - (1 - c + (q-\delta)\lambda_i)x_2][(1-c)(1-\delta\lambda_i) - (1 - c + (q-\delta)\lambda_i)x_1]$$
$$= (1-c)^2(1-\delta\lambda_i)^2 - (1-c)(1-\delta\lambda_i)(1 - c + (q-\delta)\lambda_i)(x_1 + x_2)$$
$$\quad + (1 - c + (q-\delta)\lambda_i)^2 \cdot x_1 x_2$$
$$= (1-c)^2(1-\delta\lambda_i)^2 - (1-c)(1-\delta\lambda_i)(1 - c + (q-\delta)\lambda_i)(1 + c - q\lambda_i)$$
$$\quad + (1 - c + (q-\delta)\lambda_i)^2 \cdot c(1-\delta\lambda_i)$$
$$= (1-\delta\lambda_i)(q-\delta)(q-c\delta)\lambda_i^2$$
$$\leq c(1-\delta\lambda_i)(q-c\delta)^2\lambda_i^2,$$

where the inequality holds because $q - \delta \leq c(q - c\delta)$. Therefore,

$$|(1-c)(1-\delta\lambda_i) - (1 - c + (q-\delta)\lambda_i)x_2| \leq (q-c\delta)\lambda_i\sqrt{c(1-\delta\lambda_i)}. \tag{K.35}$$

(K.34) can thus be further bounded by

$$\left|\left(\sum_{k=0}^{t-1}\mathbf{A}_i^{j+k}\begin{bmatrix}1\\1\end{bmatrix}\right)_1\right|$$

$$\leq \sqrt{c(1-\delta\lambda_i)} \cdot |x_1| \cdot \left[ 2 \left| \frac{x_2^{j-1} - x_1^{j-1}}{x_2 - x_1} \right| + |x_1|^{j-1} \cdot \left| \frac{x_2^t - x_1^t}{x_2 - x_1} \right| \right] + \frac{2(1-c)(1-\delta\lambda_i)}{(q-c\delta)\lambda_i} \cdot |x_2^{j-1}|$$

$$\leq \left\{ t[c(1-\delta\lambda_i)]^{t/2} + \frac{2(1-c)}{(q-c\delta)\lambda_i} \sqrt{\frac{1-\delta\lambda_i}{c}} + 2(j-1) \right\} \cdot [c(1-\delta\lambda_i)]^{j/2}, \tag{K.36}$$

where the first inequality holds because $|1 - x_2^t| \leq 2$ and due to (K.33), and the second inequality holds due to Lemma E.2 and Lemma K.13.

For $k^{\ddagger} < i \leq \hat{k}$, we can further bound (K.36) as

$$\left| \left( \sum_{k=0}^{t-1} \mathbf{A}_i^{j+k} \begin{bmatrix} 1 \\ 1 \end{bmatrix} \right)_1 \right| \leq \left[ \frac{2\sqrt{c(1-\delta\lambda_i)}}{\delta\lambda_i} + \frac{2(1-c)}{(q-c\delta)\lambda_i} \cdot \sqrt{\frac{1-\delta\lambda_i}{c}} + 2(j-1) \right] [c(1-\delta\lambda_i)]^{j/2}$$

$$\leq \left[ \frac{2}{\delta\lambda_i} + \frac{2}{\delta\lambda_i} \cdot 1 + 2j \right] [c(1-\delta\lambda_i)]^{j/2}$$

$$= 2j[c(1-\delta\lambda_i)]^{j/2} + \frac{4}{\delta\lambda_i}[c(1-\delta\lambda_i)]^{j/2},$$

where the first inequality holds due to Lemma K.14, and the second inequality holds because $\sqrt{c(1-\delta\lambda_i)} \leq 1$, $(1-c)/(q-c\delta) \leq 1/\delta$, $1-\delta\lambda_i \leq c$ and $j-1 < j$.

For $\hat{k} < i \leq k^{\dagger}$, note that

$$\frac{1-c}{(q-c\delta)\lambda_i} \sqrt{\frac{1-\delta\lambda_i}{c}}$$

$$\leq \frac{1-c}{\sqrt{c}(q-c\delta)} \cdot \frac{(\sqrt{q-c\delta} + \sqrt{c(q-\delta)})^2}{(1-c)^2} \cdot \sqrt{1 - \frac{\delta(1-c)^2}{(\sqrt{q-c\delta} + \sqrt{c(q-\delta)})^2}}$$

$$= \frac{(\sqrt{q-c\delta} + \sqrt{c(q-\delta)})(\sqrt{q-\delta} + \sqrt{c(q-c\delta)})}{\sqrt{c}(1-c)(q-c\delta)}$$

$$\leq \frac{(1+c)\sqrt{q-c\delta} \cdot 2\sqrt{c(q-c\delta)}}{\sqrt{c}(1-c)(q-c\delta)}$$

$$\leq \frac{4}{1-c}, \tag{K.37}$$

where the first inequality holds due to (E.5), the second inequality holds because $q - \delta \leq c(q-c\delta)$, and the third inequality holds because $c \leq 1$. Therefore, (K.36) can be further bounded by

$$\left| \left( \sum_{k=0}^{t-1} \mathbf{A}_i^{j+k} \begin{bmatrix} 1 \\ 1 \end{bmatrix} \right)_1 \right| \leq \left[ \frac{2\sqrt{c(1-\delta\lambda_i)}}{1-c} + \frac{2(1-c)}{(q-c\delta)\lambda_i} \sqrt{\frac{1-\delta\lambda_i}{c}} + 2(j-1) \right] [c(1-\delta\lambda_i)]^{j/2}$$

$$\leq \left[ \frac{2}{1-c} + \frac{8}{1-c} + 2j \right] [c(1-\delta\lambda_i)]^{j/2}$$

$$= 2j[c(1-\delta\lambda_i)]^{j/2} + \frac{10}{1-c}[c(1-\delta\lambda_i)]^{j/2},$$

where the first inequality holds due to Lemma K.12, and the second inequality holds because $\sqrt{c(1-\delta\lambda_i)} \leq 1$, $j-1 < j$ and due to (K.37).

For $j = 0$ and $t \geq 1$, we have

$$\left| \left( \sum_{k=0}^{t-1} \mathbf{A}_i^{j+k} \begin{bmatrix} 1 \\ 1 \end{bmatrix} \right)_1 \right|$$

$$= \frac{1}{(q-c\delta)\lambda_i} \left| \frac{[(1-c)(1-\delta\lambda_i) - (1-c+(q-\delta)\lambda_i)x_1](1-x_2^t)}{x_2 - x_1} \right.$$

$$\left. - \frac{[(1-c)(1-\delta\lambda_i) - (1-c+(q-\delta)\lambda_i)x_2](1-x_1^t)}{x_2 - x_1} \right|$$

$$= \frac{1}{(q - c\delta)\lambda_i} \left| (1 - c + (q - \delta)\lambda_i) - (1 - c)(1 - \delta\lambda_i)x_1^{t-1} \right.$$

$$\left. - [(1 - c)(1 - \delta\lambda_i) - (1 - c + (q - \delta)\lambda_i)x_1]x_2 \cdot \frac{x_2^{t-1} - x_1^{t-1}}{x_2 - x_1} \right|$$

$$\leq \frac{1}{(q - c\delta)\lambda_i} \left[ (1 - c) + (q - \delta)\lambda_i + (1 - c)(1 - \delta\lambda_i)|x_1^{t-1}| \right.$$

$$\left. + |(1 - c)(1 - \delta\lambda_i) - (1 - c + (q - \delta)\lambda_i)x_1| \cdot |x_2| \cdot \left| \frac{x_2^{t-1} - x_1^{t-1}}{x_2 - x_1} \right| \right]$$

$$\leq \frac{1 - c}{(q - c\delta)\lambda_i} + \frac{q - \delta}{q - c\delta} + \frac{(1 - c)(1 - \delta\lambda_i)}{(q - c\delta)\lambda_i} [c(1 - \delta\lambda_i)]^{(t-1)/2}$$

$$+ c(1 - \delta\lambda_i) \cdot (t - 1)[c(1 - \delta\lambda_i)]^{(t-2)/2}, \tag{K.38}$$

where the first inequality holds due to triangle inequality, and the second inequality holds due to Lemma K.13. When $k^{\ddagger} < i \leq \widehat{k}$, (K.38) can be further bounded by

$$\left| \left( \sum_{k=0}^{t-1} \mathbf{A}_i^{j+k} \begin{bmatrix} 1 \\ 1 \end{bmatrix} \right)_1 \right|$$

$$\leq \frac{1 - c}{(q - c\delta)\lambda_i} + \frac{q - \delta}{q - c\delta} + \frac{(1 - c)(1 - \delta\lambda_i)}{(q - c\delta)\lambda_i} + \frac{c(1 - \delta\lambda_i)}{\delta\lambda_i}$$

$$\leq \frac{1 - c}{(q - c\delta)\lambda_i} + 1 + \frac{1 - c}{(q - c\delta)\lambda_i} + \frac{1}{\delta\lambda_i}$$

$$\leq \frac{1}{\delta\lambda_i} + \frac{1}{\delta\lambda_i} + \frac{1}{\delta\lambda_i} + \frac{1}{\delta\lambda_i} = \frac{4}{\delta\lambda_i},$$

where the first inequality holds due to Lemma K.14, the second inequality holds because $(q - c\delta)/(1 - c) \geq \delta$, $1 - \delta\lambda_i \leq 1$ and $c \leq 1$, and the last inequality holds because $\frac{q - c\delta}{1 - c} \geq \delta$ and $\delta\lambda_i \leq 1$. When $\widehat{k} < i \leq k^{\dagger}$, (K.38) can be further bounded by

$$\left| \left( \sum_{k=0}^{t-1} \mathbf{A}_i^{j+k} \begin{bmatrix} 1 \\ 1 \end{bmatrix} \right)_1 \right|$$

$$\leq \frac{1 - c}{(q - c\delta)\lambda_i} + \frac{q - \delta}{q - c\delta} + \frac{(1 - c)(1 - \delta\lambda_i)}{(q - c\delta)\lambda_i} + \frac{c(1 - \delta\lambda_i)}{1 - c}$$

$$\leq \frac{1 - c}{(q - c\delta)\lambda_i} + 1 + \frac{1 - c}{(q - c\delta)\lambda_i} + \frac{1}{1 - c}$$

$$\leq \frac{2(1 - c)}{(q - c\delta)} \cdot \frac{(\sqrt{q - c\delta} + \sqrt{c(q - \delta)})^2}{(1 - c)^2} + \frac{1}{1 - c} + \frac{1}{1 - c}$$

$$= \frac{2}{1 - c} \cdot \left( 1 + \sqrt{\frac{c(q - \delta)}{q - c\delta}} \right)^2 + \frac{2}{1 - c}$$

$$\leq \frac{2}{1 - c} \cdot (1 + 1)^2 + \frac{2}{1 - c} = \frac{10}{1 - c},$$

where the first inequality holds due to Lemma K.12, the second inequality holds because $q - \delta \leq q - c\delta$, $1 - \delta\lambda_i \leq 1$ and $c \leq 1$, the third inequality holds due to (E.5), and the last inequality holds because $c(q - \delta) \leq q - c\delta$. Therefore, the upper bounds hold for $j = 0$.

For $i > k^{\dagger}$, note that

$$(1 - c)(1 - \delta\lambda_i) - (1 - c + (q - \delta)\lambda_i)x_2$$

$$\geq (1 - c)(1 - \delta\lambda_i) - (1 - c + (q - \delta)\lambda_i) \left( 1 - \frac{q - c\delta}{1 - c} \cdot \lambda_i \right)$$

$$= \frac{(q-\delta)(q-c\delta)}{1-c}\lambda_i^2 \geq 0, \tag{K.39}$$

where the first inequality holds due to Lemma E.2, and the second inequality holds because $q-c\delta \geq q - \delta \geq 0$. We thus have

$$\left(\sum_{k=0}^{t-1}\mathbf{A}_i^{j+k}\begin{bmatrix}1\\1\end{bmatrix}\right)_1 \leq \frac{1}{(q-c\delta)\lambda_i}\left\{\frac{[(1-c)(1-\delta\lambda_i)-(1-c+(q-\delta)\lambda_i)x_1]x_2^j(1-x_2^t)}{x_1-x_2}\right.$$

$$\left.-\frac{[(1-c)(1-\delta\lambda_i)-(1-c+(q-\delta)\lambda_i)x_2]x_1^j(1-x_2^t)}{x_2-x_1}\right\}$$

$$= \frac{1-x_2^t}{(q-c\delta)\lambda_i}\left\{(1-c)(1-\delta\lambda_i)x_2^{j-1}\right.$$

$$\left.+ [(1-c)(1-\delta\lambda_i)-(1-c+(q-\delta)\lambda_i)x_2]x_1 \cdot \frac{x_2^{j-1}-x_1^{j-1}}{x_2-x_1}\right\}, \tag{K.40}$$

where the inequality holds due to (K.39) and $x_1 \leq x_2$. If $j \geq 1$, (K.40) is further bounded by

$$(1-c)(1-\delta\lambda_i)x_2^{j-1} + [(1-c)(1-\delta\lambda_i)-(1-c+(q-\delta)\lambda_i)x_2]x_1 \cdot \frac{x_2^{j-1}-x_1^{j-1}}{x_2-x_1}$$

$$\leq \left(1-\frac{q-c\delta}{1-c}\lambda_i\right)^{j-1}\left[(1-c)(1-\delta\lambda_i) + \frac{(1-c)(1-\delta\lambda_i)-(1-c+(q-\delta)\lambda_i)x_2}{1-\frac{q-c\delta}{1-c}\lambda_i - x_1} \cdot x_1\right]$$

$$= \left(1-\frac{q-c\delta}{1-c}\lambda_i\right)^{j-1}$$

$$\cdot \left[(1-c)(1-\delta\lambda_i) + \frac{(1-c)(1-\delta\lambda_i)x_1-(1-c+(q-\delta)\lambda_i)\cdot c(1-\delta\lambda_i)}{1-\frac{q-c\delta}{1-c}\lambda_i - x_1}\right]$$

$$= \left(1-\frac{q-c\delta}{1-c}\lambda_i\right)^{j} \cdot \frac{1-\delta\lambda_i}{1-\frac{q-c\delta}{1-c}\lambda_i} \cdot \frac{(1-c)^2-[(1+c)q-2c\delta]\lambda_i}{1-\frac{q-c\delta}{1-c}\lambda_i - x_1}, \tag{K.41}$$

where the inequality holds due to Lemma K.12 and Lemma E.2. We already have

$$\frac{1-\delta\lambda_i}{1-\frac{q-c\delta}{1-c}\lambda_i} \leq \frac{4}{3+c}$$

by (K.19). We also have

$$\frac{(1-c)^2-[(1+c)q-2c\delta]\lambda_i}{1-\frac{q-c\delta}{1-c}\lambda_i - x_1} = \frac{(1-c)^2-[(1+c)q-2c\delta]\lambda_i}{1-\frac{q-c\delta}{1-c}\lambda_i - \frac{(1+c-q\lambda_i)-\sqrt{(1+c-q\lambda_i)^2-4c(1-\delta\lambda_i)}}{2}}$$

$$= 2(1-c)\cdot\frac{(1-c)^2-[(1+c)q-2c\delta]\lambda_i}{(1-c)^2-[(1+c)q-2c\delta]\lambda_i+(1-c)\sqrt{(1+c-q\lambda_i)^2-4c(1-\delta\lambda_i)}}$$

$$\leq 2(1-c),$$

where the inequality holds because $(1-c)\sqrt{(1+c-q\lambda_i)^2-4c(1-\delta\lambda_i)} \geq 0$. We thus have

$$\frac{1-\delta\lambda_i}{1-\frac{q-c\delta}{1-c}\lambda_i} \cdot \frac{(1-c)^2-[(1+c)q-2c\delta]\lambda_i}{1-\frac{q-c\delta}{1-c}\lambda_i - x_1} \leq \frac{4}{3+c}\cdot 2(1-c) \leq \frac{8}{3}(1-c) \leq 3(1-c), \tag{K.42}$$

where the second inequality holds because $c \geq 0$, and the last inequality holds because $8/3 < 3$. Combining (K.42) with (K.40) and (K.41), we have

$$\left(\sum_{k=0}^{t-1}\mathbf{A}_i^{j+k}\begin{bmatrix}1\\1\end{bmatrix}\right)_1 \leq \frac{3(1-c)}{(q-c\delta)\lambda_i}\left(1-\frac{q-c\delta}{1-c}\lambda_i\right)^j(1-x_2^t)$$

$$\leq \frac{3(1-c)}{(q-c\delta)\lambda_i} \left(1 - \frac{q-c\delta}{1-c}\lambda_i\right)^j \left[1 - \left(1 - 2\frac{q-c\delta}{1-c}\lambda_i\right)^t\right],$$

where the second inequality holds due to Lemma E.2. For $j = 0$, we have

$$[(1-c)(1-\delta\lambda_i) - (1-c+(q-\delta)\lambda_i)x_2]\frac{x_2^0 - x_1^0}{x_2 - x_1} + (1-c+(q-\delta)\lambda_i)x_2^0$$

$$= 1 - c + (q-\delta)\lambda_i \leq 1 - c + (q-\delta) \cdot \frac{(1-c)^2}{(\sqrt{q-c\delta} + \sqrt{c(q-\delta)})^2}$$

$$= (1-c)\frac{(q-\delta) + (q-c\delta) + 2\sqrt{c(q-\delta)(q-c\delta)}}{c(q-\delta) + (q-c\delta) + 2\sqrt{c(q-\delta)(q-c\delta)}}$$

$$\leq (1-c)\frac{(q-\delta) + (q-\delta)/c + 2(q-\delta)}{c(q-\delta) + (q-\delta)/c + 2(q-\delta)}$$

$$= (1-c)\left[\frac{1}{1+c} + \frac{2}{(1+c)^2}\right] \leq 3(1-c),$$

where the first inequality holds due to (E.6), the second inequality holds because $q - c\delta \geq (q-\delta)/c$, and the last inequality holds because $c \geq 0$. Therefore, the upper bound also holds for $j = 0$. The lower bound of (K.30) is given by

$$\left(\sum_{k=0}^{t-1} \mathbf{A}_i^{j+k}\begin{bmatrix}1\\1\end{bmatrix}\right)_1 \geq \frac{1}{(q-c\delta)\lambda_i}\left\{\frac{[(1-c)(1-\delta\lambda_i) - (1-c+(q-\delta)\lambda_i)x_1]x_2^j(1-x_2^t)}{x_1 - x_2}\right.$$

$$\left. - \frac{[(1-c)(1-\delta\lambda_i) - (1-c+(q-\delta)\lambda_i)x_2]x_2^j(1-x_1^t)}{x_2 - x_1}\right\}$$

$$= \frac{x_2^j}{(q-c\delta)\lambda_i}\left\{(1-c+(q-\delta)\lambda_i)(1-x_2^t)\right.$$

$$\left. - [(1-c)(1-\delta\lambda_i) - (1-c+(q-\delta)\lambda_i)x_2]\frac{x_2^t - x_1^t}{x_2 - x_1}\right\}$$

$$\geq \frac{x_2^j}{(q-c\delta)\lambda_i}\left\{(1-c+(q-\delta)\lambda_i)(1-x_2^t)\right.$$

$$\left. - [(1-c)(1-\delta\lambda_i) - (1-c+(q-\delta)\lambda_i)x_2]\frac{1-x_2^t}{1-x_2}\right\}$$

$$= \frac{x_2^j(1-x_2^t)}{1-x_2} \geq 0,$$

where the first inequality holds due to (K.39) and $x_1 < x_2$, the second inequality holds due to Lemma K.12, and the third inequality holds because $0 < x_2 < 1$. $\qquad\square$

The following corollary follows from Lemma K.8.
**Corollary K.9.** With $\mathbf{A}_i$ defined in (E.1), we have

$$\sum_{i=1}^d \lambda_i w_i^2 \left(\sum_{k=0}^{N-1} \mathbf{A}_i^{s+k}\begin{bmatrix}1\\1\end{bmatrix}\right)_1^2$$

$$\leq \frac{16}{\delta^2}(c\delta/q)^{2s}\|\mathbf{w}_0 - \mathbf{w}^*\|^2_{\mathbf{H}_{0:k\ddagger}^{-1}} + 8s^2 c^s\|(\mathbf{I} - \delta\mathbf{H})^{s/2}(\mathbf{w}_0 - \mathbf{w}^*)\|^2_{\mathbf{H}_{k\ddagger:k\dagger}}$$

$$+ \frac{32}{\delta^2} \cdot c^s\|(\mathbf{I} - \delta\mathbf{H})^{s/2}(\mathbf{w}_0 - \mathbf{w}^*)\|^2_{\mathbf{H}_{k\ddagger:\widehat{k}}^{-1}} + \frac{200}{(1-c)^2} \cdot c^s\|(\mathbf{I} - \delta\mathbf{H})^{s/2}(\mathbf{w}_0 - \mathbf{w}^*)\|^2_{\mathbf{H}_{\widehat{k}:k\dagger}}$$

$$+ \frac{9(1-c)^2}{(q-c\delta)^2}\left\|\left(\mathbf{I} - \frac{q-c\delta}{1-c}\mathbf{H}\right)^s(\mathbf{w}_0 - \mathbf{w}^*)\right\|^2_{\mathbf{H}_{k\dagger:k^*}^{-1}}$$

$$+ 36N^2\left\|\left(\mathbf{I} - \frac{q-c\delta}{1-c}\mathbf{H}\right)^s(\mathbf{w}_0 - \mathbf{w}^*)\right\|^2_{\mathbf{H}_{k^*\infty}}$$

*Proof.* By Lemma K.8, we have

(a) For $i \le k^\ddagger$, we have

$$\left(\sum_{k=0}^{N-1} \mathbf{A}_i^{s+k} \begin{bmatrix} 1 \\ 1 \end{bmatrix}\right)_1^2 \le \frac{16}{\delta^2 \lambda_i^2}(c\delta/q)^{2s}.$$

(b) For $k^\ddagger < i \le \widehat{k}$, we have

$$\left(\sum_{k=0}^{N-1} \mathbf{A}_i^{s+k} \begin{bmatrix} 1 \\ 1 \end{bmatrix}\right)_1^2 \le \left(2s[c(1-\delta\lambda_i)]^{s/2} + \frac{4}{\delta\lambda_i}[c(1-\delta\lambda_i)]^{s/2}\right)^2$$

$$\le 8s^2[c(1-\delta\lambda_i)]^s + \frac{32}{\delta^2\lambda_i^2}[c(1-\delta\lambda_i)]^s,$$

where the inequality holds due to Cauchy-Schwarz inequality.

(c) For $\widehat{k} < i \le k^\dagger$, we have

$$\left(\sum_{k=0}^{N-1} \mathbf{A}_i^{s+k} \begin{bmatrix} 1 \\ 1 \end{bmatrix}\right)_1^2 \le \left(2s[c(1-\delta\lambda_i)]^{s/2} + \frac{10}{1-c}[c(1-\delta\lambda_i)]^{s/2}\right)^2$$

$$\le 8s^2[c(1-\delta\lambda_i)]^s + \frac{200}{(1-c)^2}[c(1-\delta\lambda_i)]^s,$$

where the inequality holds due to Cauchy-Schwarz inequality.

(d) For $i > k^\dagger$, we have

$$\left(\sum_{k=0}^{N-1} \mathbf{A}_i^{s+k} \begin{bmatrix} 1 \\ 1 \end{bmatrix}\right)_1^2 \le \frac{9(1-c)^2}{(q-c\delta)^2\lambda_i^2}\left[1-\left(1-2\frac{q-c\delta}{1-c}\lambda_i\right)^N\right]^2\left(1-\frac{q-c\delta}{1-c}\lambda_i\right)^{2s}$$

$$\le \min\left\{\frac{9(1-c)^2}{(q-c\delta)^2\lambda_i^2}, 36N^2\right\}\left(1-\frac{q-c\delta}{1-c}\lambda_i\right)^{2s},$$

where the second inequality holds because $1-(1-r)^N \le 1$ and $1-(1-r)^N \le rN$ hold for all $r \in (0,1)$.

Concluding all the above, we have

$$\sum_{i=1}^{d} \lambda_i w_i^2 \left(\sum_{k=0}^{N-1} \mathbf{A}_i^{s+k} \begin{bmatrix} 1 \\ 1 \end{bmatrix}\right)_1^2$$

$$\le \sum_{i\le k^\ddagger} \lambda_i w_i^2 \cdot \frac{16}{\delta^2\lambda_i^2}(c\delta/q)^{2s} + \sum_{k^\ddagger < i \le \widehat{k}} \lambda_i w_i^2 \cdot \left(8s^2[c(1-\delta\lambda_i)]^s + \frac{32}{\delta^2\lambda_i^2}[c(1-\delta\lambda_i)]^s\right)$$

$$+ \sum_{\widehat{k} < i \le k^\dagger} \lambda_i w_i^2 \cdot \left(8s^2[c(1-\delta\lambda_i)]^s + \frac{200}{(1-c)^2}[c(1-\delta\lambda_i)]^s\right)$$

$$+ \sum_{k^\dagger < i \le k^*} \lambda_i w_i^2 \cdot \frac{9(1-c)^2}{(q-c\delta)^2\lambda_i^2}\left(1-\frac{q-c\delta}{1-c}\lambda_i\right)^{2s} + \sum_{i>k^*} \lambda_i w_i^2 \cdot 36N^2 \left(1-\frac{q-c\delta}{1-c}\lambda_i\right)^{2s}$$

$$= \frac{16}{\delta^2}(c\delta/q)^{2s}\|\mathbf{w}_0 - \mathbf{w}^*\|_{\mathbf{H}_{0:k^\ddagger}^{-1}}^2 + 8s^2c^s\|(\mathbf{I}-\delta\mathbf{H})^{s/2}(\mathbf{w}_0-\mathbf{w}^*)\|_{\mathbf{H}_{k^\ddagger:k^\dagger}}^2$$

$$+ \frac{32}{\delta^2}\cdot c^s\|(\mathbf{I}-\delta\mathbf{H})^{s/2}(\mathbf{w}_0-\mathbf{w}^*)\|_{\mathbf{H}_{k^\ddagger:\widehat{k}}^{-1}}^2 + \frac{200}{(1-c)^2}\cdot c^s\|(\mathbf{I}-\delta\mathbf{H})^{s/2}(\mathbf{w}_0-\mathbf{w}^*)\|_{\mathbf{H}_{\widehat{k}:k^\dagger}}^2$$

$$+ \frac{9(1-c)^2}{(q-c\delta)^2}\left\|\left(\mathbf{I}-\frac{q-c\delta}{1-c}\mathbf{H}\right)^s(\mathbf{w}_0-\mathbf{w}^*)\right\|_{\mathbf{H}_{k^\dagger:k^*}^{-1}}^2$$

$$+ 36N^2\left\|\left(\mathbf{I}-\frac{q-c\delta}{1-c}\mathbf{H}\right)^s(\mathbf{w}_0-\mathbf{w}^*)\right\|_{\mathbf{H}_{k^*\infty}}^2,$$

where the first inuequality holds because the upper bound $\frac{9(1-c)^2}{(q-c\delta)^2\lambda_i^2}$ is applied for $k^\dagger < i \le k^*$ and $36N^2$ is applied for $i > k^*$. $\qquad\square$

**Lemma K.10.** With $\mathbf{A}_i$ defined in (E.1), let $x_1$ and $x_2$ be the eigenvalues of $\mathbf{A}_i$ as defined in (E.2) and (E.3). Then

- For all $i \le k^\ddagger$, we have

$$\sum_{k=0}^{t-1}\left(\mathbf{A}_i^k\begin{bmatrix}1\\1\end{bmatrix}\right)_2^2 \le \frac{7}{2\delta\lambda_i};$$

- For all $k^\ddagger < i \le \widehat{k}$, we have

$$\sum_{k=0}^{t-1}\left(\mathbf{A}_i^k\begin{bmatrix}1\\1\end{bmatrix}\right)_2^2 \le \frac{14}{\delta\lambda_i};$$

- For all $\widehat{k} < i \le k^\dagger$, we have

$$\sum_{k=0}^{t-1}\left(\mathbf{A}_i^k\begin{bmatrix}1\\1\end{bmatrix}\right)_2^2 \le \frac{10}{1-c};$$

- For all $i > k^\dagger$, we have

$$\sum_{k=0}^{t-1}\left(\mathbf{A}_i^k\begin{bmatrix}1\\1\end{bmatrix}\right)_2^2 \le \frac{1-c}{(q-c\delta)\lambda_i}\left[1-\left(1-2\frac{q-c\delta}{1-c}\lambda_i\right)^{2t}\right].$$

*Proof.* Note that

$$\left(\mathbf{A}_i^k\begin{bmatrix}1\\1\end{bmatrix}\right)_2 = (\mathbf{A}_i^k)_{21} + (\mathbf{A}_i^k)_{22} = -c\frac{x_2^k - x_1^k}{x_2 - x_1} + \frac{x_2^{k+1} - x_1^{k+1}}{x_2 - x_1}$$
$$= \frac{(x_2-c)x_2^k - (x_1-c)x_1^k}{x_2 - x_1}, \tag{K.43}$$

where the second equality holds due to Lemma E.3. Summing up the square of (K.43) yields

$$\sum_{k=0}^{t-1}\left(\mathbf{A}_i^k\begin{bmatrix}1\\1\end{bmatrix}\right)_2^2 = \sum_{k=0}^{t-1}\left[\frac{(x_2-c)x_2^k - (x_1-c)x_1^k}{x_2-x_1}\right]^2$$
$$= \sum_{k=0}^{t-1}\frac{(x_2-c)^2 x_2^{2k}}{(x_2-x_1)^2} - 2\sum_{k=0}^{t-1}\frac{(x_2-c)(x_1-c)(x_1 x_2)^k}{(x_2-x_1)^2} + \sum_{k=0}^{t-1}\frac{(x_1-c)^2 x_1^{2k}}{(x_2-x_1)^2}$$
$$= \frac{(x_2-c)^2(1-x_2^{2t})}{(1-x_2^2)(x_2-x_1)^2} - 2\frac{(x_2-c)(x_1-c)[1-(x_1 x_2)^t]}{(1-x_1 x_2)(x_2-x_1)^2} + \frac{(x_1-c)^2(1-x_1^{2t})}{(1-x_1^2)(x_2-x_1)^2}. \tag{K.44}$$

Denote

$$A := \frac{(x_2-c)^2}{1-x_2^2}, \quad B := \frac{(x_1-c)(x_2-c)}{1-x_1 x_2}, \quad C := \frac{(x_1-c)^2}{1-x_1^2},$$

then we have

$$\frac{A-B}{x_2-x_1} = \frac{(x_2-c)(1-cx_2)}{(1-x_2^2)(1-x_1 x_2)}, \quad \frac{B-C}{x_2-x_1} = \frac{(x_1-c)(1-cx_1)}{(1-x_1^2)(1-x_1 x_2)},$$
$$\frac{A-2B+C}{(x_2-x_1)^2} = \frac{(1+c^2)(1+x_1 x_2) - 2c(x_1+x_2)}{(1-x_1^2)(1-x_2^2)(1-x_1 x_2)}.$$

For all $i \le k^\ddagger$, (K.44) is bounded by

$$\sum_{k=0}^{t-1}\left(\mathbf{A}_i^k\begin{bmatrix}1\\1\end{bmatrix}\right)_2^2 = (1-x_2^{2t})\cdot\frac{A-2B+C}{(x_2-x_1)^2} + 2\frac{C-B}{x_2-x_1}\cdot x_2^t\cdot\frac{x_2^t - x_1^t}{x_2-x_1} - C\cdot\left(\frac{x_2^t-x_1^t}{x_2-x_1}\right)^2$$
$$\le (1-x_2^{2t})\cdot\frac{A-2B+C}{(x_2-x_1)^2} + 2x_2\cdot\frac{C-B}{x_2-x_1}\cdot\frac{x_2^{2t}-(x_1 x_2)^t}{x_2^2-x_1 x_2}$$

$$\leq \frac{A - 2B + C}{(x_2 - x_1)^2} + \frac{2x_2(C - B)}{x_2 - x_1} \cdot \frac{1}{1 - x_1 x_2}$$

$$= \frac{(1 + c^2)(1 + x_1 x_2) - 2c(x_1 + x_2)}{(1 - x_1^2)(1 - x_2^2)(1 - x_1 x_2)} + \frac{2x_2(c - x_1)(1 - cx_1)}{(1 - x_1^2)(1 - x_1 x_2)^2}, \tag{K.45}$$

where the first inequality holds because $C \left( \frac{x_2^t - x_1^t}{x_2 - x_1} \right)^2 \geq 0$, and the second inequality holds because due to Lemma K.12. Note that

$$\frac{(1 + c^2)(1 + x_2 x_2) - 2c(x_1 + x_2)}{(1 - x_1^2)(1 - x_2^2)}$$

$$= \frac{(1 + c)^2}{2(1 + x_1)(1 + x_2)} + \frac{(1 - c)^2}{2(1 - x_1)(1 - x_2)} \leq \frac{(1 + c)^2}{2} + \frac{(1 - c)^2}{2(q - c\delta)\lambda_i}$$

$$\leq \frac{(1 + c)^2}{2} + \frac{(\sqrt{q - c\delta} - \sqrt{c(q - \delta)})^2}{2(q - c\delta)} \leq \frac{(1 + 1)^2}{2} + \frac{q - c\delta}{2(q - c\delta)} = \frac{5}{2}, \tag{K.46}$$

where the first inequality holds because $1 + x_2 \geq 1 + x_1 \geq 1$, the second inequality holds due to (E.6), and the last inequality holds because $\sqrt{q - c\delta} - \sqrt{c(q - \delta)} \leq \sqrt{q - c\delta}$. We also have

$$\frac{(c - x_1)(1 - cx_1)x_2}{1 - x_1^2} \leq (c - x_1)x_2 = \frac{(c - x_1)(c - x_2)x_2}{c - x_2} = \frac{c(q - \delta)\lambda_i \cdot x_2}{c - x_2}$$

$$\leq \frac{c(q - \delta)\lambda_i \cdot \frac{c\delta - \sqrt{c(q - \delta)(q - c\delta)}}{q}}{c - \frac{c\delta - \sqrt{c(q - \delta)(q - c\delta)}}{q}} = \frac{\sqrt{c(q - \delta)}(c\delta - \sqrt{c(q - \delta)(q - c\delta)})}{\sqrt{q - c\delta} + \sqrt{c(q - \delta)}}\lambda_i$$

$$\leq \frac{\sqrt{c(q - \delta)} \cdot \delta}{\sqrt{c(q - \delta)} + \sqrt{c(q - \delta)}}\lambda_i = \frac{\delta \lambda_i}{2}, \tag{K.47}$$

where the first inequality holds because $1 - cx_1 \leq 1 - x_1^2$ (due to the fact that $x_1 \leq x_2 \leq c\delta/q \leq c$), the second inequality holds due to Lemma E.2, and the last inequality holds because $\sqrt{q - c\delta} \geq \sqrt{c(q - \delta)}$ and $c\delta - \sqrt{c(q - \delta)(q - c\delta)} \leq c\delta \leq \delta$. We finally have

$$1 - x_1 x_2 = 1 - c(1 - \delta \lambda_i) \geq \delta \lambda_i. \tag{K.48}$$

Substituting (K.46), (K.47) and (K.48) into (K.45), we have

$$\sum_{k=0}^{t-1} \left( \mathbf{A}_i^k \begin{bmatrix} 1 \\ 1 \end{bmatrix} \right)_2^2 \leq \frac{5}{2} \cdot \frac{1}{\delta \lambda_i} + \delta \lambda_i \cdot \frac{1}{(\delta \lambda_i)^2} = \frac{7}{2\delta \lambda_i}.$$

For $k^{\ddagger} < i \leq k^{\dagger}$, (K.44) can be bounded as

$$\sum_{k=0}^{t-1} \left( \mathbf{A}_i^k \begin{bmatrix} 1 \\ 1 \end{bmatrix} \right)_2^2$$

$$= (1 - (x_1 x_2)^t) \cdot \frac{A - 2B + C}{(x_2 - x_1)^2} - \frac{A + C}{2} \cdot \left( \frac{x_2^t - x_1^t}{x_2 - x_1} \right)^2 + \frac{C - A}{2(x_2 - x_1)} \cdot \frac{x_2^{2t} - x_1^{2t}}{x_2 - x_1}$$

$$\leq |1 - (x_1 x_2)^t| \cdot \left| \frac{A - 2B + C}{(x_2 - x_1)^2} \right| + \frac{|A + C|}{2} \cdot \left| \frac{x_2^t - x_1^t}{x_2 - x_1} \right|^2 + \frac{1}{2} \cdot \left| \frac{C - A}{x_2 - x_1} \right| \cdot \left| \frac{x_2^{2t} - x_1^{2t}}{x_2 - x_1} \right|$$

$$\leq \left| \frac{A - 2B + C}{(x_2 - x_1)^2} \right| + \frac{|A + C|}{2} \cdot (t[(1 - \delta \lambda_i)]^{(t-1)/2})^2 + \frac{1}{2} \cdot \left| \frac{C - A}{x_2 - x_1} \right| \cdot 2t[c(1 - \delta \lambda_i)]^{(2t-1)/2}$$

$$= \frac{1}{1 - x_1 x_2} \cdot \left| \frac{(1 + c^2)(1 + x_1 x_2) - 2c(x_1 + x_2)}{(1 - x_1^2)(1 - x_2^2)} \right| + \frac{|A + C|}{2} \cdot (t[(1 - \delta \lambda_i)]^{(t-1)/2})^2$$

$$+ \left| \frac{2c(1 + x_1 x_2) - (1 + c^2)(x_1 + x_2)}{2(1 - x_1^2)(1 - x_2^2)} \right| \cdot 2t[c(1 - \delta \lambda_i)]^{(2t-1)/2}, \tag{K.49}$$

where the first inequality holds due to triangle inequality, and the second inequality holds because $0 \leq 1 - (x_1 x_2)^t \leq 1$ and due to Lemma K.13. We now bound the coefficients. Note that

$$\frac{(1 - c)^2}{(1 - x_1)(1 - x_2)} = \frac{(1 - c)^2}{(q - c\delta)\lambda_i} \leq \frac{(\sqrt{q - c\delta} + \sqrt{c(q - \delta)})^2}{q - c\delta} \leq \frac{(\sqrt{q - c\delta} + \sqrt{q - c\delta})^2}{q - c\delta} = 4$$

where the first inequality holds due to (E.5), and the second inequality holds because $c(q - \delta) \le q - c\delta$. We also note that

$$\frac{(1+c)^2}{(1+x_1)(1+x_2)} = \frac{(1+c)^2}{2(1+c) - (q+c\delta)\lambda_i} \le \frac{(1+c)^2}{2(1+c) - (1+2c)\delta\lambda_i}$$

$$\le \frac{(1+c)^2}{2(1+c) - (1+2c)} \le \frac{(1+1)^2}{1} = 4,$$

where the first equality holds due to Lemma K.2(c), the first inequality holds because $q \le (1+c)\delta$, the second inequality holds because $\delta\lambda_i \le 1$, and the last inequality holds because $c \le 1$. We thus have

$$\left| \frac{(1+c^2)(1+x_1 x_2) - 2c(x_1 + x_2)}{(1-x_1^2)(1-x_2^2)} \right| = \frac{1}{2} \left| \frac{(1+c)^2}{(1+x_1)(1+x_2)} + \frac{(1-c)^2}{(1-x_1)(1-x_2)} \right| \le 4, \quad \text{(K.50)}$$

$$\left| \frac{2c(1+x_1 x_2) - (1+c^2)(x_1 + x_2)}{(1-x_1^2)(1-x_2^2)} \right| = \frac{1}{2} \left| \frac{(1+c)^2}{(1+x_1)(1+x_2)} - \frac{(1-c)^2}{(1-x_1)(1+x_2)} \right| \le 2. \quad \text{(K.51)}$$

We then bound $|A + C|/2$. Note that

$$\frac{A+C}{2} = \frac{1}{2}\left[ \frac{(x_1-c)^2}{1-x_1^2} + \frac{(x_2-c)^2}{1-x_2^2} \right] = \frac{(x_1-c)^2(1-x_2^2) + (x_2-c)^2(1-x_1^2)}{2(1-x_1^2)(1-x_2^2)}$$

$$= \frac{2c^2 - 2c(x_1+x_2) + (1-c^2)(x_1^2+x_2^2) + 2cx_1 x_2(x_1+x_2) - 2x_1^2 x_2^2}{2(q-c\delta)\lambda_i \cdot [2(1+c) - (q+c\delta)\lambda_i]}$$

$$= \frac{(1+c)(1-c)^3 - 2(1-c)[(1+c+c^2)q - c(1+2c)\delta]\lambda_i + [(1-c^2)q^2 + 2c^2\delta q - 2c^2\delta^2]\lambda_i^2}{2(q-c\delta)\lambda_i \cdot [2(1+c) - (q+c\delta)\lambda_i]}.$$

$$\text{(K.52)}$$

For $k^\ddagger < i \le \hat{k}$, we aim to bound the denominator of (K.52) by $\lambda_i^2$ multiplied by a constant. Denote the denominator divided by $\lambda_i^2$ as

$$\phi(\lambda_i) := \frac{(1+c)(1-c)^3}{\lambda_i^2} - \frac{2(1-c)[(1+c+c^2)q - c(1+2c)\delta]}{\lambda_i} + [(1-c^2)q^2 + 2c^2\delta q - 2c^2\delta^2],$$

then

$$\frac{\partial \phi}{\partial \frac{1}{\lambda_i}} = 2(1-c)\left[ \frac{(1+c)(1-c)^2}{\lambda_i} - [(1+c+c^2)q - c(1+2c)\delta] \right]$$

$$\le 2(1-c)\left[ \frac{(1+c)(1-c)^2}{(1-c)/\delta} - [(1+c+c^2)q - c(1+2c)\delta] \right]$$

$$= -2(1-c)(1+c+c^2)(q - \delta) \le 0,$$

where the second inequality holds due to (E.7), and the last inequality holds because $q \ge \delta$, so $\phi$ is a decreasing function in $1/\lambda_i$. We thus have

$$\phi(\lambda_i) \ge \phi((1-c)/\delta)$$

$$= \frac{(1+c)(1-c)^3}{(1-c)^2/\delta^2} - \frac{2(1-c)[(1+c+c^2)q - c(1+2c)\delta]}{(1-c)/\delta} + [(1-c^2)q^2 + 2c^2\delta q - 2c^2\delta^2]$$

$$= (1+c)(q-\delta)[(1-c)q - (1+c)\delta]$$

$$\ge (1+c)(q-\delta)[(1-c)\delta - (1+c)\delta] = -2c\delta(1+c)(q-\delta)$$

$$\ge -4\delta(q - c\delta),$$

where the first inequality holds because $\lambda_i \ge (1-c)/\delta$, the second inequality holds because $q \ge \delta$, and the last inequality holds because $c \le 1$ and $q - \delta \le q - c\delta$. We also note that $2(1+c) - (q + c\delta)\lambda_i \ge 1$, so $(A+C)/2 \le 2\delta\lambda_i$. We also have

$$\phi(\lambda_i) \le \phi\left( \frac{(\sqrt{q-c\delta} + \sqrt{c(q-\delta)})^2}{q^2} \right),$$

$$2(1+c) - (q+c\delta)\lambda_i \geq 2(1+c) - (q+c\delta) \cdot \frac{(\sqrt{q-c\delta} + \sqrt{c(q-\delta)})^2}{q^2},$$

so the upper bound of $\frac{A+C}{2}$ is

$$\begin{aligned}
\frac{A+C}{2} &\leq \frac{(c(q-\delta) + \sqrt{c(q-\delta)(q-c\delta)})^2}{1 - (c\delta - \sqrt{c(q-\delta)(q-c\delta)})^2/q^2} \cdot \frac{\lambda_i}{(\sqrt{q-c\delta} + \sqrt{c(q-\delta)})^2} \\
&\leq \frac{(c(q-\delta) + \sqrt{c(q-\delta)(q-c\delta)})^2}{1 - (c\delta - \sqrt{c(q-\delta)(q-c\delta)})/q} \cdot \frac{\lambda_i}{(\sqrt{q-c\delta} + \sqrt{c(q-\delta)})^2} \\
&= \frac{q \cdot c(q-\delta)\lambda_i}{(q-c\delta) + \sqrt{c(q-\delta)(q-c\delta)}} \\
&\leq \frac{2\delta(q-c\delta)\lambda_i}{q-c\delta} = 2\delta\lambda_i,
\end{aligned}$$

where the second inequality holds because $(c\delta - \sqrt{c(q-\delta)(q-c\delta)})/q \leq 1$, and the last inequality holds because $c(q-\delta) \leq q - c\delta$, $q \leq (1+c)\delta \leq 2\delta$ and $\sqrt{c(q-\delta)(q-c\delta)} \geq 0$. Therefore,

$$\frac{|A+C|}{2} \leq 2\delta\lambda_i, \tag{K.53}$$

where the second inequality holds because $2(1+c) - (q+c\delta)\lambda_i \geq 1$. For $\widehat{k} < i \leq k^\dagger$, we aim to bound the denominator of (K.52) as $\lambda_i$ multiplied by a constant. Denote the denominator devided by $\lambda_i$ as

$$\varphi(\lambda_i) := \frac{(1+c)(1-c)^3}{\lambda_i} - 2(1-c)[(1+c+c^2)q - c(1+2c)\delta] + [(1-c^2)q^2 + 2c^2\delta q - 2c^2\delta^2]\lambda_i,$$

then the lower bound of $\varphi(\lambda_i)$ is given by

$$\begin{aligned}
\varphi(\lambda_i) &\geq -2(1-c)[(1+c+c^2)q - c(1+2c)\delta] \\
&= -2(1-c)[(1+c)(q-c\delta) + c^2(q-\delta)] \\
&\geq -2(1-c)(1+c+c^2)(q-c\delta),
\end{aligned}$$

where the first inequality holds because $\frac{(1+c)(1-c)^3}{\lambda_i} \geq 0$ and $[(1-c^2)q^2 + 2c^2\delta q - 2c^2\delta^2]\lambda_i \geq 0$, and the second inequality holds because and $q - \delta \leq q - c\delta$. Note that the maximum of $\varphi(\lambda_i)$ is attained at either $\frac{1-c}{\delta}$ or $\frac{(1-c)^2}{(\sqrt{q-c\delta} + \sqrt{c(q-\delta)})^2}$. For the former, we have

$$\begin{aligned}
\varphi((1-c)/\delta) &= \frac{(1+c)(1-c)^3}{(1-c)/\delta} - 2(1-c)[(1+c+c^2)q - c(1+2c)\delta] \\
&\quad + [(1-c^2)q^2 + 2c^2\delta q - 2c^2\delta^2] \cdot \frac{1-c}{\delta} \\
&= (1-c)(1+c)(q-\delta)[(1-c)q/\delta - (1+c)] \\
&\leq (1-c)(1+c)(q-\delta)[(1-c)(1+c) - (1+c)] \\
&= -c(1-c)(1+c)(q-\delta) \leq 0,
\end{aligned}$$

where the first inequality holds because $q \leq (1+c)\delta$, and the second inequality holds because $q \leq \delta$. For the latter,

$$\begin{aligned}
&\varphi\left(\frac{(1-c)^2}{(\sqrt{q-c\delta} + \sqrt{c(q-\delta)})^2}\right) \\
&= 2(q-c\delta) \cdot \frac{(c(q-\delta) - \sqrt{c(q-\delta)(q-c\delta)})^2}{q^2} \cdot \frac{q+c\delta + \sqrt{c(q-\delta)(q-c\delta)}}{q-c\delta - \sqrt{c(q-\delta)(q-c\delta)}} \\
&= 2(1-c)c(q-\delta) \cdot \frac{\sqrt{q-c\delta}}{\sqrt{q-c\delta} + \sqrt{c(q-\delta)}} \cdot \frac{q+c\delta + \sqrt{c(q-\delta)(q-c\delta)}}{q}
\end{aligned}$$

$$\leq 2c^2(1-c)(q-c\delta) \cdot 1 \cdot \frac{q+c\delta+q-c\delta}{q} = 2c^2(1-c)(q-c\delta),$$

where the inequality holds because $q - \delta \leq c(q - c\delta)$ and $c(q - \delta) \leq q - c\delta$. We finally have

$$2(1+c) - (q+c\delta)\lambda_i \geq 2(1+c) - (1+2c)\delta\lambda_i \geq 2(1+c) - (1+2c)(1-c) = 1 + c + 2c^2,$$

where the first inequality holds because $q \leq (1+c)\delta$, and the second inequality holds because $\delta\lambda_i \leq 1 - c$ (due to definition of $\widehat{k}$). Therefore,

$$\frac{|A+C|}{2} \leq \max\left\{\frac{1+c+c^2}{1+c+2c^2}, \frac{c^2}{1+c+2c^2}\right\} \cdot (1-c) \leq (1-c), \tag{K.54}$$

where the second inequality holds because $1 + c + c^2 \leq 1 + c + 2c^2$ and $c^2 \leq 1 + c + 2c^2$. Therefore, when $k^\ddagger < i \leq \widehat{k}$, $1 - x_1 x_2 \geq \delta\lambda_i$, so (K.49) can be further bounded by

$$\sum_{k=0}^{t-1}\left(\mathbf{A}_i^k\begin{bmatrix}1\\1\end{bmatrix}\right)_2^2 \leq \frac{4}{\delta\lambda_i} + 2\delta\lambda_i \cdot (t[c(1-\delta\lambda_i)]^{(t-1)/2})^2 + \frac{1}{2} \cdot 2 \cdot 2t[c(1-\delta\lambda_i)]^{(2t-1)/2}$$

$$\leq \frac{4}{\delta\lambda_i} + 2\delta\lambda_i \cdot \frac{4}{\delta^2\lambda_i^2} + \frac{2}{\delta\lambda_i} = \frac{14}{\delta\lambda_i},$$

where the first inequality holds due to (K.50), (K.51) and (K.53), and the second inequality holds due to Lemma K.14. When $\widehat{k} < i \leq k^\dagger$, $1 - x_1 x_2 \geq 1 - c$, so (K.49) is further bounded by

$$\sum_{k=0}^{t-1}\left(\mathbf{A}_i^k\begin{bmatrix}1\\1\end{bmatrix}\right)_2^2 \leq \frac{4}{1-c} + (1-c) \cdot ([t(1-\delta\lambda_i)]^{(t-1)/2})^2 + \frac{1}{2} \cdot 2 \cdot 2t[c(1-\delta\lambda_i)]^{(2t-1)/2}$$

$$\leq \frac{4}{1-c} + (1-c) \cdot \frac{4}{(1-c)^2} + \frac{2}{1-c} = \frac{10}{1-c},$$

where the first inequality holds due to (K.50), (K.51), and the second inequality holds due to Lemma K.14.

For all $i > k^\dagger$, (K.44) can by bounded as

$$\sum_{k=0}^{t-1}\left(\mathbf{A}_i^k\begin{bmatrix}1\\1\end{bmatrix}\right)_2^2 = (1-x_2^{2t}) \cdot \frac{A-2B+C}{(x_2-x_1)^2} - 2\frac{B-C}{x_2-x_1} \cdot \frac{x_2^{2t}-(x_1x_2)^t}{x_2-x_1} - C\left(\frac{x_2^t-x_1^t}{x_2-x_1}\right)^2$$

$$\leq (1-x_2^{2t}) \cdot \frac{A-2B+C}{(x_2-x_1)^2} = (1-x_2^{2t})\frac{(1+c^2)(1+x_1x_2)-2c(x_1+x_2)}{(1-x_1^2)(1-x_2^2)(1-x_1x_2)}, \tag{K.55}$$

where the inequality holds because negative terms are dropped. Note that

$$\frac{(1-c)^2}{(1-x_1)(1-x_2)} = \frac{(1-c)^2}{(q-c\delta)\lambda_i},$$

and

$$\frac{(1+c)^2}{(1+x_1)(1+x_2)} \leq \frac{(1+c)^2}{(1+c)^2} = 1 \leq \frac{(1-c)^2}{(\sqrt{q-c\delta}+\sqrt{c(q-\delta)})^2\lambda_i} \leq \frac{(1-c)^2}{(q-c\delta)\lambda_i},$$

where the first inequality holds because $c \leq x_1 \leq x_2$, the second inequality holds due to (E.5), and the last inequality holds because $\sqrt{c(q-\delta)} \geq 0$. We thus have

$$\frac{(1+c^2)(1+x_1x_2)-2c(x_1+x_2)}{(1-x_1^2)(1-x_2^2)} = \frac{(1+c)^2}{2(1+x_1)(1+x_2)} + \frac{(1-c)^2}{2(1-x_1)(1-x_2)}$$

$$\leq \frac{(1-c)^2}{2(q-c\delta)\lambda_i} + \frac{(1-c)^2}{2(q-c\delta)\lambda_i} = \frac{(1-c)^2}{(q-c\delta)\lambda_i}. \tag{K.56}$$

We also have

$$1 - x_1 x_2 = 1 - c + c\delta\lambda_i \geq 1 - c, \tag{K.57}$$

where the inequality holds because $c\delta\lambda_i \geq 0$. Substituting (K.56) and (K.57) into (K.55), we have

$$\sum_{k=0}^{t-1}\left(\mathbf{A}_i^k\begin{bmatrix}1\\1\end{bmatrix}\right)_2^2 \leq (1-x_2^{2t}) \cdot \frac{1-c}{(q-c\delta)\lambda_i} \leq \frac{1-c}{(q-c\delta)\lambda_i}\left[1-\left(1-2\frac{q-c\delta}{1-c}\lambda_i\right)^{2t}\right],$$

where the second inequality holds due to Lemma E.2. $\qquad\square$

The following lemma follows from Lemma K.8.

**Corollary K.11.** With $\mathbf{A}_i$ defined in (E.1), we have

$$\sum_{i=1}^{d} \lambda_i w_i^2 \sum_{k=0}^{t-1} \left( \mathbf{A}_i^k \begin{bmatrix} 1 \\ 1 \end{bmatrix} \right)_2^2 \leq \frac{14}{\delta} \|\mathbf{w}_0 - \mathbf{w}^*\|_{\mathbf{I}_{0:\hat{k}}}^2 + \frac{10}{1-c} \|\mathbf{w}_0 - \mathbf{w}^*\|_{\mathbf{H}_{\hat{k}:k^\dagger}}^2$$
$$+ \frac{1-c}{q - c\delta} \|\mathbf{w}_0 - \mathbf{w}^*\|_{\mathbf{I}_{k^\dagger:k^*}}^2 + 4t \|\mathbf{w}_0 - \mathbf{w}^*\|_{\mathbf{H}_{k^*:\infty}}^2.$$

*Proof.* By Lemma K.8, specifically for $k^\dagger < i \leq k^*$, we have

$$\sum_{k=0}^{t-1} \left( \mathbf{A}_i^k \begin{bmatrix} 1 \\ 1 \end{bmatrix} \right)_2^2 \leq \frac{1-c}{(q-c\delta)\lambda_i} \left[ 1 - \left( 1 - 2\frac{q-c\delta}{1-c}\lambda_i \right)^{2t} \right] \leq \frac{1-c}{(q-c\delta)\lambda_i},$$

where the inequality holds because $1 - \left( 1 - 2\frac{q-c\delta}{1-c}\lambda_i \right)^{2t} \leq 1$; For $i > k^*$,

$$\sum_{k=0}^{t-1} \left( \mathbf{A}_i^k \begin{bmatrix} 1 \\ 1 \end{bmatrix} \right)_2^2 \leq \frac{1-c}{(q-c\delta)\lambda_i} \left[ 1 - \left( 1 - 2\frac{q-c\delta}{1-c}\lambda_i \right)^{2t} \right] \leq \frac{1-c}{(q-c\delta)\lambda_i} \cdot 4t \frac{(q-c\delta)\lambda_i}{1-c} = 4t,$$

where the inequality holds because $1 - \left( 1 - 2\frac{q-c\delta}{1-c}\lambda_i \right)^{2t} \leq 4t\frac{(q-c\delta)\lambda_i}{1-c}$. Therefore,

$$\sum_{i=1}^{d} \lambda_i w_i^2 \sum_{k=0}^{t-1} \left( \mathbf{A}_i^k \begin{bmatrix} 1 \\ 1 \end{bmatrix} \right)_2^2$$
$$\leq \sum_{i \leq k^\ddagger} \lambda_i w_i^2 \cdot \frac{7}{2\delta\lambda_i} + \sum_{k^\ddagger < i \leq \hat{k}} \lambda_i w_i^2 \cdot \frac{14}{\delta\lambda_i} + \sum_{\hat{k} < i \leq k^\dagger} \lambda_i w_i^2 \cdot \frac{10}{1-c}$$
$$+ \sum_{k^\dagger < i \leq k^*} \lambda_i w_i^2 \cdot \frac{1-c}{(q-c\delta)\lambda_i} + \sum_{i > k^*} \lambda_i w_i^2 \cdot 4t$$
$$= \frac{7}{2\delta} \|\mathbf{w}_0 - \mathbf{w}^*\|_{\mathbf{I}_{0:k^\ddagger}}^2 + \frac{14}{\delta} \|\mathbf{w}_0 - \mathbf{w}^*\|_{\mathbf{I}_{k^\ddagger:\hat{k}}}^2 + \frac{10}{1-c} \|\mathbf{w}_0 - \mathbf{w}^*\|_{\mathbf{H}_{\hat{k}:k^\dagger}}^2$$
$$+ \frac{1-c}{q-c\delta} \|\mathbf{w}_0 - \mathbf{w}^*\|_{\mathbf{I}_{k^\dagger:k^*}}^2 + 4t \sum_{i > k^*} \|\mathbf{w}_0 - \mathbf{w}^*\|_{\mathbf{H}_{k^*:\infty}}^2$$
$$\leq \frac{14}{\delta} \|\mathbf{w}_0 - \mathbf{w}^*\|_{\mathbf{I}_{0:\hat{k}}}^2 + \frac{10}{1-c} \|\mathbf{w}_0 - \mathbf{w}^*\|_{\mathbf{H}_{\hat{k}:k^\dagger}}^2$$
$$+ \frac{1-c}{q-c\delta} \|\mathbf{w}_0 - \mathbf{w}^*\|_{\mathbf{I}_{k^\dagger:k^*}}^2 + 4t \|\mathbf{w}_0 - \mathbf{w}^*\|_{\mathbf{H}_{k^*:\infty}}^2,$$

where the second inequality holds because $7/2 < 14$. $\square$

**Lemma K.12.** For any $0 < x_1, x_2 \leq \theta < 1$ ($x_1 \neq x_2$) and integer $t \geq 0$, we have

$$\frac{x_2^t - x_1^t}{x_2 - x_1} \leq \frac{\theta^t - x_1^t}{\theta - x_1}.$$

*Proof.* The lemma holds trivially for $t = 0$. For $t \geq 1$, we have

$$\frac{x_2^t - x_1^t}{x_2 - x_1} = \sum_{k=0}^{t-1} x_1^k x_2^{t-1-k} \leq \sum_{k=0}^{t-1} x_1^k \cdot \theta^{t-1-k} = \frac{\theta^t - x_1^t}{\theta - x_1},$$

where the inequality holds because $x_2 \leq \theta$. $\square$

**Lemma K.13.** Suppose $x_1, x_2$ are complex eigenvalues of $\mathbf{A}_i$ for $k^\ddagger < i \leq k^\dagger$. Then for any $t \geq 0$,

$$\left| \frac{x_2^t - x_1^t}{x_2 - x_1} \right| \leq t[c(1 - \delta\lambda_i)]^{(t-1)/2}.$$

*Proof.* We have

$$\left| \frac{x_2^t - x_1^t}{x_2 - x_1} \right| = \left| \sum_{k=0}^{t-1} x_2^k x_1^{t-1-k} \right| \le \sum_{k=0}^{t-1} |x_2^k| \cdot |x_1^{t-1-k}| = t[c(1 - \delta\lambda_i)]^{(t-1)/2},$$

where the inequality holds due to triangle inequality, and the second equality holds due to Lemma E.2. □

**Lemma K.14.** For any $t \ge 0$, we have

$$t[c(1 - \delta\lambda_i)]^{(t-1)/2} \le \min\left\{ \frac{2}{\delta\lambda_i}, \frac{2}{1-c} \right\}.$$

*Proof.* Note that

$$t[c(1 - \delta\lambda_i)]^{(t-1)/2} = \sum_{k=0}^{t-1} [c(1 - \delta\lambda_i)]^{(t-1)/2} \le \sum_{k=0}^{t-1} [c(1 - \delta\lambda_i)]^{k/2} = \frac{1 - [c(1 - \delta\lambda_i)]^{t/2}}{1 - \sqrt{c(1 - \delta\lambda_i)}}$$

$$\le \frac{1}{1 - \sqrt{c(1 - \delta\lambda_i)}} \le \frac{1}{1 - \sqrt{1 - \delta\lambda_i}} \le \frac{2}{\delta\lambda_i},$$

where the first inequality holds because $c(1 - \delta\lambda_i) \le 1$, the second inequality holds because $1 - [c(1 - \delta\lambda_i)]^{t/2} \le 1$, the third inequality holds because $c \le 1$, and the last inequality holds because $1 - \sqrt{1 - \delta\lambda_i} \ge \delta\lambda_i/2$. Similarly we have

$$t[c(1 - \delta\lambda_i)]^{(t-1)/2} \le \frac{1}{1 - \sqrt{c(1 - \delta\lambda_i)}} \le \frac{1}{1 - \sqrt{c}} \le \frac{2}{1-c},$$

where the second inequality holds because $1 - \delta\lambda_i \le c$, and the last inequality holds because $1 - \sqrt{c} \ge (1-c)/2$. □

