# OpenReview forum: "Risk Bounds of Accelerated SGD for Overparameterized Linear Regression"
_ICLR.cc/2024/Conference — ICLR 2024 poster_

### Official Review · Reviewer_MrsP · 2023-11-01

**Soundness:** 3 good
**Presentation:** 3 good
**Contribution:** 3 good
**Rating:** 6
**Confidence:** 3

**Summary:**

This paper studies the generalization of ASGD for overparameterized linear regression, which is possibly the simplest setting of learning with overparameterization. The authors establish an instance-dependent excess risk bound for ASGD within each eigen-subspace of the data covariance matrix. The theoretical findings show that (i) ASGD outperforms SGD in the subspace of small eigenvalues, exhibiting a faster rate of exponential decay for bias error, while in the subspace of large eigenvalues, its bias error decays slower than SGD; and (ii) the variance error of ASGD is always larger than that of SGD. Our result suggests that ASGD can outperform SGD when the difference between the initialization and the true weight vector is mostly confined to the subspace of small eigenvalues.  Finally, sufficient experiment verify the effectiveness of the theoretical findings.

**Strengths:**

1. The theoretical findings are solid.
2. This paper is well-written and easy to follow.

**Weaknesses:**

The experimental results do not provide strong support for the theoretical findings in this paper.

**Questions:**

1. How is overparameterization reflected in Theorem 4.1?

2. Please provide a more detailed explanation of the challenges encountered in theoretical analysis and how to address them in the submission.

3. Can you provide more instances that satisfy Assumption 3.2?

4. The experiments are not comprehensive enough, for example, the paper finds that the variance error of ASGD is always larger than that of SGD, but there is not enough experimental support for this claim.

---

> ### Author Response · Authors · 2023-11-21
> **Response to Reviewer MrsP**
>
> Thank you for your positive feedback and suggestions!
>
> **Q1**. The experiments are not comprehensive enough, for example, the paper finds that the variance error of ASGD is always larger than that of SGD, but there is not enough experimental support for this claim.
>
> **A1**. Thank you for your suggestion. We have provided additional experiments to justify that the variance error of ASGD is larger than that of SGD. We have also provided additional experiments that compare both the bias error and the variance error of ASGD and SGD for linear regression instances with different spectra of $\mathbf{H}$, including $\lambda_k=k^{-2}$, $\lambda_k=k\log (k+1)$ and $\lambda_k=e^{-k/2}$. In these experiments, we use the same hyperparameters as those used in Section 6. $\mathbf{w}_0$ is initialized as $\mathbf{w}_0=10\cdot\\mathbf{e}\_1$, representing the case where $\mathbf{w}_0-\mathbf{w}^*$ is refined mainly to the subspace of large eigenvalues, or $\mathbf{w}_0=10\cdot\mathbf{e}\_{10}$, representing the case where $\mathbf{w}_0-\mathbf{w}^*$ is refined mainly to the subspace of small eigenvalues. The experimental results are shown in Figures 3-5 in Appendix A. From our experiments, we can observe that the variance error of ASGD is larger than that of SGD, and that the bias error of ASGD is smaller when $\mathbf{w}_0-\mathbf{w}^*$ is refined mainly to the subspace of small eigenvalues. Please refer to Appendix A in the revision for more details.
>
> ---
>
> **Q2**. How is overparameterization reflected in Theorem 4.1?
>
> **A2**. The excess risk bound provided in Theorem 4.1 is not explicitly related to the model dimension $d$. Instead, it depends on $k^*$ which is referred to as the effective dimension. For many data covariance matrix $\mathbf{H}$, even if $d$ is very large or even goes to infinity (i.e., overparameterization regime), $k^*$ can still be small or finite, and the excess risk bound does not go to infinity. This improves previous results that explicitly depend on the model dimension $d$ (Jain et al. 2018). This is how overparameterization is reflected in Theorem 4.1.
>
> ---
>
> **Q3**. Please provide a more detailed explanation of the challenges encountered in theoretical analysis and how to address them in the submission.
>
> **A3**. The most significant challenge is that, different from previous methods analyzing SGD (Zou et al. ,2021; Wu et al. ,2022), the monotonicity of the second moment for the bias error $\mathbf{B}_t$ is violated in ASGD. Therefore, we need to calculate the explicit expression of $\mathbf{A}_i^k$, and analyze quantities like those in Lemma K.5, Lemma K.6 and Lemma K.7.
>
> Another challenge is the identification of the eigenvalue cutoffs $k^\dagger$, $k^\ddagger$, $\hat k$ and $k^*$. This is much more complicated than SGD because $\lambda_i$ significantly affects the decay rate of $\mathbf{A}_i^k$. Specifically, the eigenvalues of $\mathbf{A}_i$ can be complex or real, which also depends on $\lambda_i$. As a comparison, in the SGD results (Zou et al. ,2021; Wu et al. ,2022), there are only two cutoffs for eigenvalue.
>
> Last but not least, we develop a new choice of parameters (Eq. (4.2)) for ASGD in the overparameterized regime. This choice of parameters is obtained through a fine-grained analysis of the effect of fourth-moment (see details in Appendix F.2), where we manage to avoid the model dimension $d$ in our choice of parameters.
>
> ---
>
> **Q4**. Can you provide more instances that satisfy Assumption 3.2?
>
> **A4**. Yes. The first example is: $\mathbf{H}^{-1/2}\mathbf{x}$ is $\sigma^2$-sub-Gaussian, then Assumption 3.2 is satisfied with $\psi=16\sigma^4$ (see Lemma A.1 of Zou et al., 2021). Another example is: $\mathbf{H}^{-1/2}\mathbf{x}$ is $\sigma^2$-sub-exponential, then Assumption 3.2 is also satisfied with $\psi=256\sigma^4$.
>
> ---
>
> Jain et al. "Accelerating stochastic gradient descent for least squares regression." In Conference On Learning Theory, pp. 545-604. PMLR, 2018.
>
> Zou et al. "Benign overfitting of constant-stepsize SGD for linear regression." In Conference on Learning Theory, pp. 4633-4635. PMLR, 2021.
>
> Wu et al. "Last iterate risk bounds of sgd with decaying stepsize for overparameterized linear regression." In International Conference on Machine Learning, pp. 24280-24314. PMLR, 2022.

---

### Official Review · Reviewer_BSyC · 2023-11-01

**Soundness:** 4 excellent
**Presentation:** 3 good
**Contribution:** 2 fair
**Rating:** 6
**Confidence:** 4

**Summary:**

The authors analyze stochastic gradient descent with momentum in the overparametrized linear regression setting and they study excess risk bounds, showing that the variance of this algorithm is never better than for SGD and that the bias is also larger for the subspace of largest eigenvalues. They show that for the subspace of the the lowest eigenvalues, the bias of the algorithm is smaller than that of SGD

**Strengths:**

In the strongly convex regime, the authors get better bias error term than previous work (except for a term, that the authors claim that can be removed but that wasn't removed). They extend the techniques in Jain et al 2018 to the overparametrized setting which allows them to specify what happens in their setting with ASGD
The paper is well written.

**Weaknesses:**

It is known from before in many settings that accelerated gradient descent does work very well with noise, and in fact, this is what is corroborated by this work for the setting of overparametrized linear regression. The variance is shown to be greater, the bias is shown to be greater for the subspace of large eigenvalues, which are the most important ones. The claim in the abstract about ASGD outperforming SGD if the initialization minus optimizers lives in the subspace of the small eigenvalues is true but a bit useless, since it is very unlikely this happens. It is a low dimensional subspace and most of the rest of the space is dominated by the large eigenvalues.  It is informative and of value to have all of the details in this setting that are provided in this work, but the results are weak, essentially a negative result that could have maybe been anticipated for this kind of algorithm.

Edit: thanks for the reply to the questions. I edited my score.

**Questions:**

Can you provide any examples of settings in which you can guarantee you can initialize to be in the good regime that you show for ASGD, i.e. when $w_0 -w^\ast$ is essentially aligned with the subspace associated to small eigenvalues?

---

> ### Author Response · Authors · 2023-11-21
> **Response to Reviewer BSyC**
>
> Thank you for your helpful feedback.
>
>
> **Q1**. The claim in the abstract about ASGD outperforming SGD if the initialization minus optimizers lives in the subspace of the small eigenvalues is true but a bit useless, since it is very unlikely this happens. It is a low dimensional subspace and most of the rest of the space is dominated by the large eigenvalues.
>
> **A1**. We would like to emphasize that this result in actually useful in the sense that (1) even for the simplest possible problem of linear regression, no prior work has demonstrated when ASGD can outperform SGD, and we are the first to give a quantitative condition, which deepens the understanding of ASGD vs. SGD in terms of generalization performance; (2) It is not “unlikely” that $\mathbf{w}_0-\mathbf{w}^*$ aligns well with the subspace of small eigenvalues. In practice, one often uses zero initialization or randomly initialization $\mathbf{w}_0$ that is close to zero. In this case, when the ground truth $\mathbf{w}^*$ is more aligned with the subspace of small eigenvalues, we will have $\mathbf{w}_0-\mathbf{w}^*$ is more aligned well with the subspace of small eigenvalues.
>
>
> ---
>
>
> **Q2**. It is informative and of value to have all of the details in this setting that are provided in this work, but the results are weak, essentially a negative result that could have maybe been anticipated for this kind of algorithm.
>
> **A2**. We would like to emphasize that our work not only provides a “negative” result, but also a “positive” result, which identifies a class of linear regression instances for which ASGD can outperform SGD (in Theorem 5.1). Moreover, we believe that our result is strong: it is the first result characterizing the excess risk of ASGD in the overparameterized regime. From a technical point of view, our result is also highly nontrivial to obtain (See the key techniques in Section 7).
>
> ---
>
> **Q3**. Can you provide any examples of settings in which you can guarantee you can initialize to be in the good regime that you show for ASGD, i.e. when $\mathbf{w}_0-\mathbf{w}^*$ is essentially aligned with the subspace associated to small eigenvalues?
>
> **A3**. Yes, we can first consider a $2$-dimensional model where $\lambda_1=1$ and $\lambda_2=0.01$, the ground truth weight vector is $\mathbf{w}^*=(1, 10)^\top$, and the initialization is $\mathbf{w}_0=(0, 0)^\top$. This is exactly the case where $\mathbf{w}_0-\mathbf{w}^*$ lies primarily in the subspace of small eigenvalues of the data covariance matrix. This type of ill-posed model is quite common in practice. It is also very easy to extend this example to the high-dimensional settings.

---

> ### Author Response · Authors · 2023-11-22
> **Gentle Reminder**
>
> Dear Reviewer BSyC,
>
> Thank you again for your insightful comments. We would like to follow up with you and address any outstanding questions if you still have. In our response, we have provided a representative linear regression instance where $\mathbf{w}_0-\mathbf{w}^*$ is mainly refined to the space of large eigenvalues of $\mathbf{H}$. Please let us know if you have any other suggestions.

---

> > ### Comment · Reviewer_BSyC · 2023-11-22
> >
> > When I asked for examples of settings I was not referring to this. It is clear that one can set up a synthetic example that satisfies the condition tautologically. A priori I don't know if my solution is going to be aligned with the smallest eigenvalues, so if I were to run SGD or ASGD, I'd go for SGD or I would apply a random rotation and then run SGD. I was asking whether you could give any examples of problems, that maybe because of their nature they have some structure so we know this low-eigenvalue alignement condition is going to be satisfied.

---

> > > ### Author Response · Authors · 2023-11-22
> > > **Response to Reviewer BSyC**
> > >
> > > Many thanks for your feedback!
> > >
> > > We would like to clarify that the linear regression instance we just provided is a representative one. The two dimensions have equal contributions to the model output because $\mathrm{var}(w_1x_1)=\mathrm{var}(w_2x_2)$, but the scale of the two dimensions are very different.
> > >
> > > In practical applications, it is often the case that two features contribute similarly to a model's output, yet their scales differ significantly. In these instances, the ground truth vector is better aligned with the feature of smaller scale. For example, in a face recognition task, the model may capture both the color of the skin, which is a feature of larger scale, and the shape of the nose, which is a feature of smaller scale. Under these circumstances, the condition is satisfied, making ASGD the preferable option.
> > >
> > > A broader application scenario could be the case of spurious feature, which refers to the potential existence of a large-scale feature that is irrelevant to the task itself but appears frequently within a class. Solely focusing on this large-scale (spurious) feature leads to poor out-of-distribution (OOD) performance. Conversely, greater emphasis should be placed on learning the small-scale (core) feature. The setting of large-scale spurious feature and small-scale spurious feature can also be found in the data model of [1]. In such scenarios, ASGD is more preferable. And the proposed method in [1] for a more robust learning also heavily relies on momentum. Instances of datasets with these characteristics are not rare, including Waterbirds, Colored MNIST, CelebA, CivilComments, etc. For instance, the Waterbirds dataset presents a bird classification task, and is often characterized as containing two main features: the background and the bird, where the background is at larger-scale and spurious. Our analysis applies to these scenarios, indicating that ASGD facilitates a more robust learning than SGD.
> > >
> > > [1] "Robust Learning with Progressive Data Expansion Against Spurious Correlation." Advances in neural information processing systems. 2023.

---

### Official Review · Reviewer_hi74 · 2023-11-05

**Soundness:** 3 good
**Presentation:** 3 good
**Contribution:** 3 good
**Rating:** 6
**Confidence:** 3

**Summary:**

This paper studies the generalization of ASGD for overparameterized linear regression, which is possibly the simplest setting of learning with overparameterization. This paper establishes an instance dependent excess risk bound for ASGD within each eigen-subspace of the data covariance matrix. The analysis shows that (i) ASGD outperforms SGD in the subspace of small eigenvalues, exhibiting a faster rate of exponential decay for bias error, while in the subspace of large eigenvalues, its bias error decays slower than SGD; and (ii) the variance error of ASGD is always larger than that of SGD.
The result suggests that ASGD can outperform SGD when the difference between the initialization and the true weight vector is mostly confined to the subspace of small eigenvalues. Additionally, when the analysis is specialized to linear regression in the strongly convex setting, it yields a tighter bound for bias error than the best-known result.

**Strengths:**

The analysis shows that (i) ASGD outperforms SGD in the subspace of small eigenvalues, exhibiting a faster rate of exponential decay for bias error, while in the subspace of large eigenvalues, its bias error decays slower than SGD; and (ii) the variance error of ASGD is always larger than that of SGD.
The result suggests that ASGD can outperform SGD when the difference between the initialization and the true weight vector is mostly confined to the subspace of small eigenvalues. Additionally, when the analysis is specialized to linear regression in the strongly convex setting, it yields a tighter bound for bias error than the best-known result.

**Weaknesses:**

No

**Questions:**

No

---

> ### Author Response · Authors · 2023-11-21
> **Response to Reviewer hi74**
>
> Thank you very much for your strong support!

---

### Meta-Review · Area_Chair_v7vd · 2023-12-07

**Metareview:**

Summary:
This paper studies the generalization of ASGD for overparameterized linear regression, which is possibly the simplest setting of learning with overparameterization. This paper establishes an instance-dependent excess risk bound for ASGD within each eigen-subspace of the data covariance matrix. The analysis shows that (i) ASGD outperforms SGD in the subspace of small eigenvalues, exhibiting a faster rate of exponential decay for bias error, while in the subspace of large eigenvalues, its bias error decays slower than SGD, and (ii) the variance error of ASGD is always larger than that of SGD. The result suggests that ASGD can outperform SGD when the difference between the initialization and the true weight vector is mostly confined to the subspace of small eigenvalues. Additionally, when the analysis is specialized to linear regression in the strongly convex setting, it yields a tighter bound for bias error than the best-known result.

Strengths:
+ In the strongly convex regime, the authors get better bias error terms than previous work (except for a term, that the authors claim can be removed but that wasn't removed). They extend the techniques in Jain et al 2018 to the over-parametrized setting which allows them to specify what happens in their setting with ASGD
+ The paper is well written.
+ The theoretical findings are solid.
- This paper is well-written and easy to follow.

Weaknesses:
- Most comments were addressed during the discussion period

**Justification For Why Not Higher Score:**

Such a theoretical work might have limited reception in such a wide audience. Better discussions will occur in front of the poster.

**Justification For Why Not Lower Score:**

For the reasons above

---

### Decision · Program_Chairs · 2024-01-16

Accept (poster)